# Unifying the known and unknown microbial coding sequence space

Chiara Vanni[1,2], Matthew S Schechter[1,3], Silvia G Acinas[4], Albert Barberán[5], Pier Luigi Buttigieg[6], Emilio O Casamayor[7], Tom O Delmont[8], Carlos M Duarte[9], A Murat Eren[3,10], Robert D Finn[11], Renzo Kottmann[1], Alex Mitchell[11], Pablo Sánchez[4], Kimmo Siren[12], Martin Steinegger[13,14], Frank Oliver Gloeckner[2,15,16], Antonio Fernàndez-Guerra[1,17]*

[1]Microbial Genomics and Bioinformatics Research G, Max Planck Institute for Marine Microbiology, Bremen, Germany; [2]Jacobs University Bremen, Bremen, Germany; [3]Department of Medicine, University of Chicago, Chicago, United States; [4]Department of Marine Biology and Oceanography, Institut de Ciències del Mar (CSIC), Barcelona, Spain; [5]Department of Environmental Science, University of Arizona, Tucson, United States; [6]Alfred Wegener Institute, Helmholtz Centre for Polar and Marine Research, Alfred Wegener Institute, Bremerhaven, Germany; [7]Center for Advanced Studies of Blanes CEAB-CSIC, Spanish Council for Research, Blanes, Spain; [8]Génomique Métabolique, Genoscope, Institut François Jacob, CEA, CNRS, Univ Evry, Université Paris-Saclay, Evry, France; [9]Red Sea Research Centre and Computational Bioscience Research Center, King Abdullah University of Science and Technology, Thuwal, Saudi Arabia; [10]Josephine Bay Paul Center, Marine Biological Laboratory, Woods Hole, United States; [11]European Molecular Biology Laboratory, European Bioinformatics Institute (EMBL-EBI), Wellcome Genome Campus, Hinxton, United Kingdom; [12]Section for Evolutionary Genomics, The GLOBE Institute, University of Copenhagen, Copenhagen, Denmark; [13]School of Biological Sciences, Seoul National University, Seoul, Republic of Korea; [14]Institute of Molecular Biology and Genetics, Seoul National University, Seoul, Republic of Korea; [15]University of Bremen and Life Sciences and Chemistry, Bremen, Germany; [16]Computing Center, Helmholtz Center for Polar and Marine Research, Bremerhaven, Germany; [17]Lundbeck Foundation GeoGenetics Centre, GLOBE Institute, University of Copenhagen, Copenhagen, Denmark

*For correspondence: antonio.fernandez-guerra@sund.ku.dk

Competing interest: The authors declare that no competing interests exist.

**Abstract** Genes of unknown function are among the biggest challenges in molecular biology, especially in microbial systems, where 40–60% of the predicted genes are unknown. Despite previous attempts, systematic approaches to include the unknown fraction into analytical workflows are still lacking. Here, we present a conceptual framework, its translation into the computational workflow AGNOSTOS and a demonstration on how we can bridge the known-unknown gap in genomes and metagenomes. By analyzing 415,971,742 genes predicted from 1749 metagenomes and 28,941 bacterial and archaeal genomes, we quantify the extent of the unknown fraction, its diversity, and its relevance across multiple organisms and environments. The unknown sequence space is exceptionally diverse, phylogenetically more conserved than the known fraction and predominantly taxonomically restricted at the species level. From the 71 M genes identified to be of unknown function, we compiled a collection of 283,874 lineage-specific genes of unknown function for *Cand.* Patescibacteria (also known as Candidate Phyla Radiation, CPR), which provides a signifi-cant resource to expand our understanding of their unusual biology. Finally, by identifying a target

gene of unknown function for antibiotic resistance, we demonstrate how we can enable the generation of hypotheses that can be used to augment experimental data.

## Editor's evaluation

In this paper, the authors develop a sensitive and specific computational workflow for comprehensively summarizing known and unknown gene content across large collections of genomes and metagenomes. In addition to clustering and categorizing genes on a large scale, the authors show how to use their approach to both explore lineage-specific genes and generate hypotheses for the function of unknown genes.

## Introduction

Thousands of isolate, single-cell, and metagenome-assembled genomes are guiding us toward a better understanding of life on Earth (*Almeida et al., 2019*; *Cross et al., 2019*; *Hug et al., 2016*; *Kopf et al., 2015*; *Pachiadaki et al., 2019*; *Pasolli et al., 2019*; *Sunagawa et al., 2015*). At the same time, the ever-increasing number of genomes and metagenomes, unlocking uncharted regions of life's diversity, (*Brown et al., 2015*; *Eloe-Fadrosh et al., 2016*; *Hug et al., 2016*) are providing new perspectives on the evolution of life (*Parks et al., 2018*; *Spang et al., 2015*). However, our rapidly growing inventories of new genes have a glaring issue: between 40% and 60% cannot be assigned to a known function (*Almeida et al., 2021*; *Bernard et al., 2018*; *Carradec et al., 2018*; *Price et al., 2018*). Current analytical approaches for genomic and metagenomic data (*Chen et al., 2019*; *Franzosa et al., 2018*; *Huerta-Cepas et al., 2017*; *Mitchell et al., 2020*; *Quince et al., 2017*) generally do not include this uncharacterized fraction in downstream analyses, constraining their results to conserved pathways and housekeeping functions (*Quince et al., 2017*). This inability to handle the unknown is an immense impediment to realizing the potential for discovery of microbiology and molecular biology at large (*Bernard et al., 2018*; *Hanson et al., 2009*). Predicting function from traditional single sequence similarity appears to have yielded all it can (*Arnold, 2018*; *Arnold, 1998*; *Brandenberg et al., 2017*), thus several groups have attempted to resolve gene function by other means. Such efforts include combining biochemistry and crystallography *Jaroszewski et al., 2009*; using environmental co-occurrence *Buttigieg et al., 2013*; by grouping those genes into evolutionarily related families (*Bateman et al., 2010*; *Brum et al., 2016*; *Wyman et al., 2018*; *Yooseph et al., 2007*); using remote homologies (*Bitard-Feildel and Callebaut, 2017*; *Lobb et al., 2015*); or more recently using deep learning approaches (*Bileschi et al., 2019*; *Liu, 2017*). In 2018, (*Price et al., 2018*) developed a high-throughput experimental pipeline that provides mutant phenotypes for thousands of bacterial genes of unknown function being one of the most promising methods to tackle the unknown. Despite their promise, experimental methods are labor-intensive and require novel computational methods that could bridge the existing gap between the genes with known and unknown function.

Here, we present a conceptual framework and its translation to a computational workflow that closes the gap by connecting genomic and metagenomic data by the exploitation of groups of homologous genes, and that facilitates the integration of genes of unknown function into ecological, evolutionary, and biotechnological investigations. The conceptual framework is based on the partitioning of the known and unknown fractions into four different categories that reflects the level of darkness (*Figure 1A*). The category "Known" (K) contains these sequences predicted to harbor domains of known function described by Pfam (*Finn et al., 2016*). Some sequences without Pfam domains of unknown function, like the ones encoding for small and intrinsically disordered proteins, can have homology to characterized proteins, hence we classify them as 'Known without Pfam' (KWP). Lastly, sequences that do not meet either of these criteria are classified as "Genomic Unknown" (GU) if they can be found in genomes from reference databases or contain domains of unknown function (DUF); and 'Environmental Unknown' (EU) when only have been observed in environmental samples. By contextualizing the different categories with information from several sources (*Figure 1C*), we hope this will prove an invaluable resource for including genes of unknown function in evolutionary and ecological studies (*Delmont et al., 2022*; *Gaïa et al., 2021*; *Holland-Moritz et al., 2021*) as well as enhancing the current methods for its experimental characterization.

**eLife digest** It is estimated that scientists do not know what half of microbial genes actually do. When these genes are discovered in microorganisms grown in the lab or found in environmental samples, it is not possible to identify what their roles are. Many of these genes are excluded from further analyses for these reasons, meaning that the study of microbial genes tends to be limited to genes that have already been described.

These limitations hinder research into microbiology, because information from newly discovered genes cannot be integrated to better understand how these organisms work. Experiments to understand what role these genes have in the microorganisms are labor-intensive, so new analytical strategies are needed.

To do this, Vanni et al. developed a new framework to categorize genes with unknown roles, and a computational workflow to integrate them into traditional analyses. When this approach was applied to over 400 million microbial genes (both with known and unknown roles), it showed that the share of genes with unknown functions is only about 30 per cent, smaller than previously thought. The analysis also showed that these genes are very diverse, revealing a huge space for future research and potential applications. Combining their approach with experimental data, Vanni et al. were able to identify a gene with a previously unknown purpose that could be involved in antibiotic resistance.

This system could be useful for other scientists studying microorganisms to get a more complete view of microbial systems. In future, it may also be used to analyze the genetics of other organisms, such as plants and animals.

The application of our approach to 415,971,742 genes predicted from 1749 metagenomes and 28,941 bacterial and archaeal genomes revealed that the unknown fraction (1) is smaller than has been previously reported (*Salazar et al., 2019*; *Thomas and Segata, 2019*), (2) is exceptionally diverse, and (3) is phylogenetically more conserved than the known fraction and predominantly taxonomically restricted at the species level. Finally, we show how we can connect all the knowledge produced by our approach to augment the results from experimental data and add context to genes of unknown function through hypothesis-driven molecular investigations.

## Results

### AGNOSTOS, a computational workflow to unify the known and the unknown sequence space

Driven by the concepts defined in the conceptual framework, we developed AGNOSTOS, a computational workflow that infers, validates, refines, and classifies groups of homologous genes or gene clusters (GCs) in the four proposed categories (*Figure 1A*; *Figure 1B*; *Appendix 1—figure 1*). AGNOSTOS produces GCs with a highly conserved intra-homogeneous structure (*Figure 1B*), both in terms of sequence similarity and domain architecture homogeneity; it exhausts any existing homology to known genes and provides a proper delimitation of the unknown genes before classifying each GC in one of the four categories (Materials and methods). In the last step, we decorate each GC with a rich collection of contextual data compiled from different sources or generated by analyzing the GC contents in different contexts (*Figure 1A*). For each GC, we also offer several products that can be used for analytical purposes like improved representative sequences, consensus sequences, sequence profiles for MMseqs2 (*Steinegger and Söding, 2017*) and HHblits (*Steinegger et al., 2019a*), or the GC members as a sequence similarity network (Methods). To complement the collection, we also provide a subset of what we define as *high-quality* GCs. The defining criteria are (1) the representative is a complete gene and (2) more than one-third of genes within a GC are complete genes.

First, we used AGNOSTOS in a metagenomic context to show its ability to process complex and noisy datasets. We explored the unknown sequence space of 1,749 human and marine metagenomes. In total, we predicted 322,248,552 genes from the environmental dataset and assigned Pfam annotations to 44% of them (*Appendix 1—figure 2A*). Next, AGNOSTOS clustered the predicted genes in 32,465,074 GCs and flagged those gene clusters that contain less than ten genes (*Skewes-Cox et al., 2014*). Flagged gene clusters are not processed through the validation workflow because they

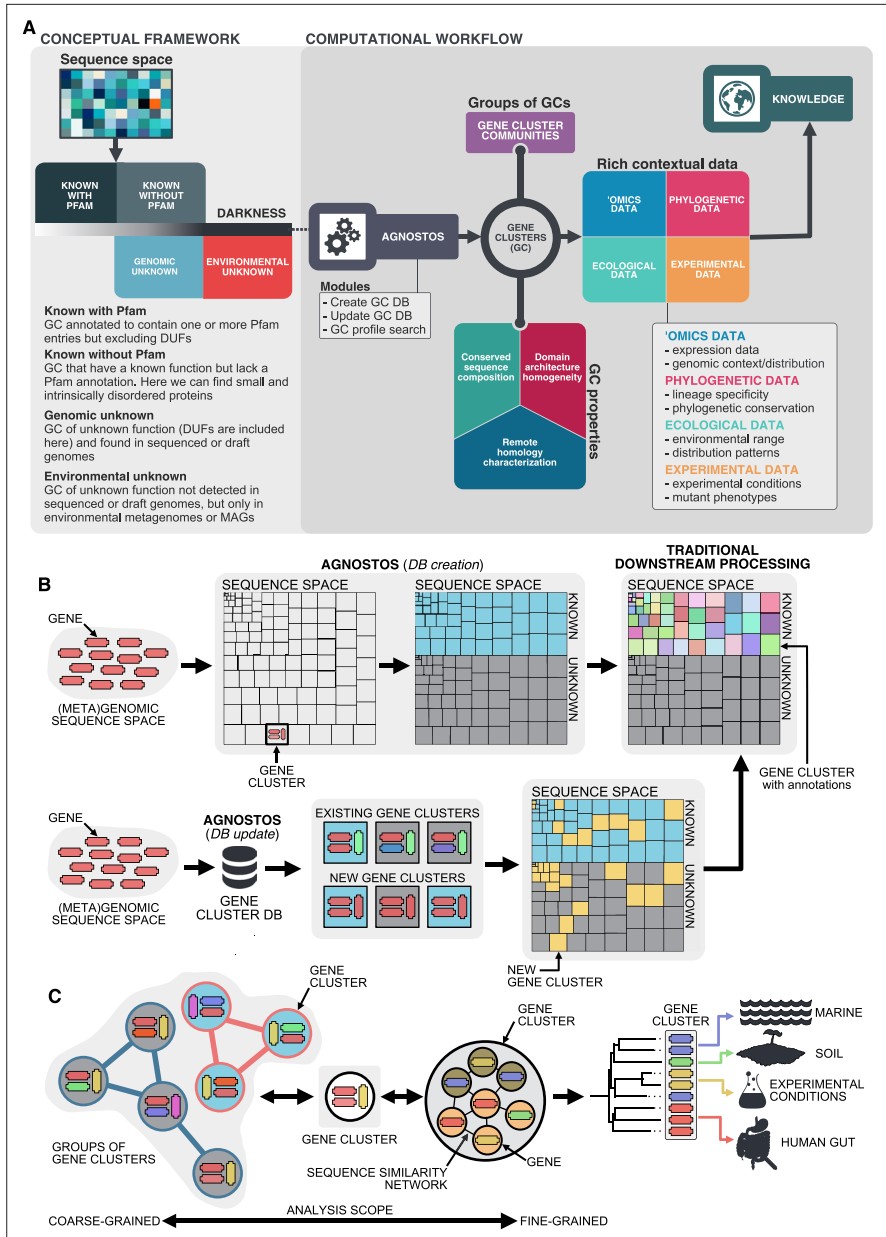

**Figure 1.** Conceptual framework to unify the known and unknown sequence space and integration of the framework in the current analytical workflows. (**A**) Link between the conceptual framework and the computational workflow to partition the sequence space in the four conceptual categories. AGNOSTOS infers, validates and refines the GCs and combines them in gene cluster communities (GCCs). Then, it classifies them in one of the four conceptual categories based on their level of 'darkness'. Finally, we add context to each GC based on several sources of information, providing a robust framework for generating hypotheses that can be used to augment experimental data. (**B**) The computational workflow provides two mechanisms to structure sequence space using GCs, de novo creation of the GCs (*DB creation*), or integrating the dataset in an existing GC database (*DB update*). The structured sequence space can then be plugged into traditional analytical workflows to annotate the genes within each GC of the known fraction. With AGNOSTOS, we provide the opportunity to integrate the unknown fraction into microbiome analyses easily. (**C**) The versatility of the GCs enables analyses at different scales depending on the scope of our experiments. We can group GCs in gene cluster communities based on their shared homologies to perform coarse-grained analyses. On the other hand, we can design fine-grained analyses using the relationships between the genes in a GC, that is detecting network modules in the GC inner sequence similarity network. Additionally, given that GCs are conserved across environments, organisms and experimental conditions give us access to an unprecedented amount of information to design and interpret experimental data.

don't have enough sequences to provide reliable results. We flagged 29,461,177 gene clusters, with 19,911,324 singletons (*Appendix 1—figure 2A*; Appendix 2). A total of 3,003,897 GCs (83% of the original genes) will go through the validation steps. The validation process selected 2,940,257 *good-quality* clusters (*Figure 1B*; *Appendix 1—table 1*; Appendix 3), of which 43% were identified as unknowns after the classification and remote homology refinement steps (*Appendix 1—figure 2A*, Appendix 4).

Lastly, we demonstrate how AGNOSTOS can integrate a new dataset into the already existing metagenomic database by integrating 28,941 genomes from the GTDB_r86 (*Appendix 1—figure 2A*). With this integration we can build links between the environmental and genomic sequence space by expanding the final collection of GCs with the genes predicted from GTDB_r86. Surprisingly, the integration showed that the environmental GCs (human and marine) not only included a large proportion of the genes predicted from GTDB_r86 (72%) but also that most of these genes (90%) are part of the known sequence space (*Appendix 1—figure 2A*; Appendix 5). Only 22% of the GTDB_r86 genes created new GCs (2,400,037), with most of these genes (84%) classified as unknowns by AGNOSTOS (*Appendix 1—figure 2A*; Appendix 5). Finally, only a small proportion of the genes from GTDB_r86 (6%) resulted in singletons (*Appendix 1—figure 2A*; Appendix 5).

The final dataset after being validated and categorized resulted in 5,287,759 GCs (*Appendix 1—figure 2A*), with both datasets sharing only 922,599 GCs (*Appendix 1—figure 2B*). The integration of GTDB_r86 into the metagenomic database increased the proportion of GCs in the unknown sequence space from 43% to the 54%.

Additionally, AGNOSTOS identified a subset of 203,217 *high-quality* GCs (*Appendix 1—table 2*). In these *high-quality* GCs, we identified 12,313 clusters potentially encoding for small proteins ($\leq 50$ amino acids). Most of these GCs are unknown (66%), which agrees with recent findings on novel small proteins from metagenomes (*Sberro et al., 2019*). We also observed that the KWP category contains the largest proportion of incomplete genes (*Appendix 1—table 3*), disrupting the detection and assignment of Pfam domains. But it also incorporates sequences with an unusual amino acid composition that have homology to proteins with high levels of disorder in the DPD database (*Perdigão et al., 2017*) and has characteristic functions of intrinsically disordered proteins (*Habchi et al., 2014*) like cellular processes and signaling as predicted by eggNOG annotations (*Appendix 1—table 4*).

As part of the workflow, each GC is complemented with a rich set of information, as shown in *Figure 1A* (*Supplementary file 2A*; Appendix 6).

## Beyond the twilight zone with AGNOSTOS, communities of gene clusters

To find relationships between gene clusters, we implemented in AGNOSTOS a method to group them in gene cluster communities (GCCs) (*Figure 2A*) using remote homologies. As we are dealing with the unknown, we identified GCCs independently in each category. AGNOSTOS uses the gene clusters from category K as a reference to automatically identify the best parameters for the clustering of the gene cluster homology network (Materials and methods; *Figure 2B*; Appendix 7) and applies the learnt parameters to the other categories. The grouping reduced the final collection of GCs by 87%, producing 673,601 GCCs (Materials and methods; *Figure 2B*; Appendix 7).

We validated our approach to capture remote homologies between related GCs using two well-known gene families present in our environmental datasets, proteorhodopsins (*Béjà et al., 2000*; *Béjà et al., 2001*) and bacterial ribosomal proteins (*Méheust et al., 2019*). Theoretically, we would expect to have one or a very low number of GCCs for each of these gene families.

We identified 64 GCs (12,184 genes) and 3 GCCs (Appendix 7) containing genes classified as proteorhodopsin (PR). One community from the category K contained 99% of the PR annotated genes (*Figure 2C*), except 85 genes taxonomically annotated as viral and assigned to the *PR Supercluster I* (*Boeuf et al., 2015*) within two GU communities (five GU gene clusters; Appendix 7).

For the ribosomal proteins, the results were not so satisfactory. We identified 1843 GCs (781,579 genes) and 98 GCCs. The number of GCCs is larger than the expected number of ribosomal protein families (16) used for validation. When we used *high-quality* GCs (Appendix 7), we got closer to the expected number of GCCs (*Figure 2D*). With this subset, we identified 26 GCCs and 145 GCs (1687 genes). The cross-validation of our method against the approach used in *Méheust et al., 2019* (Appendix 7) confirms the intrinsic complexity of analyzing metagenomic data. Both approaches

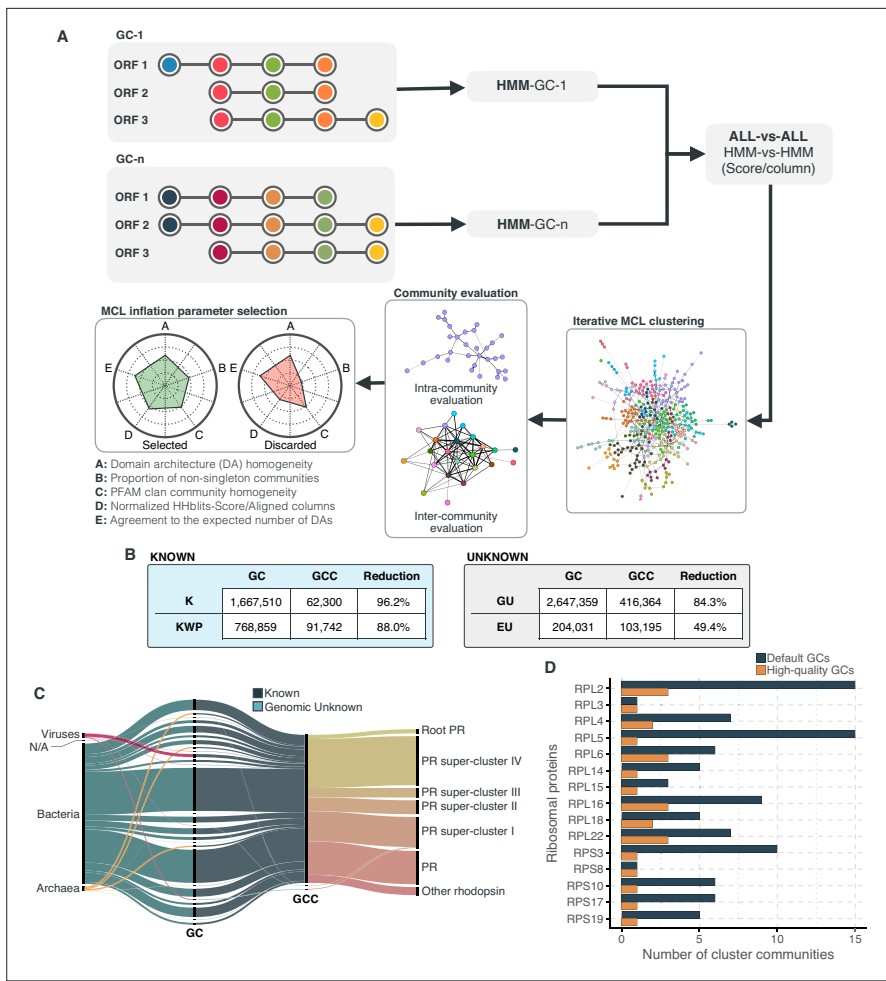

**Figure 2.** Overview and validation of the workflow to aggregate GCs in communities. (**A**) We inferred a gene cluster homology network using the results of an all-vs-all HMM gene cluster comparison with HHBLITS. The edges of the network are based on the HHblits-score/Aligned-columns. Communities are identified by an iterative screening of different MCL inflation parameters and evaluated using five different metrics that consider the inter- and intra-community properties. (**B**) Comparison of the number of GCs and GCCs for each of the functional categories. (**C**) Validation of the GCCs inference based on the environmental genes annotated as proteorhodopsins. Ribbons in the alluvial plot are genes, and each stacked bar corresponds (from left to right) to the (1) gene taxonomic classification at the domain level, (2) GC membership, (3) GCC membership and (4) MicRhoDE operational classification. (**D**) Validation of the GCCs inference based on ribosomal proteins based on standard and high-quality GCs.

showed a high agreement in the GCCs identified (*Appendix 7—table 2*). Still, our method inferred fewer GCCs for each of the ribosomal protein families (*Appendix 7—figure 4*), coping better with the complexities of a metagenomic setup, such as incomplete genes (*Appendix 1—table 5*).

## AGNOSTOS uncovers a smaller yet highly diverse unknown sequence space

Among our primary design goals in developing AGNOSTOS is to unearth ecological and evolutionary information while unifying the known and unknown sequence space of genomes and metagenomes. One can use AGNOSTOS GCs and GCCs as analogs to operational protein families (*Schloss and Handelsman, 2008*) with the benefits of using high-quality gene clusters and remote homology searches.

Our exhaustive analysis of the AGNOSTOS inferred GCs and GCCs described in the following sections, revealed that the unknown sequence space is smaller than what has been reported so far

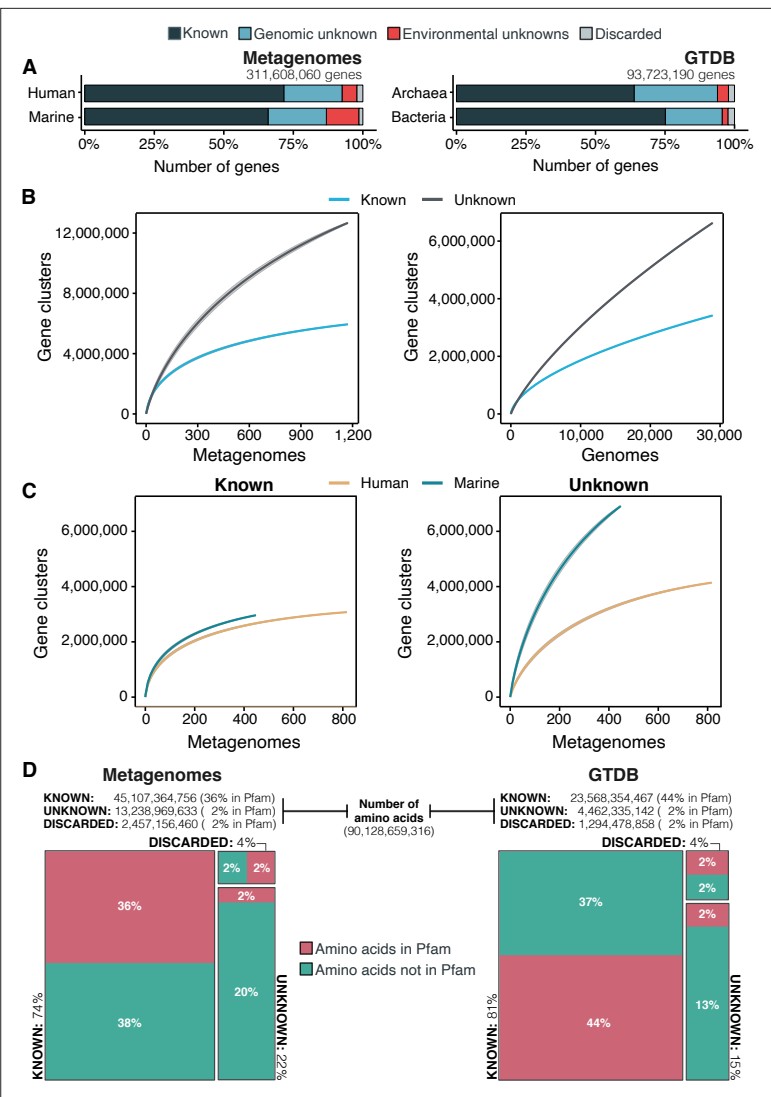

**Figure 3.** The extent of the known and unknown sequence space. (**A**) Proportion of genes in the known and unknown. (**B**) Accumulation curves for the known and unknown sequence space at the GC- level for the metagenomic and genomic data. from TARA, MALASPINA, OSD2014 and HMP-I/II projects. (**C**) Collector curves comparing the human and marine biomes. Colored lines represented the mean of 1000 permutations and shaded in gray the standard deviation. Non-abundant singleton clusters were excluded from the accumulation curves calculation. (**D**) Amino acid distribution in the known and unknown sequence space. In all cases, the four categories have been simplified as known (K, KWP) and unknown (GU, EU).

through traditional genomic and metagenomic analysis approaches, where the unknown fraction can reach up to 60% in marine metagenomes (*Salazar et al., 2019*) or up to 40% in human metagenomes (*Thomas and Segata, 2019*, *Figure 3A*). Our workflow recruited as much as 71% of genes in human-related metagenomic samples and 65% of the genes in marine metagenomes into the known sequence space. In both human and marine microbiomes, the genomic unknown fraction showed a similar proportion of genes (21%, *Figure 3A*). The number of genes corresponding to EU gene clusters is higher in marine metagenomes; 12% of the genes are part of this GC category. We obtained a comparable result when we evaluated the genes from the GTDB_r86, 75% of bacterial and 64% of archaeal genes were part of the known sequence space. Archaeal genomes contained more unknowns than those from Bacteria, where 30% of the genes are classified as genomic unknowns in Archaea, and only 20% in Bacteria (*Figure 3A*; *Supplementary file 1B*).

To further evaluate the differences between the known and unknown sequence space, we calculated the accumulation rates of GCs and GCCs combining the categories K and KWP as knowns, and GU and EU as unknowns to get a general overview of both fractions. For the metagenomic dataset we used 1264 metagenomes (18,566,675 GCs and 282,580 GCCs) and for the genomic dataset 28,941 genomes (9,586,109 GCs and 496,930 GCCs). The rate of accumulation of unknown GCs was three times higher than the known (2 X for the genomic), and in both cases the curves were far from reaching a plateau (*Figure 3B*). This is not the case for the GCC accumulation curves (*Appendix 1—figure 4B*), which reached a plateau.

The accumulation rate is largely determined by the number of singletons, especially singletons from EUs (Appendix 8 and *Appendix 1—figure 5*). While the accumulation rate of known GCs between marine and human metagenomes is almost identical, there are striking differences for the unknown GCs (*Figure 3C*). These differences are maintained even when we remove the virus-enriched samples from the marine metagenomes (*Appendix 1—figure 4A*). Although the marine metagenomes include a large variety of environments, from coastal to the deep sea, the known space remains quite constrained.

Next, we wanted to know how much of the sequence space we integrated with AGNOSTOS was found in other databases (Appendix 9). Despite only including marine and human metagenomes in our database, we already cover, in average, 76% of the sequence space of seven datasets spanning different databases and environments (*Appendix 9—figure 1*). By screening MGnify (*Mitchell et al., 2020*) (release 2018_09; 11 biomes; 843,535,6116 proteins) we identified freshwater, soil and human non-digestive as the biomes less covered by our data (*Appendix 1—figure 6*). Two of the seven analyzed datasets are designed to study genes of unknown function (*Appendix 9—table 1*). On *Wyman et al., 2018*, where they defined Function Unknown Families of homologous proteins (FunkFams), we identified 20% of their FunkFams to be members of the known sequence space. On *Price et al., 2018*, we classified as known, 44% of the genes of unknown function used in their experimental conditions.

One indirect consequence of our approach is that we can provide a detailed view of the sequence space at the amino acid level. We estimated the number of amino acids belonging to the known (K and KWP combined) or unknown (GU and EU combined) sequence space, and how many of these amino acids are contained in a Pfam domain. From the 90,128,659,316 amino acids analyzed, most of the amino acids in metagenomes (74%) and genomes (80%) are in the known sequence space (*Figure 3D*; *Supplementary file 1B*) while only 22% in metagenomes and 15% in genomes are part of the unknowns. In both cases, approximately 40% of the amino acids in the known sequence space were part of a Pfam domain (*Figure 3D*; *Supplementary file 1B*). While this result is expected based on the large number of genes present in the known space (*Figure 3A*), what is surprising is the low proportion of amino acids (2%) corresponding to DUF Pfam domains in genomes and metagenomes (*Figure 3D*). If we use as reference the proportions of amino acids observed in the known sequence space, we can hypothesize that there are still many DUFs to be unearthed. With AGNOSTOS and its thorough validation and characterization of the genomic and environmental unknowns, we provide the basic building blocks (gene clusters' multiple sequence alignments) to identify conserved regions that might become new potential DUFs.

## The unknown sequence space has a limited ecological distribution in human and marine environments

Although the role of the unknown fraction in the environment is still a mystery, the large number of gene counts and abundance observed underlines its inherent ecological relevance (*Figure 4A*). In some metagenomes, the genomic unknown fraction can account for more than 40% of the total gene abundance observed (*Figure 4A*). The environmental unknown fraction is also relevant in several samples, where singleton GCs are the majority (*Figure 4A*). We identified two metagenomes with an unusual composition in terms of environmental unknown singletons. The marine metagenome corresponds to a sample from Lake Faro (OSD42), a meromictic saline with a unique extreme environment where Archaea plays an important role (*La Cono et al., 2013*). The HMP metagenome (SRS143565) that corresponds to a human sample from the right cubital fossa from a healthy female subject. To understand this unusual composition, we should perform further analyses to discard potential technical artifacts like sample contamination.

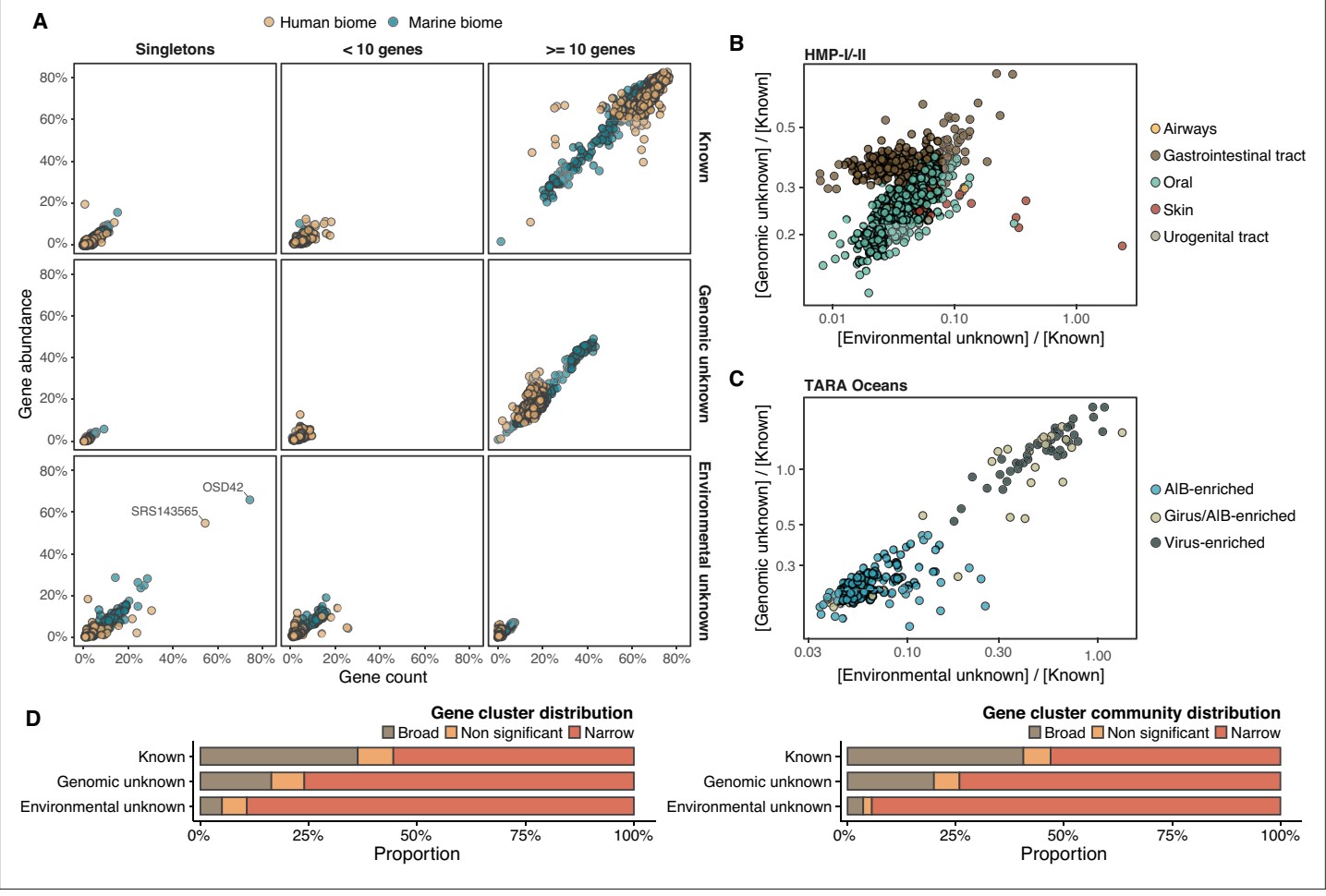

**Figure 4.** Distribution of the unknown sequence space in the human and marine metagenomes. (**A**) Ratio between the proportion of the number of genes and their estimated abundances per cluster category and biome. Columns represented in the facet depicts three cluster categories based on the size of the clusters. (**B**) Relationship between the ratio of Genomic unknowns and Environmental unknowns in the HMP-I/II metagenomes. Gastrointestinal tract metagenomes are enriched in Genomic unknown sequences compared to the other body sites. (**C**) Relationship between the ratio of Genomic unknowns and Environmental unknowns in the TARA Oceans metagenomes. Girus- and virus-enriched metagenomes show a higher proportion of both unknown sequences (genomic and environmental) than the Archaea|Bacteria enriched fractions. (**D**) Environmental distribution of GCs and GCCs based on Levin's niche breadth index. We obtained the significance values after generating 100 null gene cluster abundance matrices using the *quasiswap* algorithm.

The ratio between the unknown and known GCs is useful to reveal which metagenomes are enriched in GCs of unknown function (upper left quadrant in *Figure 4B–C*) and it can be used as a proxy to assess the sequence contained in a metagenome. In human metagenomes, this ratio can distinguish between body sites, with the gastrointestinal tract, an ecologically complex environment (*Qin et al., 2010*), significantly enriched with genomic unknowns. Furthermore, it is not surprising that the human and marine metagenomes with the largest ratio of unknowns are those samples enriched with viral sequences. Specifically, in the HMP metagenomes are those samples identified to contain crAssphages (*Dubinkina et al., 2016*; *Edwards et al., 2019*) and HPV viruses (*Ma et al., 2014*, *Supplementary file 1C*; *Appendix 1—figure 7*). In marine metagenomes (*Figure 4C*), the highest ratio in genomic and environmental unknowns correspond to the ones enriched with viruses and giant viruses.

We performed a large-scale analysis to investigate the occurrence patterns of the GCs in the environment by analyzing their abundance and distribution breadth. The narrow distribution of the unknown fraction (*Figure 4D*) suggests that these GCs might provide a selective advantage and be necessary to adapt to specific environmental conditions. But the pool of broadly distributed

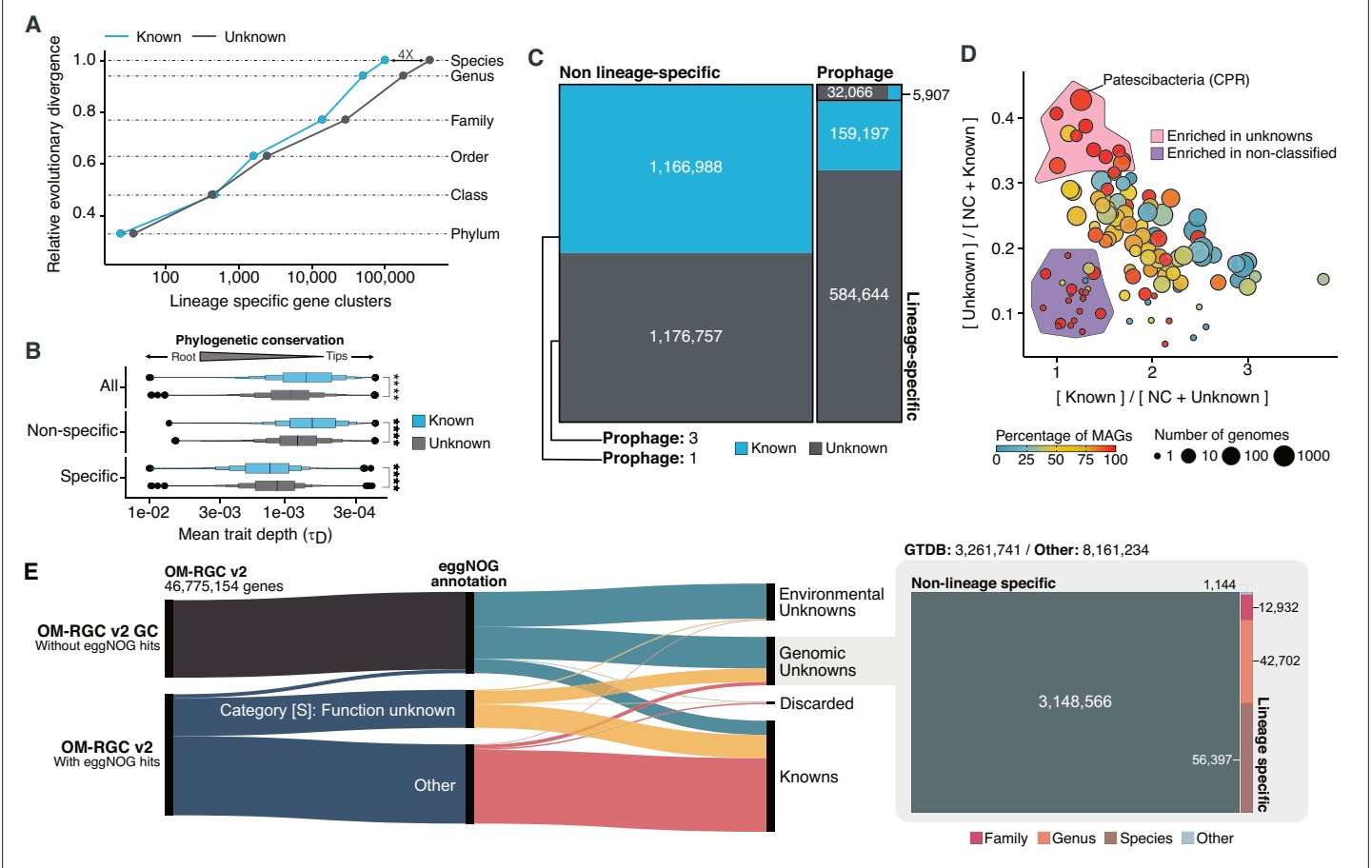

**Figure 5.** Phylogenomic exploration of the unknown sequence space. (**A**) Distribution of the lineage-specific GCs by taxonomic level. Lineage-specific unknown GCs are more abundant in the lower taxonomic levels (genus, species). (**B**) Phylogenetic conservation of the known and unknown sequence space in 27,372 bacterial genomes from GTDB_r86. We observe differences in the conservation between the known and the unknown sequence space for lineage- and non-lineage specific GCs (paired Wilcoxon rank-sum test; all p-values < 0.0001). (**C**) The majority of the lineage-specific clusters are part of the unknown sequence space, and only a small proportion was found in prophages present in the GTDB_r86 genomes. (**D**) Known and unknown sequence space of the 27,732 GTDB_r86 bacterial genomes grouped by bacterial phyla. Phyla are partitioned based on the ratio of known to unknown GCs and vice versa. Phyla enriched in MAGs have higher proportions in GCs of unknown function. Phyla with a high proportion of non-classified clusters (NC; discarded during the validation steps) tend to contain a small number of genomes. (**E**) The alluvial plot's left side shows the uncharacterized (OM-RGC v2 GC) and characterized (OM-RGC v2) fraction of the gene catalog. The functional annotation is based on the eggNOG annotations provided by **Salazar et al., 2019**. The right side of the alluvial plot shows the new organization of the OM-RGC v2 sequence space based on the approach described in this study. The treemap in the right links the metagenomic and genomic space adding context to the unknown fraction of the OM-RGC v2.

environmental unknowns is the most exciting result. We identified traces of potential ubiquitous organisms left uncharacterized by traditional approaches, as more than 80% of these GCs cannot be associated with a metagenome-assembled genome (MAG) (**Appendix 1—table 6**, Appendix 10).

## The genomic unknown sequence space is lineage-specific

With the inclusion of the genomes from GTDB_r86, we have access to a phylogenomic framework that can be used to assess how taxonomically restricted is a GC within a lineage, hereafter referred to as lineage-specific genes (**Johnson, 2018**; **Mendler et al., 2019**) and how conserved (phylogenetic conservation) a GC is across the different clades in the GTDB_r86 phylogenomic tree (**Martiny et al., 2013**). We identified 781,814 lineage-specific GCs and 464,923 phylogenetically conserved (p < 0.05) GCs in Bacteria (**Supplementary file 1D**; Appendix 11 for Archaea). The number of lineage-specific GCs increases with the Relative Evolutionary Distance (**Parks et al., 2018**, **Figure 5A**) and differences between the known and the unknown fraction start to be evident at the family level resulting in 4 X

more unknown lineage-specific GCs at the species level. In general terms, the unknown GCs are more phylogenetically conserved (GCs shared among members of deep clades) than the known (*Figure 5B*, p < 0.0001), revealing the importance of the genome's uncharacterized fraction. However, the lineage-specific unknown GCs are less phylogenetically conserved (*Figure 5B*) than the known, agreeing with the large number of lineage-specific GCs observed at genus and species level (*Figure 5A*).

One potential confounding factor that might contribute to inflate the number of lineage-specific GCs in the unknown fraction, is the presence of prophages owing to their potential host specificity (*Ross et al., 2016*). To discard the possibility that the lineage-specific GCs of unknown function have a viral origin, we screened all GTDB_r86 genomes for prophages. We only found 37,163 lineage-specific GCs (86% of unknown function) in prophage genomic regions.

After unveiling the potential relevance of the GCs of unknown function in bacterial genomes, we identified phyla in GTDB_r86 enriched with these types of clusters. A clear pattern emerged when we partitioned the phyla based on the ratio of known to unknown GCs and vice versa (*Figure 5D*), the phyla with a larger number of MAGs are enriched in GCs of unknown function (*Figure 5D*). Phyla with a high proportion of non-classified GCs (those discarded during the validation steps) contain a small number of genomes and are primarily composed of MAGs. These groups of phyla highly enriched in unknowns and represented mainly by MAGs include newly described phyla such as *Cand*. Riflebacteria and *Cand*. Patescibacteria (*Anantharaman et al., 2018*; *Brown et al., 2015*; *Rinke et al., 2013*), both with the largest unknown to known ratio. We performed an in-depth exploration of the *Cand*. Patescibacteria phylum, and we provide a collection of 54,343 lineage-specific GCs (283,874 genes) of unknown function at different taxonomic level resolutions (*Appendix 1—table 7*; Appendix 12).

One of the strengths of AGNOSTOS is the possibility of bridging genomic and metagenomic data and simultaneously unifying the known and unknown sequence space. We further demonstrated this by integrating the new Ocean Microbial Reference Gene Catalog (*Salazar et al., 2019*, OM-RGC v2) into our database. We assigned 26,170,875 genes to known GCs, 11,422,975 to genomic unknowns, 8,661,221 to environmental unknown and 520,083 were discarded. From the 11,422,975 genes classified as genomic unknowns, we could associate 3,261,741 to a GTDB_r86 genome and we identified 113,175 as lineage-specific. The alluvial plot in *Figure 5E* depicts the new organization of the OM-RGC v2 after being integrated into our framework and how we can provide context to the two original types of unknowns in the OM-RGC (those annotated as category S in eggNOG [*Huerta-Cepas et al., 2019*] and those without known homologs in the eggNOG database [*Salazar et al., 2019*]) that can lead to potential experimental targets at the organism level to complement the metatranscriptomic approach proposed by *Salazar et al., 2019*.

## A structured sequence space augments the interpretation of experimental data

We selected one of the experimental conditions tested in *Price et al., 2018* to demonstrate the potential of our approach to augment experimental data. We compared the fitness values in plain rich medium with added Spectinomycin dihydrochloride pentahydrate to the fitness in plain rich medium (LB) in *Pseudomonas fluorescens FW300-N2C3* (*Figure 6A*). This antibiotic inhibits protein synthesis and elongation by binding to the bacterial 30 S ribosomal subunit and interferes with the peptidyl tRNA translocation. We identified the gene with locus id AO356_08590 that presents a strong phenotype (fitness = –3.1; t = –9.1) and has no known function. This gene belongs to the genomic unknown GC GU_19737823. We can track this GC into the environment and explore the occurrence in the different samples we have in our database. As expected, the GC is mostly found in non-human metagenomes (*Figure 6B*) as *Pseudomonas* are common inhabitants of soil and water environments (*Heffernan et al., 2009*). However, finding this GC also in human-related samples is very interesting due to the potential association of *P. fluorescens* and human disease where Crohn's disease patients develop serum antibodies to this microbe (*Scales et al., 2014*). We can add another layer of information to the selected GC by looking at the associated remote homologs in the GCC GU_c_21103 (*Figure 6C*). We identified all the genes in the GTDB_r86 genomes that belong to the GCC GU_c_21103 (*Supplementary file 1E*) and explored their genomic neighborhoods. All members from GU_c_21103 are constrained to the class *Gammaproteobacteria*, and interestingly GU_19737823 is mostly exclusive to

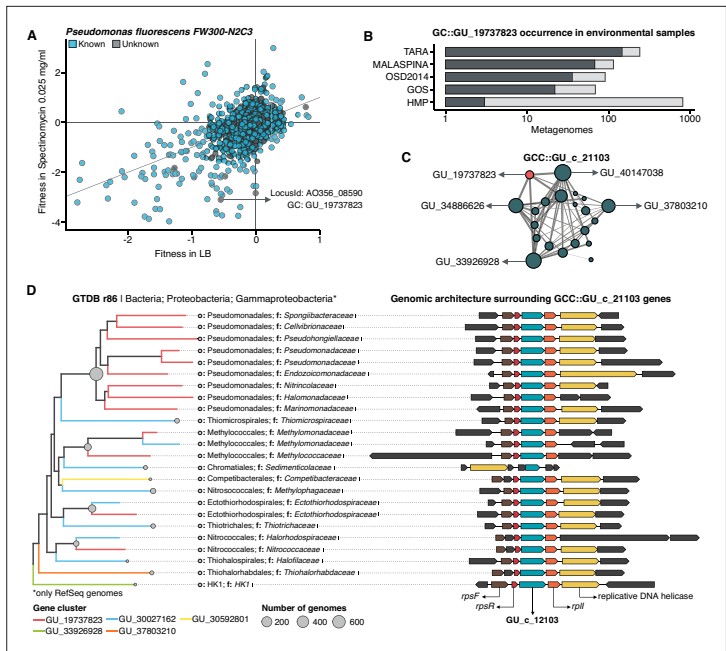

**Figure 6.** Augmenting experimental data with GCs of unknown function. (**A**) We used the fitness values from the experiments from *Price et al., 2018* to identify genes of unknown function that are important for fitness under certain experimental conditions. The selected gene belongs to the genomic unknown GC GU_19737823 and presents a strong phenotype (fitness = –3.1; t = –9.1) (**B**) Occurrence of GU_19737823 in the metagenomes used in this study. Darker bars depict the number of metagenomes where the GC is found. (**C**) GU_19737823 is a member of the GCC GU_c_21103. The network shows the relationships between the different GCs members of the gene cluster community GU_c_21103. The size of the node corresponds to the node degree of each GC. Edge thickness corresponds to the bitscore/column metric. Highlighted in red is GU_19737823. (**D**) We identified all the genes in the GTDB_r86 genomes that belong to the GCC GU_c_21103 and explored their genomic neighborhoods. GU_c_21103 members were constrained to the class *Gammaproteobacteria*, and GU_19737823 is mostly exclusive to the order *Pseudomonadales*. The gene order in the different genomes analyzed is highly conserved, finding GU_19737823 after the *rpsF::rpsR* operon and before *rplI*. *rpsF* and *rpsR* encode for the *30 S ribosomal protein S6* and *30 S ribosomal protein S18*, respectively. The GTDB_r86 subtree only shows RefSeq genomes. Branch colors correspond to the different GCs found in GU_c_21103. The bubble plot depicts the number of genomes with a gene that belongs to GU_c_21103.

the order *Pseudomonadales*. The gene order in the different genomes analyzed is highly conserved, finding GU_19737823 after the *rpsF::rpsR* operon and before *rplI*. *rpsF* and *rpsR* encode for 30 S ribosomal proteins, the prime target of spectinomycin. The combination of the experimental evidence and the associated data inferred by our approach provides strong support to generate the hypothesis that the gene AO356_08590 might be involved in the resistance to Spectinomycin.

## Discussion

We describe a new conceptual framework and how it has been implemented in AGNOSTOS, a computational workflow for unifying the known and unknown sequence space. We used this newly developed framework to perform an in-depth exploration of the microbial unknown sequence space, demonstrating that we can link the unknown fraction of metagenomic studies to specific genomes and provide a powerful new approach for hypothesis generation. The framework introduces a subtle change of paradigm compared to traditional approaches where our objective is to provide the best representation of the unknown space. We gear all our efforts toward finding sequences without any evidence of known homologies by pushing the search space beyond the *twilight zone* of sequence similarity (*Rost, 1999*). With this objective in mind, we use gene clusters instead of genes as the fundamental unit to compartmentalize the sequence space owing to their unique properties (*Figure 1B*). Gene clusters (1) provide a structured sequence space that helps to reduce its complexity, (2) are

independent of the known and unknown fraction, (3) are conserved across environments and organisms, and (4) can be used to aggregate information from different sources (*Figure 1A*). Moreover, GCs provide a good compromise in terms of resolution for analytical purposes, and owing to their unique properties, one can perform analyses at different scales. For fine-grained analyses, we can exploit the gene associations within each GC; and for coarse-grained analyses, we can create groups of GCs based on their shared homologies (*Figure 1B*).

AGNOSTOS integrates transparently into the standard operating procedure for analyzing metagenomes (*Quince et al., 2017*) adopted by the microbiome community. It can briefly be summarized into (1) assembly, (2) gene prediction, (3) gene catalog inference, (4) binning, and (5) characterization. AGNOSTOS exploits recent computational developments (*Steinegger and Söding, 2018*; *Steinegger and Söding, 2017*) to maximize the information used when analyzing genomic and metagenomic data. In addition, we provide a mechanism to reconcile top-down and bottom-up approaches, thanks to the well-structured sequence space proposed by our framework. AGNOSTOS can create environmental- and organism-specific variations of a seed database based on gene clusters. Then, it integrates the predicted genes from new genomes and metagenomes and dynamically creates and classifies new GCs with those genes not integrated during the initial step (*Figure 1B*). Afterward, the potential functions of the known GCs can be carefully characterized by incorporating them into the traditional standard operating procedure described previously.

One of the most appealing characteristics of our approach is that the GCs provide unified groups of homologous genes across environments and organisms independently if they belong to the known or unknown sequence space, and we can contextualize the unknown fraction using this genomic and environmental information. Our combination of partitioning and contextualization features a smaller unknown sequence space than previously reported (*Salazar et al., 2019*; *Thomas and Segata, 2019*). On average, only 30% of the genes fall in the unknown fraction for our genomic and metagenomic data. One hypothesis to reconcile this surprising finding is that the methodologies to identify remotely homologous sequences in large datasets were computationally prohibitive until recently. New methods (*Steinegger et al., 2019a*; *Steinegger and Söding, 2017*), like the ones used in AGNOSTOS, enable large-scale remote homology searches. Still, one must apply conservative measures to control the trade-off between specificity and sensitivity to avoid overclassification.

We found that most of the sequence space at the gene and amino acid level is known, both in genomes and metagenomes. However, the GC accumulation curves showed that the unknown fraction is far more diverse than the known. When we combine the high diversity and its narrow ecological distribution, we can unveil the magnitude of the untapped unknown functional fraction and its potential importance for niche adaptation. In a genomic context and after ruling out the effect of prophages, the unknown fraction is predominantly species' lineage-specific and phylogenetically more conserved than the known fraction, supporting the signal observed in the environmental data emphasizing that we should not ignore the unknown fraction. It is worth noting that the high diversity observed in the unknowns only represents the 20% of the amino acids in the sequence space we analyzed, and only 10% of this unknown amino acid space is part of a Pfam domain (DUF and others). This contrasts with the numbers observed in the known sequence space, where Pfam domains include 50% of the amino acids. All this evidence combined strengthens the hypothesis that the genes of unknown function, especially the lineage-specific ones, might be associated with the mechanisms of microbial diversification and niche adaptation due to the constant diversification of gene families and the survival of new gene lineages (*Francino, 2012*; *Muller, 2019*).

Metagenome-assembled genomes are not only unveiling new regions of the microbial universe (42% the genomes in GTDB_r86), but they are also enriching the tree of life with genes of unknown function (*Overmann et al., 2017*). One excellent example is *Cand*. Patescibacteria, more commonly known as Candidate Phyla Radiation (CPR), a phylum that has raised considerable interest due to its unusual biology (*Brown et al., 2015*) and for which we provide an extensive catalog of 283,874 lineage-specific genes of unknown function at different taxonomic level resolutions, which will provide a valuable resource for further research on CPR.

One of the ultimate goals of our approach is to provide a mechanism to unlock the large pool of likely relevant data that remains untapped to analysis and discovery and boost insights from model organism experiments. We demonstrated the value of our approach by identifying a potential target gene of unknown function for antibiotic resistance. Furthermore, the advent of new methods for protein structure prediction, such as AlphaFold2 (*Jumper et al., 2021*), and fast and sensitive comparisons of large sets of structures (*van Kempen et al., 2022*) will make it possible to use our GCCs as starting points for revealing connections between the known and unknown at even deeper levels than those presented here.

But severe challenges remain, such as the dependence on the quality of the assemblies and their gene predictions (*Salzberg, 2019*), as shown by the analysis of the ribosomal protein GCCs where many of the recovered genes are incomplete. While sequence assembly has been an active area of research (*Roumpeka et al., 2017*), this has not been the case for gene prediction methods (*Roumpeka et al., 2017*; *Sommer and Salzberg, 2021*), which are becoming outdated (*Ivanova et al., 2014*) and cannot cope with the current amount of data. Alternatives like protein-level assembly (*Steinegger et al., 2019b*) combined with exploring the assembly graphs' neighborhoods (*Brown et al., 2020*) become very attractive for our purposes. In any case, we still face the challenge of discriminating between genuine and artifactual singletons (*Höps et al., 2018*). There are currently no methods that both provide a plausible solution and are scalable. We devise a potential solution in the recent developments in unsupervised deep learning methods where they use large corpora of proteins to define a language model *embedding* for protein sequences (*Heinzinger et al., 2019*). These models could be applied to predict *embeddings* in singletons, which could be clustered or used to determine their coding potential. Another concern in our approach is that we may artificially inflate the number of GCs. We follow a conservative approach to avoid mixing multi-domain proteins in GCs owing to the fragmented nature of the metagenome assemblies that could result in the split of a GC. However, not only splitting GCs, but also lumping unrelated genes or GCs owing to the use of remote homologies can be problematic. Although we use very sensitive methods to compare profile HMMs to infer GCCs, low sequence diversity in GCs can limit the method effectiveness. Moreover, our approach is affected by the presence and propagation of contamination in reference databases, a significant problem in 'omics (*Breitwieser et al., 2019*; *Steinegger and Salzberg, 2020*). In our case, we only use Pfam (*Finn et al., 2016*) as a source for annotation owing to its high-quality and manual curation process. The categorization process of our GCs depends on the information from other databases, and to minimize the potential impact of contamination, we apply methods that weight the annotations of the identified homologs to discriminate if a GC belongs to the known or unknown sequence space.

The results presented here prove that the integration and the analysis of the unknown fraction are possible. We are unveiling a brighter future, not only for microbiome analyses but also for boosting eukaryotic-related studies, thanks to the increasing number of projects, including metatranscriptomic data (*Delmont et al., 2022*; *Vorobev et al., 2020*). Furthermore, our work lays the foundations for further developments of clear guidelines and protocols to define the different levels of unknown (*Thomas and Segata, 2019*) and should encourage the scientific community for a collaborative effort to tackle this challenge.

# Materials and methods

## Key resources table

| Reagent type (species) or resource | Designation | Source or reference | Identifiers | Additional information |
|---|---|---|---|---|
| Software, algorithm | Snakemake | Snakemake | RRID: SCR_003475 | Workflow manager |
| Software, algorithm | Prodigal | Prodigal | RRID: SCR_021246 | Gene prediction |
| Software, algorithm | MMseqs2 | MMseqs2 | RRID: SCR_010277 | Sequence clustering and search |
| Software, algorithm | HHMER | HMMER | RRID: SCR_005305 | Sequence-Profile search |
| Software, algorithm | HHblits | HHblits | RRID: SCR_010277 | Profile-Profile search |
| Software, algorithm | PARASAIL | PARASAIL | RRID: SCR_021805 | Sequence alignment |
| Software, algorithm | FAMSA | FAMSA | RRID: SCR_021804 | Sequence alignment |
| Software, algorithm | LEON-BIS | LEON-BIS | RRID: SCR_021803 | Sequence alignment evaluation |
| Software, algorithm | OD-SEQ | OD-SEQ | | Sequence alignment http://www.bioinf.ucd.ie/download/od-seq.tar.gz |

*Continued on next page*

*Continued*

| Reagent type (species) or resource | Designation | Source or reference | Identifiers | Additional information |
|---|---|---|---|---|
| Software, algorithm | SEQKIT | SEQKIT | RRID: SCR_018926 | Fasta file manipulation |
| Software, algorithm | R | R | RRID: SCR_002394 | |
| Software, algorithm | HH-SUITE | HH-SUITE | RRID: SCR_016133 | |
| Software, algorithm | RAXML | RAXML | RRID: SCR_006086 | Phylogeny |
| Software, algorithm | PPLACER | PPLACER | RRID: SCR_004737 | Phylogeny |
| Software, algorithm | PAPARA | PAPARA | | Sequence alignment https://cme.hits.org/exelixis/resource/download/software/papara_nt-2.5-static_x86_64.tar.gz |
| Software, algorithm | Anvi'o | Anvi'o | RRID:SCR_021802 | Omics analysis and visualization https://merenlab.org/software/anvio |
| Software, algorithm | BWA mapper | BWA mapper | RRID: SCR_010910 | Sequence alignment |
| Software, algorithm | BEDTOOLS | BEDTOOLS | RRID: SCR_006646 | |
| Software, algorithm | PhageBoost | PhageBoost | | https://github.com/ku-cbd/PhageBoost |
| Software, algorithm | EGGNOG-mapper | EGGNOG-mapper | RRID: SCR_021165 | |

## Genomic and metagenomic dataset

We used a set of 583 marine metagenomes from four of the major metagenomic surveys of the ocean microbiome: Tara Oceans expedition (TARA) (*Sunagawa et al., 2015*), Malaspina expedition (*Duarte, 2015*), Ocean Sampling Day (OSD) (*Kopf et al., 2015*), and Global Ocean Sampling Expedition (GOS) (*Rusch et al., 2007*). We complemented this set with 1246 metagenomes obtained from the Human Microbiome Project (HMP) phase I and II (*Lloyd-Price et al., 2017*). We used the assemblies provided by TARA, Malaspina, OSD and HMP projects and the long Sanger reads from GOS (*Sanger et al., 1977*). A total of 156 M (156,422,969) contigs and 12.8 M long-reads were collected (*Appendix 1—table 5*).

For the genomic dataset, we used the 28,941 prokaryotic genomes (27,372 bacterial and 1569 archaeal) from the Genome Taxonomy Database (*Parks et al., 2018*) (GTDB) Release 03-RS86 (19th August 2018).

## Computational workflow development

We implemented a computation workflow based on Snakemake (*Köster, 2018*) for the easy processing of large datasets in a reproducible manner. The workflow provides three different strategies to analyze the data. The module *DB-creation* creates the gene cluster database, validates and partitions the gene clusters (GCs) in the main functional categories. The module *DB-update* allows the integration of new sequences (either at the contig or predicted gene level) in the existing gene cluster database. In addition, the workflow has a *profile-search* function to quickly screen samples using the gene cluster PSSM profiles in the database.

## Metagenomic and genomic gene prediction

We used Prodigal (v2.6.3) (*Hyatt et al., 2010*) in metagenomic mode to predict the genes from the metagenomic dataset. For the genomic dataset, we used the gene predictions provided by Annotree (*Mendler et al., 2019*), since they were obtained, consistently, with Prodigal v2.6.3. We identified potential spurious genes using the *AntiFam* database (*Eberhardt et al., 2012*). Furthermore, we screened for 'shadow' genes using the procedure described in *Yooseph et al., 2008*.

## PFAM annotation

We annotated the predicted genes using the *hmmsearch* program from the *HMMER* package (version: 3.1b2) (*Finn et al., 2011*) in combination with the Pfam database v31 (*Finn et al., 2016*). We kept the matches exceeding the internal gathering threshold and presenting an independent e-value <1e-5 and coverage >0.4. In addition, we considered multi-domain annotations, and we removed overlapping annotations when the overlap is larger than 50%, keeping the ones with the smaller e-value.

## Determination of the gene clusters

We clustered the metagenomic predicted genes using the cascaded-clustering workflow of the MMseqs2 software (*Steinegger and Söding, 2018*) ("*--cov-mode 0 c 0.8 --min-seq-id 0.3*"). We discarded from downstream analyses the singletons and clusters with a size below a threshold identified after applying a modification of the broken-stick model (*Macarthur, 1957*). We randomly split the number of gene clusters into p subsets, where p is defined by the proportion of outlier genes per gene cluster. The subsets are then sorted by decreasing size. We iterated over all subsets averaging the results over all iterations. The broken stick model generates the outlier gene proportions, which would occur by chance alone, that is, the distribution of outlier gene proportions if there were no structure in the data.

We integrated the genomic data into the metagenomic cluster database using the "DB-update" module of the workflow. This module uses the *clusterupdate* module of MMseqs2 (*Steinegger and Söding, 2017*), with the same parameters used for the metagenomic clustering.

## Quality-screening of gene clusters

We examined the GCs to ensure their high intra-cluster homogeneity. We applied two methodologies to validate their cluster sequence composition and functional annotation homogeneity. We identified non-homologous sequences inside each cluster combining the identification of a new cluster representative sequence via a sequence similarity network (SSN) analysis, and the investigation of intra-cluster multiple sequence alignments (MSAs), given the new representative. Initially, we generated an SSN for each cluster, using the semi-global alignment methods implemented in *PARASAIL* (*Daily, 2016*) (version 2.1.5). We trimmed the SSN using a filtering algorithm (*Chafee et al., 2018*; *Žure et al., 2017*) that removes edges while maintaining the network structural integrity and obtaining the smallest connected graph formed by a single component. Finally, the new cluster representative was identified as the most central node of the trimmed SSN by the eigenvector centrality algorithm, as implemented in igraph (*Csardi and Nepusz, 2006*). After this step, we built a multiple sequence alignment for each cluster using *FAMSA* (*Deorowicz et al., 2016*) (version 1.1). Then, we screened each cluster-MSA for non-homologous sequences to the new cluster representative. Owing to computational limitations, we used two different approaches to evaluate the cluster-MSAs. We used *LEON-BIS* (*Vanhoutreve et al., 2016*) for the clusters with a size ranging from 10 to 1000 genes and OD-SEQ (*Jehl et al., 2015*) for the clusters with more than 1000 genes. In the end, we applied a broken-stick model (*Macarthur, 1957*) to determine the threshold to discard a cluster.

The predicted genes can have multi-domain annotations in different orders, therefore to validate the consistency of intra-cluster Pfam annotations, we applied a combination of w-shingling (*Broder, 1997*) and Jaccard similarity. We used w-shingling (k-shingle = 2) to group consecutive domain annotations as a single object. We measured the homogeneity of the *shingle sets* (sets of domains) between genes using the Jaccard similarity and reported the median similarity value for each cluster. Moreover, we took into consideration the Clan membership of the Pfam domains and that a gene might contain N-, C-, and M-terminal domains for the functional homogeneity validation. We discarded clusters with a median similarity <1.

After the validation, we refined the gene cluster database removing the clusters identified to be discarded and the clusters containing ≥30% *shadow genes*. Lastly, we removed the single shadow, spurious and non-homologous genes from the remaining clusters (Appendix 3).

## Remote homology classification of gene clusters

To partition the validated GCs into the four main categories, we processed the set of GCs containing Pfam annotated genes and the set of not annotated GCs separately. For the annotated GCs, we inferred a consensus protein domain architecture (DA) (an ordered combination of protein domains) for each annotated gene cluster. To identify each gene cluster consensus DA, we created directed acyclic graphs connecting the Pfam domains based on their topological order on the genes using *igraph* (*Csardi and Nepusz, 2006*). We collapsed the repetitions of the same domain. Then we used the gene completeness as a positive-weighting value for the selection of the cluster consensus DA. Within this step, we divided the GCs into 'Knowns' (Known) if annotated to at least one Pfam domains of known function (DKFs) and 'Genomic unknowns' (GU) if annotated entirely to Pfam domains of unknown function (DUFs).

We aligned the sequences of the non-annotated GCs with FAMSA (*Deorowicz et al., 2016*) and obtained cluster consensus sequences with the *hhconsensus* program from *HH-SUITE* (*Steinegger et al., 2019a*). We used the cluster consensus sequences to perform a nested search against the UniRef90 database (release 2017_11) (*The UniProt Consortium, 2017*) and NCBI *nr* database (release 2017_12) (*NCBI Resource Coordinators, 2018*) to retrieve non-Pfam annotations with *MMSeqs2* (*Steinegger and Söding, 2017*) (*"-e 1e-05 --cov-mode 2 -c 0.6"*). We kept the hits within 60% of the Log(best-e-value) and searched the annotations for any of the terms commonly used to define proteins of unknown function (*Supplementary file 1G*). We used a quorum majority voting approach to decide if a gene cluster would be classified as *Genomic Unknown* or *Known without Pfams* based on the annotations retrieved. We searched the consensus sequences without any homologs in the UniRef90 database against NCBI *nr*. We applied the same approach and criteria described for the first search. Ultimately, we classified as *Environmental Unknown* those GCs whose consensus sequences did not align with any of the NCBI *nr* entries.

In addition, we developed some conservative measures to control the trade-off between specificity and sensitivity for the remote homology searches such as (1) a modification of the algorithm described in *Hingamp et al., 2013* to get a confident group of homologs to determine if a query protein is known or unknown by a quorum majority voting approach (Appendix 4); (2) strict parameters in terms of iterations, bidirectional coverage and probability thresholds for the HHblits alignments to minimize the inclusion of non-homologous sequences; and (3) avoid providing annotations for our gene clusters, as we believe that annotation should be a careful process done on a smaller scale and with experimental context.

## Gene cluster remote homology refinement

We refined the *Environmental Unknown* GCs to ensure the lack of any characterization by searching for remote homologies in the Uniclust database (release 30_2017_10) using the HMM/HMM alignment method *HHblits* (*Remmert et al., 2011*). We created the HMM profiles with the *hhmake* program from the *HH-SUITE* (*Steinegger et al., 2019a*). We only accepted those hits with an *HHblits-probability* ≥90% and we re-classified them following the same majority vote approach as previously described. The clusters with no hits remained as the refined set of EUs. We applied a similar refinement approach to the KWP clusters to identify GCs with remote homologies to Pfam protein domains. The KWP HMM profiles were searched against the Pfam *HH-SUITE* database (version 31), using *HHblits*. We accepted hits with a probability ≥90% and a target coverage >60% and removed overlapping domains as described earlier. We moved the KWP with remote homologies to known Pfams to the Known set, and those showing remote homologies to Pfam DUFs to the GUs. The clusters with no hits remained as the refined set of KWP.

## Gene cluster characterization

We used the *MMseqs2 taxonomy* module (commit: b43de8b7559a3b45c8e5e9e02cb3023dd339231a) in combination with the UniProtKB (release of January 2018) (*The UniProt Consortium, 2018*) to retrieve the taxonomic ids of all genes in a gene cluster. The *taxonomy* module implements the 2bLCA (*Hingamp et al., 2013*) to compute the lowest common ancestor of query sequence. We used the following parameters *"-e 1e-05 –cov-mode 0 c 0.6"* for the search. To retrieve the taxonomic lineages, we used the R package *CHNOSZ* (*Dick, 2008*).

We used eggNOG-mapper (*Huerta-Cepas et al., 2017*) and the EggNog5 database (*Huerta-Cepas et al., 2019*) to provide functional annotations for each gene in a gene cluster. We refined the functional annotations by selecting the orthologous group within the lowest taxonomic level predicted by EggNog-mapper.

We measured the intra-cluster taxonomic and functional admixture by applying the *entropy.empirical*() function from the *entropy* R package (*Hausser and Strimmer, 2008*). This function estimates the Shannon entropy based on the different taxonomic and functional annotation frequencies. For each cluster, we also retrieved the cluster consensus taxonomic and functional annotation using a quorum majority voting approach.

In addition to the taxonomic and functional annotations, we evaluated the clusters' level of darkness and disorder using the Dark Proteome Database (DPD) (*Perdigão et al., 2017*) as reference. We searched the cluster genes against the DPD, applying the MMseqs2 search program (*Steinegger and*

*Söding, 2017*) with "*-e 1e-20 --cov-mode 0 -c 0.6*". For each cluster, we then retrieved the mean and the median level of darkness, based on the gene DPD annotations.

## High-quality clusters

We defined a subset of high-quality clusters based on the completeness of the cluster genes and their representatives. We identified the minimum required percentage of complete genes per cluster by a broken-stick model (*Macarthur, 1957*) applied to the percentage distribution. Then, we selected the GCs found above the threshold and with a complete representative.

## A set of non-redundant domain architectures

We estimated the number of potential domain architectures present in the *Known* GCs considering the large proportion of fragmented genes in the metagenomic dataset and that could inflate the number of potential domain architectures. To identify fragments of larger domain architecture, we considered their topological order in the genes. To reduce the number of comparisons, we calculated the pairwise string cosine distance (q-gram = 3) between domain architectures and discarded the pairs that were too divergent (cosine distance ≥0.9). We collapsed a fragmented domain architecture to the larger one when it contained less than 75% of complete genes.

## Inference of gene cluster communities

We aggregated distant homologous GCs into GCCs. The community inference approach combined an all-vs-all HMM gene cluster comparison with Markov Cluster Algorithm (MCL) (*van Dongen and Abreu-Goodger, 2012*) community identification. We started performing the inference on the Known GCs to use the Pfam DAs as constraints. We aligned the gene cluster HMMs using HHblits (*Remmert et al., 2011*) (-n 2 -Z 10000000 -B 10000000 -e 1) and we built a homology graph using the cluster pairs with probability ≥50% and bidirectional coverage >60%. We used the ratio between HHblits-bitscore and aligned-columns as the edge weights (Appendix 7). We used MCL (*van Dongen and Abreu-Goodger, 2012*) (v. 12–068) to identify the communities present in the graph. We developed an iterative method to determine the optimal MCL inflation parameter that tries to maximize the relationship of five intra-/inter-community properties: (1) the proportion of MCL communities with one single DA, based on the consensus DAs of the cluster members; (2) the ratio of MCL communities with more than one cluster; (3) the proportion of MCL communities with a PFAM clan entropy equal to 0; (4) the intra-community HHblits-score/Aligned-columns score (normalized by the maximum value); and (5) the number of MCL communities, which should, in the end, reflect the number of non-redundant DAs. We iterated through values ranging from 1.2 to 3.0, with incremental steps of 0.1. During the inference process, some of the GCs became orphans in the graph. We applied a three-step approach to assigning a community membership to these GCs. First, we used less stringent conditions (probability ≥50% and coverage ≥ 40%) to find homologs in the already existing GCCs. Then, we ran a second iteration to find secondary relationships between the newly assigned GCs and the missing ones. Lastly, we created new communities with the remaining GCs. We repeated the whole process with the other categories (KWP, GU and EU), applying the optimal inflation value found for the Known (2.2 for metagenomic and 2.5 for genomic data).

## Validation of gene cluster communities

We tested the biological significance of the GCCs using the phylogeny of proteorhodopsin (*Boeuf et al., 2015*) (PR). We used the proteorhodopsin HMM profiles (*Olson et al., 2018*) to screen the marine metagenomic datasets using *hmmsearch* (version 3.1b2) (*Finn et al., 2011*). We kept the hits with a coverage >0.4 and e-value ≤ 1e-5. We removed identical duplicates from the sequences assigned to PR with CD-HIT (*Li and Godzik, 2006*) (v4.6) and cleaned from sequences with less than 100 amino acids. To place the identified PR sequences into the MicRhode (*Boeuf et al., 2015*) PR tree first, we optimized the initial tree parameters and branch lengths with RAxML (v8.2.12) (*Stamatakis, 2014*). We used PaPaRA (v2.5) (*Berger and Stamatakis, 2012*) to incrementally align the query PR sequences against the MicRhode PR reference alignment and *pplacer* (*Matsen et al., 2010*) (v1.1.alpha19-0-g807f6f3) to place the sequences into the tree. Finally, we assigned the query PR sequences to the MicRhode PR Superclusters based on the phylogenetic placement. We further investigated the GCs annotated as viral (196 genes, 14 GC) comparing

them to the six newly discovered viral PRs (*Needham et al., 2019*) using Parasail (*Daily, 2016*) (-a sg_stats_scan_sse2_128_16 t 8 c 1 x). As an additional evaluation, we investigated the distributions of standard GCCs and HQ GCCs within ribosomal protein families. We obtained the ribosomal proteins used for the analysis combining the set of 16 ribosomal proteins from *Méheust et al., 2019* and those contained in the collection of bacterial single-copy genes of anvi'o (*Eren et al., 2021*). Also, for the ribosomal proteins, we compared the outcome of our method to the one proposed by *Méheust et al., 2019* (Appendix 7).

## Metagenomic sample selection for downstream analyses

For the subsequent ecological analyses, we selected those metagenomes with a number of genes larger or equal to the first quartile of the distribution of all the metagenomic gene counts. (*Supplementary file 1F*).

## Gene cluster abundance profiles in genomes and metagenomes

We estimated abundance profiles for the metagenomic cluster categories using the read coverage to each predicted gene as a proxy for abundance. We calculated the coverage by mapping the reads against the assembly contigs using the *bwa-mem* algorithm from *BWA mapper* (*Li and Durbin, 2010*). Then, we used *BEDTOOLS* (*Quinlan and Hall, 2010*), to find the intersection of the gene coordinates to the assemblies, and normalize the per-base coverage by the length of the gene. We calculated the cluster abundance in a sample as the sum of the cluster gene abundances in that sample, and the cluster category abundance in a sample as the sum of the cluster abundances. We obtained the proportions of the different gene cluster categories applying a total-sum-scaling normalization. For the genomic abundance profiles, we used the number of genes in the genomes and normalized by the total gene counts per genome.

## Rate of genomic and metagenomic gene clusters accumulation

We calculated the cumulative number of known and unknown GCs as a function of the number of metagenomes and genomes. For each metagenome count, we generated 1000 random sets, and we calculated the number of GCs and GCCs recovered. For this analysis, we used 1246 HMP metagenomes and 358 marine metagenomes (242 from TARA and 116 from Malaspina). We repeated the same procedure for the genomic dataset. We removed the singletons from the metagenomic dataset with an abundance smaller than the mode abundance of the singletons that got reclassified as good-quality clusters after integrating the GTDB data to minimize the impact of potential spurious singletons. To complement those analyses, we evaluated the coverage of our dataset by searching seven different state-of-the-art databases against our set of metagenomic GC HMM profiles (Appendix 9).

## Occurrence of gene clusters in the environment

We used 1264 metagenomes from the TARA Oceans, MALASPINA Expedition, OSD2014 and HMP-I/II to explore the properties of the unknown sequence space in the environment. We applied the Levins Niche Breadth (NB) index (*Levins, 1966*) to investigate the GCs and GCCs environmental distributions. We removed the GCs and cluster communities with a mean relative abundance <1e-5. We followed a divide-and-conquer strategy to avoid the computational burden of generating the null-models to test the significance of the distributions owing to the large number of metagenomes and GCs. First, we grouped similar samples based on the gene cluster content using the Bray-Curtis dissimilarity (*Bray and Curtis, 1957*) in combination with the *Dynamic Tree Cut* (*Langfelder et al., 2008*) R package. We created 100 random datasets picking up one random sample from each group. For each of the 100 random datasets, we created 100 random abundance matrices using the *nullmodel* function of the *quasiswap* count method (*Miklós and Podani, 2004*). Then we calculated the *observed* NB and obtained the 2.5% and 97.5% quantiles based on the randomized sets. We compared the observed and quantile values for each gene cluster and defined it to have a *Narrow distribution* when the *observed* was smaller than the 2.5% quantile and to have a *Broad distribution* when it was larger than the 97.5% quantile. Otherwise, we classified the cluster as *Non-significant* (*Salazar et al., 2015*). We used a majority voting approach to get a consensus distribution classification based on the ten random datasets.

## Identification of prophages in genomic sequences

We used PhageBoost (*Sirén et al., 2021*) to find gene regions in the microbial genomes that result in high viral signals against the overall genome signal. We set the following thresholds to consider a region prophage: minimum of 10 genes, maximum 5 gaps, single-gene probability threshold 0.9. We further smoothed the predictions using Parzen rolling windows of 20 periods and looked at the smoothed probability distribution across the genome. We disregarded regions that had a summed smoothed probability less than 0.5, and those regions that did differ from the overall population of the genes in a genome by using Kruskal–Wallis rank test (p-value 0.001).

## Lineage-specific gene clusters

We used the F1-score developed for AnnoTree (*Mendler et al., 2019*) to identify the lineage-specific GCs and to which rank they are specific. Following similar criteria to the ones used in *Mendler et al., 2019*, we considered a gene cluster to be lineage-specific if it is present in less than half of all genomes and at least 2 with F1-score > 0.95.

## Phylogenetic conservation of gene clusters

We calculated the phylogenetic conservation ($\tau$D) of each gene cluster using the *consenTRAIT* (*Martiny et al., 2013*) function implemented in the R package *castor* (*Martiny et al., 2013*). We used a paired Wilcoxon rank-sum test to compare the average $\tau$D values for lineage-specific and non-specific GCs.

## Evaluation of the OM-RGC V2 uncharacterized fraction

We integrated the 46,775,154 genes from the second version of the TARA Ocean Microbial Reference Gene Catalog (OM-RGC v2) (*Salazar et al., 2019*) into our cluster database using the same procedure as for the genomic data. We evaluated the uncharacterized fraction and the genes classified into the eggNOG (*Huerta-Cepas et al., 2019*) category S within the context of our database.

## Augmenting RB-TnSeq experimental Data

We searched the 129,477 bacterial genes associated with mutant phenotypes from *Price et al., 2018* against our gene cluster profiles. We kept the hits with e-value ≤1e-20 and a query coverage >60%. Then we filtered the results to keep the hits within 90% of the Log(best-e-value), and we used a majority vote function to retrieve the consensus category for each hit. Lastly, we selected the best-hits based on the smallest e-value and the largest query and target coverage values. We used the fitness values from the RB-TnSeq experiments from Price et al. to identify genes of unknown function that are important for fitness under certain experimental conditions.

## Acknowledgements

The authors thankfully acknowledge the computer resources at MareNostrum and the technical support provided by Barcelona Supercomputing Center (RES-AECT-2014-2-0085), the BMBF-funded de.NBI Cloud within the German Network for Bioinformatics Infrastructure (de.NBI) (031A537B, 031A533A, 031A538A, 031A533B, 031A535A, 031A537C, 031A534A, 031A532B), the University of Oxford Advanced Research Computing (http://dx.doi.org/10.5281/zenodo.22558) and the MARBITS bioinformatics core at ICM-CSIC. CV was supported by the Max Planck Society. AFG received funding from the European Union's Horizon 2020 research and innovation program Blue Growth: Unlocking the potential of Seas and Oceans under grant agreement no. 634,486 (project acronym INMARE). AM was supported by the Biotechnology and Biological Sciences Research Council [BB/M011755/1, BB/R015228/1] and RDF by the European Molecular Biology Laboratory core funds. EOC was supported by project INTERACTOMA RTI2018-101205-B-I00 from the Spanish Agency of Science MICIU/AEI/FEDER. SGA and PS received additional funding by the project MAGGY (CTM2017-87736-R) from the Spanish Ministry of Economy and Competitiveness. The Malaspina 2010 Expedition was supported by the Spanish Ministry of Economy and Competitiveness (MINECO) through the Consolider-Ingenio program (ref. CSD2008-00077). The authors thank Johannes Söding and Alex Bateman for helpful discussions.

## Additional information

### Funding

| Funder | Grant reference number | Author |
| --- | --- | --- |
| Max Planck Society | | Chiara Vanni |
| Horizon 2020 | INMARE | Antonio Fernàndez-Guerra |
| Biotechnology and Biological Sciences Research Council | | Alex Mitchell |
| European Molecular Biology Laboratory | | Robert D Finn |
| Spanish Agency of Science MICIU/AEI/FEDER | INTERACTOMA RTI2018-101205-B-I00 | Emilio O Casamayor |
| Spanish Ministry of Economy and Competitiveness | MAGGY (CTM2017-87736-R) | Silvia G Acinas Pablo Sánchez |

The funders had no role in study design, data collection and interpretation, or the decision to submit the work for publication.

### Author contributions

Chiara Vanni, Conceptualization, Data curation, Formal analysis, Methodology, Writing – original draft, Writing – review and editing; Matthew S Schechter, Data curation, Formal analysis, Writing – review and editing; Silvia G Acinas, Carlos M Duarte, Pablo Sánchez, Resources, Writing – review and editing; Albert Barberán, Emilio O Casamayor, Robert D Finn, Conceptualization, Writing – review and editing; Pier Luigi Buttigieg, Tom O Delmont, Conceptualization, Data curation, Writing – review and editing; A Murat Eren, Formal analysis, Methodology, Writing – review and editing; Renzo Kottmann, Alex Mitchell, Methodology, Writing – review and editing; Kimmo Siren, Formal analysis, Writing – review and editing; Martin Steinegger, Formal analysis, Software, Writing – review and editing; Frank Oliver Gloeckner, Supervision, Writing – review and editing; Antonio Fernàndez-Guerra, Conceptualization, Formal analysis, Methodology, Supervision, Writing – original draft, Writing – review and editing

### Author ORCIDs

Chiara Vanni http://orcid.org/0000-0002-1124-1147
Matthew S Schechter http://orcid.org/0000-0002-8435-3203
Emilio O Casamayor http://orcid.org/0000-0001-7074-3318
Tom O Delmont http://orcid.org/0000-0001-7053-7848
A Murat Eren http://orcid.org/0000-0001-9013-4827
Pablo Sánchez http://orcid.org/0000-0003-2787-822X
Antonio Fernàndez-Guerra http://orcid.org/0000-0002-8679-490X

### Decision letter and Author response

Decision letter https://doi.org/10.7554/eLife.67667.sa1
Author response https://doi.org/10.7554/eLife.67667.sa2

## Additional files

### Supplementary files

• Supplementary file 1. Supplementary tables. (a) KWP high-quality gene clusters (GCs) distribution in the COG groups. (b) Proportion of genes in each cluster category, and Pfam amino acids coverage per cluster category. (c) List of HMP outlier samples. (d) Number of phylogenetic conserved and lineage-specific gene clusters (GCs) in the GTDB bacterial phylogeny. (e) Clusters in the GU community GU_c_21103. (f) List of filtered samples used for the metagenomic analyses. (g) List of terms commonly used to define proteins of unknown function in public databases. (h) Sequence similarity values between viral genes and Needham et al. viral PRs. (i) Number of phylogenetic conserved and lineage-specific GCs in the GTDB archaeal phylogeny.

- Supplementary file 2. Supplementary tables describing general cluster properties. (a) Overall properties for the GCs of the integrated dataset (MG + GTDB). (b) Statistics for the integrated dataset (MG+GTDB). (c) Taxonomic variation within each gene cluster category. (d) Statistics for the metagenomic dataset. (e) Statistics for the genomic dataset.

- Transparent reporting form

### Data availability

We used public data as described in the Methods section and Appendix 1-table 5. The code used for the analyses in the manuscript is available at https://github.com/functional-dark-side/functional-dark-side.github.io/tree/master/scripts, (copy archived at swh:1:rev:86968509e38902580b-04a25786c5a58ba2777b21). A list with the program versions can be found in https://github.com/functional-dark-side/functional-dark-side.github.io/blob/master/programs_and_versions.txt. The code to create the figures is available at https://github.com/functional-dark-side/vanni_et_al-figures, (copy archived at swh:1:rev:4c8f60e761bcac0dd02f17d2fdbb65dcaf75707a), and the data for the figure can be downloaded from https://doi.org/10.6084/m9.figshare.12738476.v2. A reproducible version of the workflow is available at https://github.com/functional-dark-side/agnostos-wf, (copy archived at swh:1:rev:5f9e23e8ac524a533f81c57e500a60b56191b1f5). The data is publicly available at https://doi.org/10.6084/m9.figshare.12459056.

The following dataset was generated:

| Author(s) | Year | Dataset title | Dataset URL | Database and Identifier |
|---|---|---|---|---|
| Vanni C, Fernandez-Guerra A | 2020 | agnostosDB_dbf02445-20200519 | https://doi.org/10.6084/m9.figshare.12459056 | figshare, 10.6084/m9.figshare.12459056 |

The following previously published datasets were used:

| Author(s) | Year | Dataset title | Dataset URL | Database and Identifier |
|---|---|---|---|---|
| O'Gara F, Jackson S, Orlic S, Steinke M, Busch J, Duarte B, Caçador I, Bobrova O, Marteinsson V, Reynisson E, Loureiro C, Luna G, Quero GM, Löscher CR, Kremp A, DeLorenzo ME, Øvreås L, Tolman J, LaRoche J, Penna A, Frischer M, Davis T, Katherine B, Meyer C, Ramos S, Magalhães C, Jude-Lemeilleur F, Aguirre-Macedo ML, Wang S, Poulton N, Jones S, Collin R, Fuhrman JA, Conan P, Alonso C, Stambler N, Goodwin K, Yakimov MM, Baltar F, Bodrossy L, Kamp JV, Frampton DMF, Ostrowski M, Ruth PV, Malthouse P, Claus S, Deneudt K, Mortelmans J, Pitois S, Wallom D, Salter I, Costa R, Schroeder DC, Kandil MM, Amaral V, Biancalana F, Santana R, Pedrotti ML, Yoshida T, Ogata H, Ingleton T, Munnik K, Rodriguez-Ezpeleta N, Berteaux-Lecellier V, Wecker P, Cancio I, Vaulot D, Bienhold C, Ghazal H, Chaouni B, Essayeh S, Ettamimi S, Zaid EH, Boukhatem N, Bouali A, Chahboune R, Barrijal S, Timinouni M, Otmani F, Bennani M, Mea M, Todorova N, Karamfilov V, Hoopen P, Cochrane G, L'Haridon S, Bizsel KC, Vezzi A, Lauro FM, Martin P, Jensen RM, Hinks J, Gebbels S, Rosselli R, Pascale FD, Schiavon R, Santos A, Villar E, Pesant S, Cataletto B, Malfatti F, Edirisinghe R | 2015 | Ocean Sampling Day | https://github.com/MicroB3-IS/osd-analysis/wiki/Guide-to-OSD-2014-data | OSD, ERS667653 |
| Sunagawa A | 2015 | TARA Oceans | https://www.ebi.ac.uk/ena/browser/view/PRJEB402 | EBI European Nucleotide Archive, PRJEB402 |

*Continued on next page*

*Continued*

| Author(s) | Year | Dataset title | Dataset URL | Database and Identifier |
|---|---|---|---|---|
| Rusch DB, Halpern AL, Sutton G, Heidelberg KB, Williamson S, Yooseph S, Wu D, Eisen JA, Hoffman JM, Remington K, Beeson K, Tran B, Smith H, Baden-Tillson H, Stewart C, Thorpe J, Freeman J, Andrews-Pfannkoch C, Venter JE, Li K, Kravitz S, Heidelberg JF, Utterback T, Rogers Y, Falcón LI, Souza V, Bonilla-Rosso G, Eguiarte LE, Karl DM, Sathyendranath S, Platt T, Bermingham E, Gallardo V, Tamayo-Castillo G, Ferrari MR, Strausberg RL, Nealson K, Friedman R, Frazier M, Venter JC | 2007 | Global Ocean Sampling | https://www.ncbi.nlm.nih.gov/bioproject?cmd=PRJNA13694 | NCBI BioProject, PRJNA13694 |
| Mendler K, Chen HP, Arks DH, Lobb B, Hug LA, Doxey AC | 2019 | Annotree-GTDB_r86 | https://data.ace.uq.edu.au/public/misc_downloads/annotree/r86/ | Annotree-Genome Taxonomy Database, GTDB_r86 |
| Lloyd-Price J, Mahurkar A, Rahnavard G, Crabtree J, Orvis J, Hall AB, Brady A, Creasy HH, McCracken C, Giglio MG, McDonald D, Franzosa EA, Knight R, White O, Huttenhower C | 2017 | HMP (phase I and II) | http://hmpdacc.org | Human Microbiome Project, HMP |

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

## Appendix 1

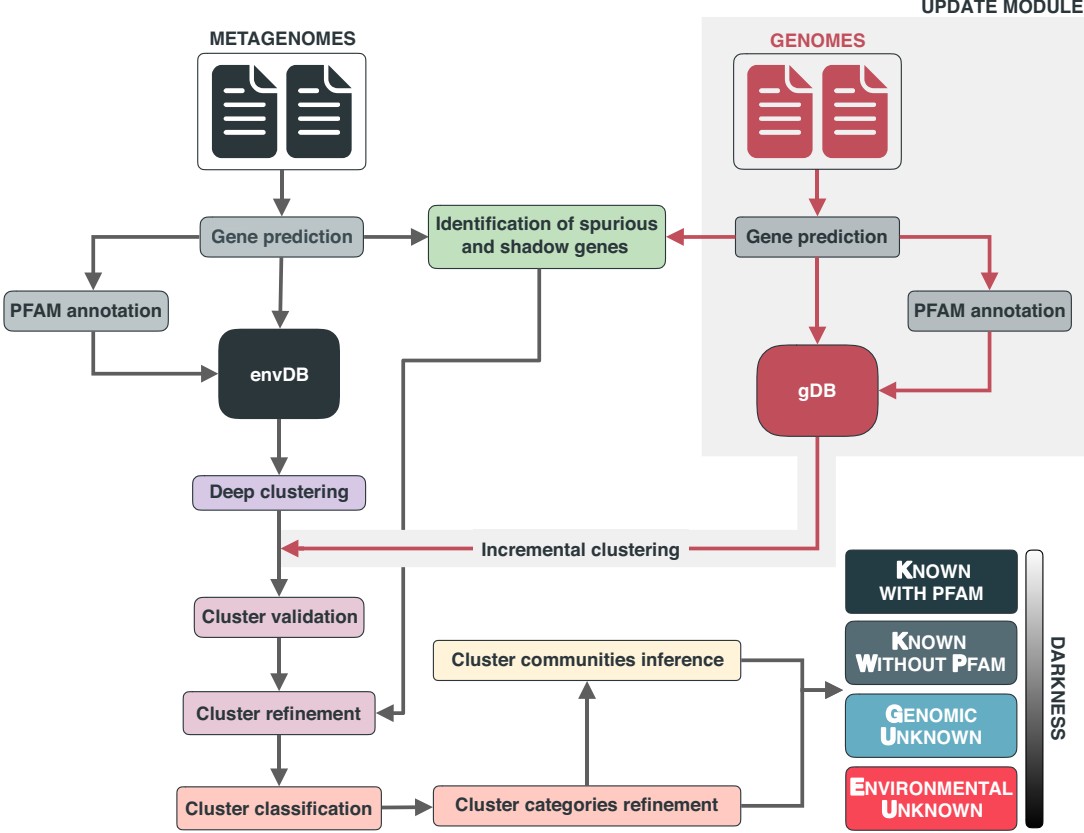

**Appendix 1—figure 1.** Overview of the workflow to partition the genomic and metagenomic sequence space between known and unknown. The workflow performs gene prediction, gene clustering, gene clustering validation and refinement, GCC inference, and partitions the sequence space in the different known and unknown categories.

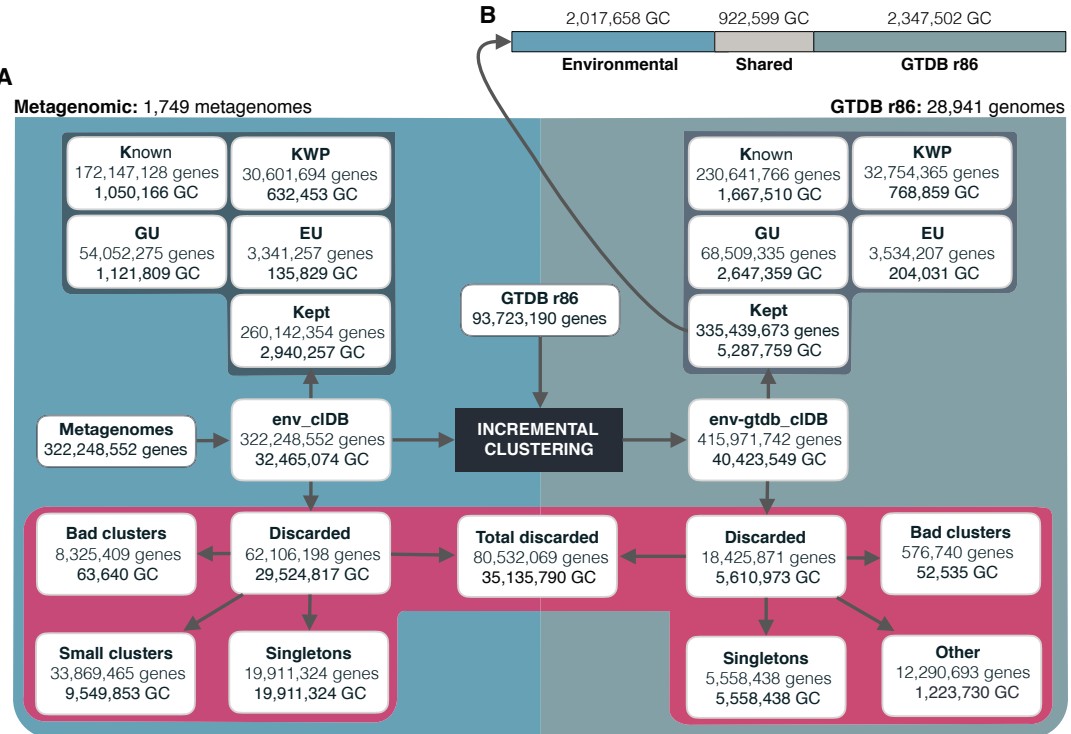

**Appendix 1—figure 2.** The diagram shows a schematic description of the number of genes and GCs that have been kept or discarded. (**A**) We analyzed a dataset of 1749 metagenomes from marine and human environments and 28,941 genomes from the GTDB_r86 summing up to 415,971,742 genes. The composition of the genomic box 'Other' is described in Appendix Note 5. (**B**) GC overlap between the environmental and genomic datasets.

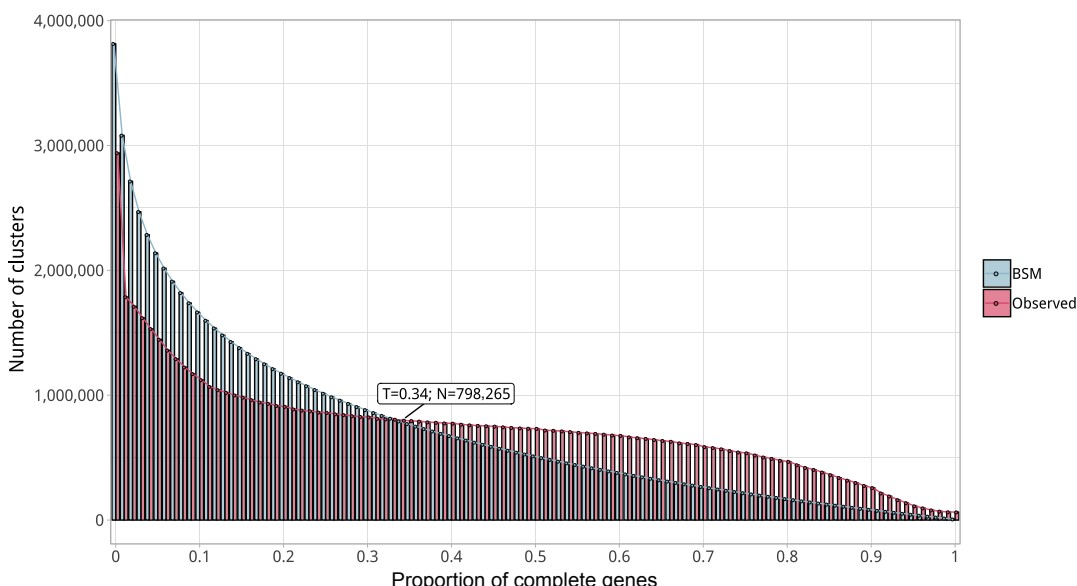

**Appendix 1—figure 3.** Proportion of complete genes per cluster. Distribution of observed values compared with those generated by the Broken-stick model. The cut-off was determined at 34% complete genes per cluster.

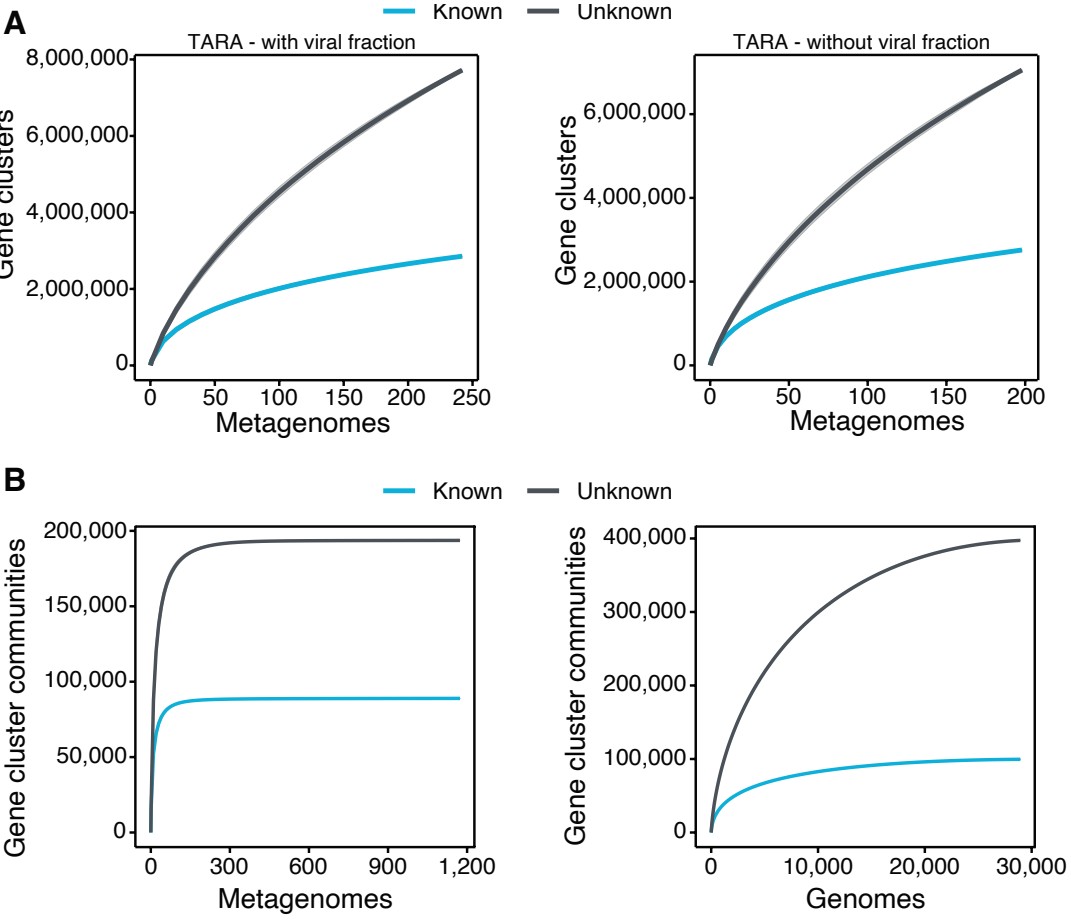

**Appendix 1—figure 4.** Collector curves for the known and unknown sequence space. (**A**) Collector curves at the gene cluster level, for the TARA metagenomes, including the viral fraction (left) and excluding it (right) from the analysis. (**B**) Collector curves at gene cluster community level for the metagenomes from TARA, MALASPINA, and HMP-I/II projects (left) and the 28,941 GTDB genomes (right).

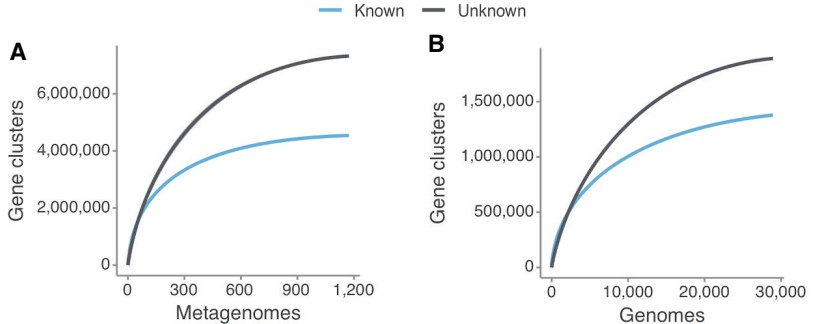

**Appendix 1—figure 5.** Collector curves for the known and unknown sequence space at the gene cluster level for (**A**) the metagenomes from TARA, MALASPINA and HMP-I/II projects, and for (**B**) the 28,941 GTDB genomes. Singletons were excluded from the calculations.

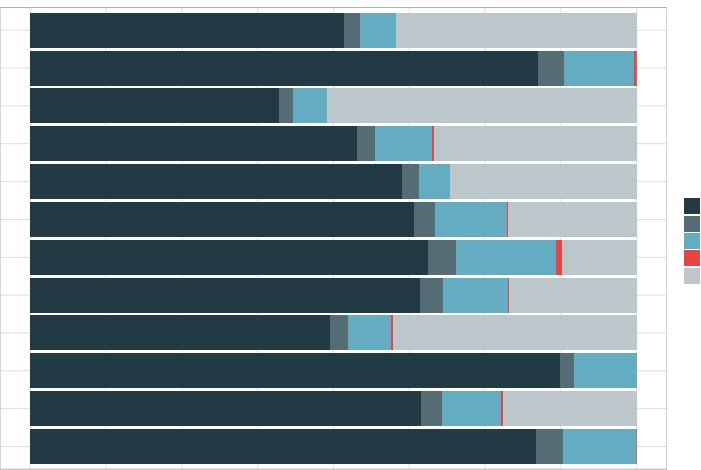

**Appendix 1—figure 6.** Proportion of gene cluster categories per biome. On the y-axis are reported the 11 main biome categories indicated by MGnify and in parenthesis the total number of genes in each biome. The gray fraction represents the pool of genes from MGnify that were not found in our dataset.

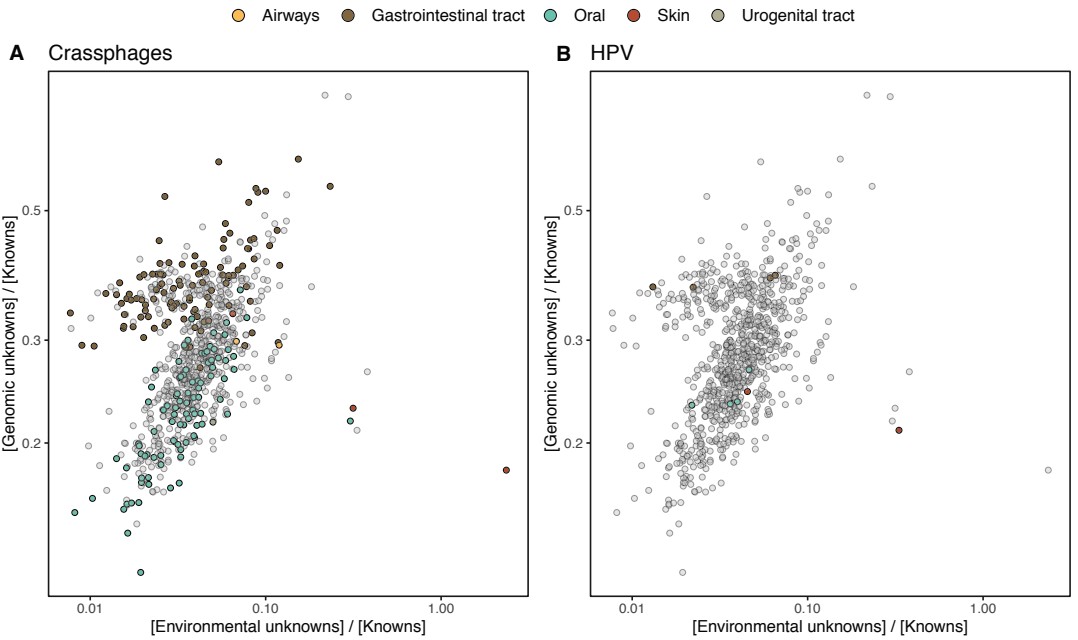

**Appendix 1—figure 7.** HMP outlier samples enriched in (**A**) crAssphages, and (**B**) papillomaviruses (HPV).

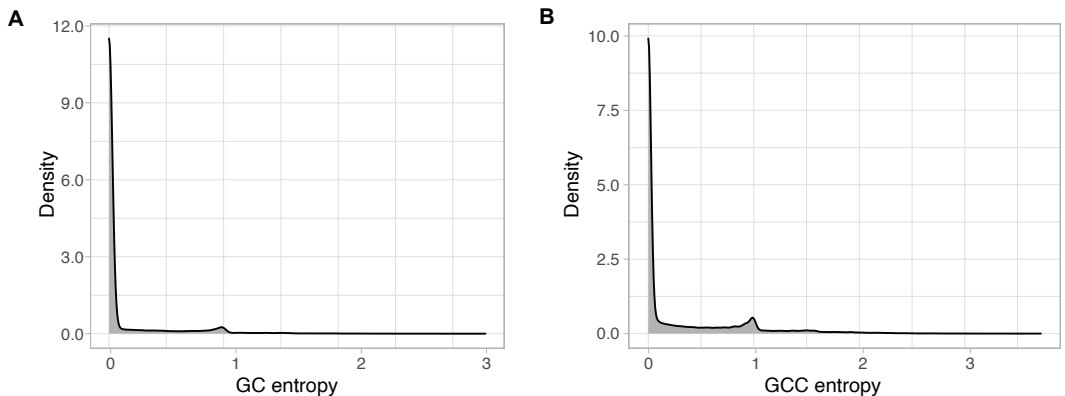

**Appendix 1—figure 8.** EggNOG annotations entropy within the GCs (**A**) and the GCCs (**B**). The entropy was calculated using the function *entropy.empirical*() from the R package 'entropy', which estimates the Shannon entropy values based on the value empirical frequencies.

**Appendix 1—table 1.** Number of metagenomic clusters and genes after the validation and refinement steps.

|  | Good-quality | Bad-quality | Total |
|---|---|---|---|
| Clusters | 2,940,257 | 63,640 | 32,465,074 |
| Genes | 260,142,354 | 8,325,409 | 322,248,552 |

**Appendix 1—table 2.** MG +GTDB high-quality (HQ) subset of gene clusters (GCs).

| Category | HQ GCs | HQ genes | pHQ GCs | pHQ genes |
|---|---|---|---|---|
| K | 76,718 | 40,710,936 | 0.0145 | 0.120 |
| KWP | 16,922 | 1,733,599 | 0.00320 | 0.005132 |
| GU | 95,370 | 9,908,630 | 0.0180 | 0.0293 |
| EU | 14,207 | 477,625 | 0.00269 | 0.00141 |
| Total | 203,217 | 52,830,790 | 0.0384 | 0.1562 |

**Appendix 1—table 3.** Mean proportion of complete genes per cluster in the four functional categories.

|  | K | KWP | GU | EU |
|---|---|---|---|---|
| Mean percentage of complete genes | 0.50 | 0.22 | 0.68 | 0.70 |

**Appendix 1—table 4.** KWP high-quality gene clusters (GCs) distribution in the COG groups. (Full table in *Supplementary file 1A*).

| COG group | Number of GCs | Proportion of GCs |
|---|---|---|
| CELLULAR PROCESSES AND SIGNALING | 2292 | 0.135 |
| INFORMATION STORAGE AND PROCESSING | 1582 | 0.0935 |
| METABOLISM | 1679 | 0.0992 |
| POORLY CHARACTERIZED | 2899 | 0.171 |
| NC | 8470 | 0.501 |

**Appendix 1—table 5.** Environmental (metagenomic) dataset description.

**(A) Number of samples and sites per metagenomic project.**

| Dataset | Reference | Samples | Sites | Contigs |
|---|---|---|---|---|
| TARA | *Sunagawa et al., 2015* | 242 | 141 | 62,404,654 |
| Malaspina | *Duarte, 2015* | 116 | 30 | 9,330,293 |
| OSD | *Kopf et al., 2015* | 145 | 139 | 4,127,095 |
| HMP | *Lloyd-Price et al., 2017* | 1,246 | 18 | 80,560,927 |
| Dataset | Reference | Samples | Sites | Reads |
| GOS | *Rusch et al., 2007* | 80 | 70 | 12,672,518 |

**(B) Number of predicted genes per completeness category.**

| Total | "00" | "10" | "01" | "11" |
|---|---|---|---|---|
| 322,248,552 | 118,717,690 | 106,031,163 | 102,966,482 | 75,694,123 |

Note: "00" = complete, both start and stop codon identified. "01" = right boundary incomplete. "10" = left boundary incomplete. "11" = both left and right edges incomplete.

**Appendix 1—table 6.** Summary of the number of EU clusters based on their presence in MAGs and their environmental distribution, obtained with the Levin's Niche Breadth index.

| | Total clusters | Broad | Narrow | Non-significant |
|---|---|---|---|---|
| Total EU | 204,031 | 471 | 8421 | 195,079 |
| EU in MAGs | 55,520 | 88 | 316 | 55,116 |
| EU not in MAGs | 148,511 (73%) | 383 (81%) | 8105 (96%) | 140,023 (72%) |

**Appendix 1—table 7.** Number of lineage-specific gene clusters of unknown function at different taxonomic levels within the Cand.

*Patescibacteria* phylum.

| Taxonomic level | Number of clusters |
|---|---|
| Phylum | 2 |
| Class | 6 |
| Order | 104 |
| Family | 1456 |
| Genus | 6987 |
| Species | 45,788 |

**Appendix 1—table 8.** Shannon entropy values for the eggNOG annotations within the gene clusters.

| | Min. | 1st qu. | Median | Mean | 3rd qu. | Max. |
|---|---|---|---|---|---|---|
| Entropy per GC | 0.000 | 0.000 | 0.000 | 0.105 | 0.000 | 3.729 |

**Appendix 1—table 9.** Shannon entropy values for the eggNOG annotations within the gene clusters communities.

| | Min. | 1st qu. | Median | Mean | 3rd qu. | Max. |
|---|---|---|---|---|---|---|
| Entropy per GCC | 0.000 | 0.000 | 0.000 | 0.285 | 0.400 | 3.721 |

# Appendix 2

## Metagenomic singletons and small gene clusters

Analysis of metagenomic singletons and gene clusters with less than ten genes.

The singletons represent 60% of the gene clusters (GCs) and 6% of the total genes. The GCs with less than ten genes, here referred to as small GCs for simplicity, represent 29% of the GCs and 10.5% of the gene dataset (*Appendix 1—figure 2A*). Although we discarded these two sets from the main study, we investigated them to obtain a complete analysis of the initial dataset. Both sets were first searched against the Pfam database of protein domain families (*Finn et al., 2016*), and subsequently classified following the steps described in Appendix Note 3. For the small GCs classification, we used the cluster consensus sequence, which we extracted using the *hhconsensus* program of the HH-SUITE (*Steinegger et al., 2019a*), from the GC multiple sequence alignments (MSAs), generated with FAMSA (*Deorowicz et al., 2016*).

We could not find any homologous in the Pfam database for the large majority of both singletons and small GCs, 95%, and 89%, respectively (*Appendix 2—table 1*). After the classification, the large majority of the singletons remained completely uncharacterized, (64% was identified as EU) (*Appendix 2—table 2*). Similarly, the small GCs were also found dominated by GCs of unknowns, with 38% of the clusters classified as EU and 29% as GU (*Appendix 2—table 2*).

**Appendix 2—table 1.** Singletons and small GCs Pfam annotations.

|  | Total | Annotated | Not annotated |
|---|---|---|---|
| Singletons | 19,911,324 | 934,548 | 18,976,776 |
| Small GCs | 9,549,853 | 1,028,076 | 8,521,777 |

**Appendix 2—table 2.** Number of singletons and small GCs per functional category.

|  | K | KWP | GU | EU |
|---|---|---|---|---|
| Singletons | 852,413 | 3,505,161 | 2,763,476 | 12,790,274 |
| Small GCs | 946,112 | 2,213,654 | 2,744,262 | 3,645,825 |

## Appendix 3

### Metagenomic gene cluster validation and refinement

To obtain a set of gene clusters characterized by a high intra-cluster homogeneity, we identified spurious, shadow and outlier genes, and we removed them from the clusters.

### Identification of spurious genes

We identified spurious genes by screening our gene data set against the *AntiFam* database (*Eberhardt et al., 2012*).

### Identification of shadow genes

We identified shadow genes using the procedure described in *Yooseph et al., 2008*. (1) Two genes on the same strand are considered overlapping if their intervals overlap by at least 60 bps; (2) genes that are on the opposite strands are considered overlapping if their intervals overlap by at least 50 bps, and their 3' ends are within each other's intervals, or if their intervals overlap by at least 120 bps and the 5' end of one is in the interval of the other.

### Identification of outlier genes

Outlier genes are sequences inside a cluster non-homologous to the other cluster genes and were identified during the cluster validation step (see Methods - **Gene cluster validation**).

The number of spurious, shadow and outlier genes identified in the data set is reported in *Appendix 3—table 1*.

**Appendix 3—table 1.** Number of spurious, shadow and outlier genes in the metagenomic clusters.

| Gene category | Clusters ≥ 10 genes | Clusters < 10 genes | Singletons |
|---|---|---|---|
| Spurious | 44,205 | 6784 | 2,335 |
| Shadow | 289,258 | 144,571 | 177,126 |
| Outliers | 3,118,850 | - | - |

### Cluster refinement

After the validation, we proceeded with the retrieval of the subset of "good" clusters. Clusters with ≥30% shadow genes were identified as shadow-clusters, as proposed in *Yooseph et al., 2008*. During the cluster validation, we identified a minimum of 10% outlier genes as the threshold to classify a cluster as "bad-quality" (*Appendix 3—figure 1*; *Appendix 3—table 2A*). We combined this threshold with a Jaccard similarity index <1, indicating a low intra-cluster Pfam domain architecture (DA) homogeneity, for the Pfam annotated clusters (*Appendix 3—table 2B*). We performed the cluster refinement in three consecutive steps:

I.   Discard the "bad" clusters ( ≥ 10% outliers & Jaccard similarity index <1)
II.  Discard the "shadow" clusters ( ≥ 30% shadow genes)
III. Remove the single shadow, spurious and outlier genes from the remaining clusters.

**Appendix 3—table 2.** Metagenomic gene cluster validation results.

**(A) Evaluation of cluster sequence composition.**

|  | Pre-Compos. validation | good quality | bad quality |
|---|---|---|---|
| Clusters | 3,003,897 | 2,958,266 | 45,631 |
| Genes | 268,467,763 | 266,268,638 | 2,199,125 |

**(B) Evaluation of cluster Pfam functional annotations.**

|  | Pre-Funct. validation | Funct. good | Funct. bad |
|---|---|---|---|
| Clusters | 1,015,924 | 1,004,166 | 11,758 |
| Genes | 181,433,541 | 178,167,583 | 3,246,002 |

The results for each step are shown in *Appendix 3—table 2* and *Appendix 3—table 3*. From the initial set of ~3 M clusters with more than ten genes, we identified 57,052 GCs as "bad" and 6,261 as "shadow". From the remaining set of 2,940,593 clusters, we removed a total of 2,708,994 shadow, spurious and outlier genes. During this last step, we discarded 336 more clusters: 244 resulted being composed only of spurious and outlier genes (one in the Pfam annotated set of clusters and 243 in the non-annotated set), and 92 clusters were discarded since they were left as singletons after refinement. Besides, we moved 1,190 Pfam annotated clusters to the non-annotated set since they were left without any annotated gene. In summary, we removed 63,640 GCs and a total of 8,325,409 genes, respectively, 2% and 3% of the initial data set. The refined set contains 2,940,592 GCs and 260,142,354 genes (*Appendix 3—table 3*).

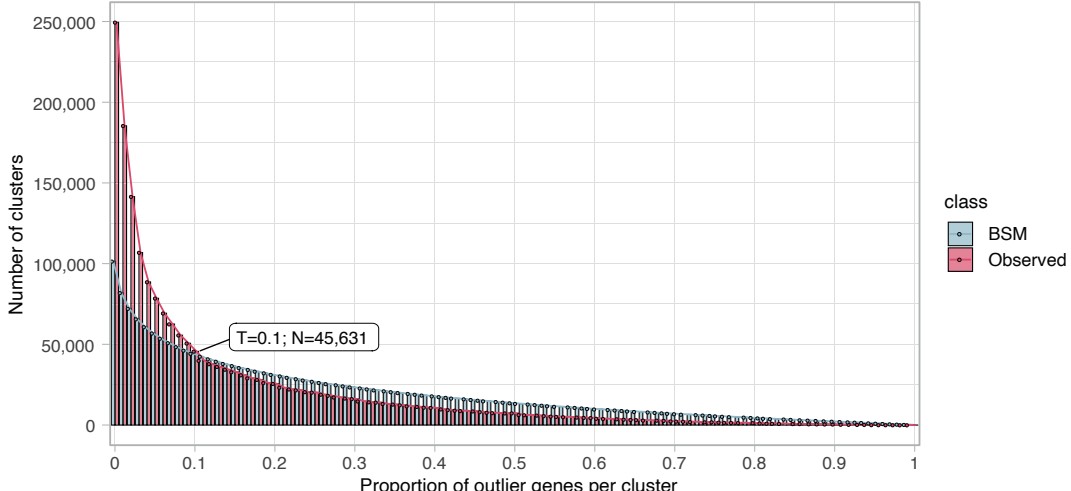

**Appendix 3—figure 1.** Proportion of outlier genes detected within each cluster MSA. Distribution of observed values compared with those generated by the Broken-stick model. The cut-off was determined at 10% outlier genes per cluster.

**Appendix 3—table 3.** Steps: Step I - Removing of the "bad clusters".
Step II - Removing of the "shadow clusters". Step III - Removing single spurious, shadow or outlier genes.

**(A) Number of clusters in each step of the cluster refinement.**

|          | Step I    | Step II   | Step III  | Refined   |
|----------|-----------|-----------|-----------|-----------|
| Clusters | 3,003,897 | 2,946,845 | 2,940,593 | 2,940,257 |
| Removed  | −57,052   | −6,252    | −336      |           |

**(B) Number of genes in each step of the cluster refinement.**

|        | Step I      | Step II     | Step III    | Refined     |
|--------|-------------|-------------|-------------|-------------|
| Genes  | 268,467,763 | 263,022,636 | 262,851,348 | 260,142,354 |
| Removed| −5,445,127  | −171,288    | −2,708,994  |             |

# Appendix 4

## Metagenomic gene cluster classification and remote homology refinement

Classification of the refined subset of gene clusters and remote homology refinement.

## Methods

We searched the gene clusters (GCs) without any Pfam annotated gene against two functional databases, the UniRef90, from UniProt (*The UniProt Consortium, 2017*), and the NCBI *nr* database (*NCBI Resource Coordinators, 2018*). We screened the two databases using the cluster consensus sequences, obtained by applying the *hhconsensus* program of the *HH-SUITE* (*Steinegger et al., 2019a*) on the clusters multiple sequence alignments (MSAs) generated with the *FAMSA* program (*Deorowicz et al., 2016*). We performed two nested searches using the *MMSeqs2* (*Steinegger and Söding, 2017*) program and following a similar workflow as the ''2bLCA'' described in Hinghamp et al. (*Hingamp et al., 2013*). The search-workflow consisted of five steps: First, we searched the consensus sequences against the functional database, with `-e 1e-05 --cov-mode` 2 c 0.6. Second, we extracted the high scoring pairs (HSP) of the best hits and we searched them again using the same parameters. Third, we merged the top hits from the first with the second search results. Fourth, we filtered out the second search hits with a bigger e-value than the first search top hits. And fifth, we selected the hits that were found in 60% of the log10(best-e-value). We first applied this search-workflow to screen the UniRef90 database (release 2017_11) (*The UniProt Consortium, 2017*). We classified the GCs as GU if their consensus sequences were found annotated to proteins labeled with any of the terms commonly used to define proteins of unknown function in public databases (*Supplementary file 1G*). We classified, instead, as KWP, the clusters with consensus annotated to functionally characterized proteins. Secondly, we applied the same search-workflow to search the consensus sequences with no homologs in the UniRef90 database, against the NCBI *nr* database (release 2017_12) (*NCBI Resource Coordinators, 2018*). We used the same criteria to classify a GC as GU or KWP. Ultimately, we classified as EU the GCs whose consensus sequences did not align with any of the NCBI *nr* entries.

We processed the Pfam annotated GCs to retrieve a GC consensus domain architecture (DA). We classified as GU the GCs with a consensus DA composed only of Pfam domain of unknown function (DUFs) and as K the rest. The methods for this step are described in Methods - **Remote homology classification of gene clusters**.

We refined the classified GCs to account for remote homologies. A detailed description of this process can be found in Methods - **Gene cluster remote homology refinement.**

## Results

From the 1,946,737 non-annotated clusters, 1,581,115 were found homologous to UniRef90 entries. Of these hits, more than 50% were found homologous to "hypothetical" proteins and classified as GU, and the other hits were labeled as KWP. The remaining 365,622 clusters, with no homologs to UniRef90, were screened against the NCBI nr database. We found 20,277 clusters in the NCBI nr, of them, 15,998 clusters were homologous to "hypothetical" proteins, and 4,279 clusters to characterized proteins and were classified respectively as GU and KWP. The remaining 345,345 clusters were not found in the NCBI nr database and therefore identified as EU. After the cascaded profile search against UniRef90 and NCBI nr, and the analysis of the GC consensus DAs, we classified the GCs into 912,551 K, 753,718 KWP, 928,643 GU, and 345,345 EU. Detailed results for each search are reported in *Appendix 4—table 1*.

**Appendix 4—table 1.** Metagenomic gene clusters classification steps.

**(A) Results from the search against the UniRef90 database**

| Search vs UniRef90 | Hits | No-hits |
| --- | --- | --- |
| Initial clusters:1,946,737 | 1,581,115 | 365,622 |
| | Characterized | Hypothetical |
| | 749,439 | 831,676 |

**(B) Results from the search against the and the NCBI nr databases**

| Search vs NCBI nr | Hits | No-hits |
|---|---|---|
| Initial clusters: 365,622 | 20,277 | 345,345 |
| | Characterized | Hypothetical |
| | 4,279 | 15,998 |

**(C) Classification of the Pfam annotated GCs based on the consensus DAs.**

| Consensus DA analysis | Annotated to DKF DAs | Annotated to DUF DAs |
|---|---|---|
| Initial clusters: 993,520 | 912,551 | 80,969 |

**Appendix 4—table 2.** Metagenomic GC remote homology refinement steps.

| | K | KWP | GU | EU |
|---|---|---|---|---|
| Initial GCs | 912,551 | 753,718 | 928,643 | 345,345 |
| EU refinement | - | + 38,333 | + 171,183 | −209,516 |
| Post-EU refinement | 912,551 | 792,051 | 1,099,826 | 135,829 |
| KWP refinement | + 137,615 | −159,598 | + 21,983 | - |
| Refined GCs | 1,050,166 | 632,453 | 1,121,809 | 135,829 |

# Appendix 5

## GTDB genomes integration

*Results from the integration of the Genome Taxonomy Database (**Parks et al., 2018**) into the metagenomic dataset.*

We integrated the metagenomic GCs with the 93,723,190 genes from the archaeal and bacterial GTDB genomes (release 86) (**Parks et al., 2018**).

The integration strategy provides comparable results to a single-step clustering strategy (**Appendix 5—figure 2** and **Appendix 5—table 9**), reducing the computational time and resources needed. Additionally, this approach is scalable (**Vanni et al., 2021**), which is crucial considering the ever-increasing amount of sequence data generated.

A total of 67,446,376 genomic genes, 72% of the whole dataset, were found in the metagenomic GCs. The remaining 26,276,814 (28% of the initial dataset) genes were then clustered separately into 7,958,475 genomic GCs (**Appendix 5—table 1**). This set of GCs was processed through our workflow steps to be validated, classified and refined.

**Appendix 5—table 1.** GTDB integration in the metagenomic dataset.

|  | Metagenomic | Shared | Genomic | Total |
|---|---|---|---|---|
| GCs | 30,301,693 | 2,163,381 | 7,958,475 | 40,423,549 |
| Genes | 199,693,614 | 190,001,314 | 26,276,814 | 415,971,742 |

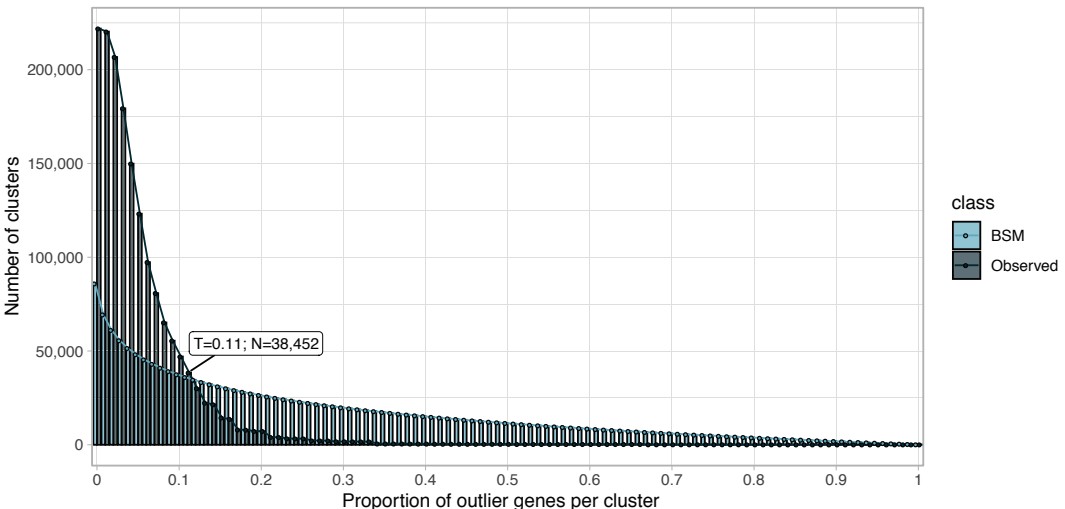

**Appendix 5—figure 1.** Proportion of outlier genomic genes identified within each cluster MSA. Distribution of observed values compared with those of the Broken-stick model.

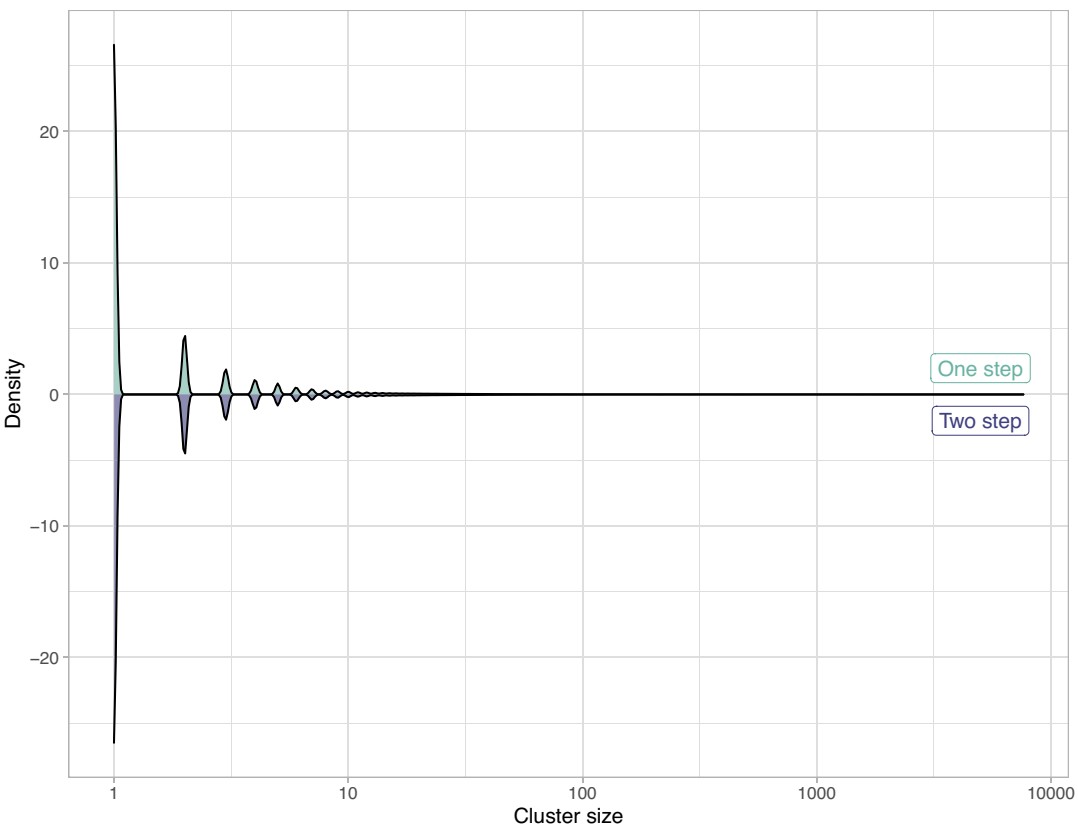

**Appendix 5—figure 2.** Comparison of the clustering results obtained with the one-step and two-step approach in terms of cluster composition.

Within the set of genomic GCs, we identified 5,558,438 singletons and 2,400,037 GCs with more than one gene. We were able to annotate to Pfam protein domain families 41% of the genomic genes. The annotation led to 556,834 annotated GCs and 1,843,203 non-annotated GCs. The validation step determined the minimum proportion of outlier genes per cluster at 11% (*Appendix 5—figure 1*). The majority of the genomic GCs showed high intra-cluster homogeneity, both in terms of sequence composition and functional annotations (*Appendix 5—table 2*).

**Appendix 5—table 2.** Genomic GC validation results.

**(A) Evaluation of cluster sequence composition.**

|       | Pre-Compos. validation | good quality | bad quality |
|-------|------------------------|--------------|-------------|
| GCs   | 2,400,037              | 2,361,585    | 38,452      |
| Genes | 20,718,376             | 20,364,454   | 353,922     |

**(B) Evaluation of Pfam functional annotations.**

|       | Pre-Funct. validation | good quality | bad quality |
|-------|-----------------------|--------------|-------------|
| GCs   | 556,834               | 542,410      | 14,424      |
| Genes | 10,091,203            | 9,865,550    | 225,653     |

**(C) Combined cluster validation results.**

|       | Pre-validation | good quality | bad quality |
|-------|----------------|--------------|-------------|
| GCs   | 2,400,037      | 2,347,502    | 52,535      |
| Genes | 20,718,376     | 20,141,636   | 576,740     |

After the validation, we refined the GCs removing the GCs identified as "bad" and the detected outliers' genes (see *Appendix 5—table 3*). We classified the refined subset of 2,347,502 GCs into the four functional categories via the same protocol applied for the metagenomic data set. The results of the GC classification are reported in *Appendix 5—table 4*. After the classification steps, we refined the EU and KWP GCs searching their HMMs profiles for remote homologies in the Uniclust (release 30_2017_10) (*Mirdita et al., 2017*) and the Pfam (v. 31.0) (*Finn et al., 2016*) databases, respectively, using HHblits (*Remmert et al., 2011*). An overview of the results step-by-step can be found in *Appendix 5—table 5A*. In the end, we obtained 617,344 GCs classified as Known, 136,406 as KWP, 1,525,550 as GU and 68,202 as EU (*Appendix 5—table 5B*). The genomic dataset appeared highly dominated by the GU, which accounts for 65% of the GCs. In the end, we retrieved a subset of genomic "High Quality" (mostly complete) GCs (*Appendix 5—table 6*). The numbers of genes and GCs for the integrated (MG + GTDB) dataset are reported in *Appendix 5—table 7*.

**Appendix 5—table 3.** Spurious, shadow, and outlier genes in the genomic GCs.

| Gene category | GCs ≥ 2 genes | Singletons |
|---|---|---|
| Spurious | 3,252 | 1,312 |
| Shadow | 223,535 | 125,262 |
| Outliers | 449,080 | - |

**Appendix 5—table 4.** Non-annotated genomic GC classification.

**(A) Results from the search against the UniRef90 database.**

| Search vs UniRef90 | Hits | | No-hits |
|---|---|---|---|
| Initial GCs: 1,816,999 | 1,570,094 | | 246,905 |
| | Characterized | Hypothetical | |
| | 304,004 | 1,266,090 | |

**(B) Results from the search against the NCBI nr database.**

| Search vs NCBI nr | Hits | | No-hits |
|---|---|---|---|
| Initial GCs: 246,905 | 28,704 | | 218,201 |
| | Characterized | Hypothetical | |
| | 1,280 | 27,424 | |

**(C) Classification of the Pfam annotated GCs based on the consensus DAs.**

| Consensus DA analysis | DKF DAs | DUF DAs |
|---|---|---|
| Initial GCs: 993,520 | 912,551 | 65,688 |

**Appendix 5—table 5.** Genomic GC remote homology refinement and final genomic GC dataset.

**(A) Remote-homology refinement steps.**

| | K | KWP | GU | EU |
|---|---|---|---|---|
| Initial GCs | 464,815 | 305,284 | 1,359,202 | 218,201 |
| EU refinement | - | + 5,704 | + 144,295 | –149,999 |
| Post-EU refinement | 464,815 | 310,988 | 1,503,497 | 68,202 |
| KWP refinement | + 152,529 | –174,582 | + 22,053 | - |
| Refined GCs | 617,344 | 136,406 | 1,525,550 | 68,202 |

**(B) Genomic GC refined dataset.**

|       | K         | KWP     | GU        | EU      | Total      |
|-------|-----------|---------|-----------|---------|------------|
| Genes | 9,997,529 | 663,107 | 9,305,621 | 175,379 | 20,141,636 |
| GCs   | 617,344   | 136,406 | 1,525,550 | 68,202  | 2,347,502  |

**Appendix 5—table 6.** Genomic high quality (HQ) GCs.

| Category | HQ GCs | HQ genes   | pHQ GCs | pHQ genes |
|----------|--------|------------|---------|-----------|
| K        | 12,202 | 25,105,156 | 0.0198  | 0.0096    |
| KWP      | 4,019  | 1,349,165  | 0.0295  | 0.0214    |
| GU       | 12,699 | 8,403,393  | 0.0083  | 0.0062    |
| EU       | 438    | 471,820    | 0.0064  | 0.0074    |

**Appendix 5—table 7.** MG +GTDB seed database.
Integrated number of genes and GCs per category.

|       | K          | KWP        | GU         | EU        | Total       |
|-------|------------|------------|------------|-----------|-------------|
| Genes | 230,641,76 | 32,754,365 | 68,509,335 | 3,534,207 | 335,439,673 |
| GCs   | 1,667,510  | 768,859    | 2,647,359  | 204,031   | 5,287,759   |

## Summary of the post-genomic integration dataset

In-detail description of the integrated metagenomic-genomic dataset.

The integration of 93,723,190 genomic genes into the metagenomic dataset (322,248,552 genes, 32,465,074 GCs) resulted into a dataset of 415,971,742 genes and 40,423,549 GCs (*Appendix 1—figure 2A* and *Appendix 5—table 1*). As shown in *Appendix 5—figure 2A*, the integrated dataset is divided into: (1) "kept" GCs and (2) "discarded" GCs.

### 1. The "kept" GCs.

The "kept" GC dataset contains the 2,940,257 metagenomic "kept" GCs with 260,142,354 genes (*Appendix 1—figure 2A*), the genomic "kept" 2,347,502 GCs with 20,141,636 genes (*Appendix 5—table 5B*), plus 55,155,683 genomic genes found in the metagenomic set of "kept" GCs (*Appendix 5—table 8*), for a total of 5,287,759 GCs and 335,439,673 genes. A description of the integrated "kept" dataset numbers of GCs and genes, and their distribution in the different categories can be found in *Appendix 1—figure 2A* and *Appendix 5—table 7*.

### 2. The "discarded" GCs.

The metagenomic "discarded" set includes 8,325,409 genes and 63,640 GCs classified as "bad" during the validation and refinement processes (Appendix 3), 19,911,324 singletons and 33,869,465 genes in 9,549,853 small GCs, that is clusters with less than 10 genes (Appendix 2), for a total of 62,106,198 genes and 29,524,817 GCs.

The genomic "discarded" dataset consists of 576,740 genes and 52,535 GCs classified as "bad", 5,558,438 singletons and 12,290,693 genomic genes found in 1,223,730 metagenomic discarded clusters. This last set of genes, labeled as "Other" in *Appendix 1—figure 2A*, includes 1,578,862 genomic genes found in the set of metagenomic "bad" clusters, 7,010,987 genomic genes found in the metagenomic small GCs and 3,700,844 genomic genes homologous to metagenomic singletons (*Appendix 5—table 8*).

The integration of the metagenomic and genomic "discarded" sets resulted in 80,532,069 genes and 35,135,790 GCs.

As described above, with the integration of genomic data we enriched metagenomic singletons and small GCs. This addition resulted in a set of 52,758 metagenomic singletons and 187,953 metagenomic small GCs becoming GCs with more than ten genes. We validated and classified the 240,711 GCs in this set. We obtained 223,229 good-quality GCs, divided into 17,383 K, 89,205 KWP, 109,636 GU and 7,005 EU.

**Appendix 5—table 8.** Overview of genomic genes found homologous to metagenomic genes.

| | Total | In MG good-quality GCs | In MG small GCs | In MG singletons | In MG bad-quality GCs |
|---|---|---|---|---|---|
| Genes | 67,446,376 | 55,155,683 | 7,010,987 | 3,700,844 | 1,578,862 |

## Evaluation of the integration strategy

Comparison of the results obtained with a one-step clustering approach against those from the integration approach.

We evaluated whether the integration of genomes into the metagenomic dataset would produce comparable results to the clustering at once of these two datasets. For the evaluation we randomly selected 20 M genes from the integrated dataset presented in the manuscript and we divided them based on their origin obtaining 17,049,704 metagenomic and 2,950,296 genomic genes. Next, we ran a one-step clustering of the metagenomic and genomic genes together as described in Methods - Determination of the gene clusters. Using the same test-datasets we then performed a two-steps clustering approach using the *clusterupdate* module of MMseqs2, as described in Methods - Determination of the gene clusters, which allows to integrate the genomic genes into the metagenomic clustering results.

The integration approach returned 0.57% more clusters than the one-step clustering and led to a reduction of 0.25% in the number of singletons (*Appendix 5—table 9*).

**Appendix 5—table 9.** Comparison of one-step and two-step clustering results in numbers.

| Approach | Total number of gene clusters | Of which singletons |
|---|---|---|
| One-step | 5,430,780 | 3,770,230 |
| Two-step | 5,462,006 | 3,779,961 |

We compared the clustering results using a custom R script. We evaluated the agreement applying the Adjusted Mutual Information (*Romano et al., 2015*),, which accounts for the higher number of small clusters found in the one-step clustering results. Overall, we obtained a high level of agreement (*Appendix 5—figure 2*) with an AMI of 0.96 (where 1 is the maximum).

The observed differences are due to new sequences not being included in the existing clusters but forming a new one. However, these gene clusters will later be aggregated during the gene cluster community inference.

## Appendix 6

### Gene cluster additional information

Additional information on the metagenomic and genomic (MG +GTDB) gene cluster dataset.

We retrieved a set of statistics for the MG +GTDB GC dataset, including the proportion of complete genes per cluster, the average gene length, the cluster level of darkness and disorder, and a cluster consensus taxonomic affiliation. The methods we applied to obtain these statistics are described in the Methods-Gene cluster characterization paragraph. Overall, the K category has the largest average GC size, 139.6 genes (and a max of 168,822 genes). The average GC size is then decreasing from the known to the unknown categories, with the EU presenting the smallest average size, with 17.36 genes per GC. Similarly, the K GCs have, on average, the longest genes (258.55 aa), followed by the GU (177.16 aa), the KWP (133.22 aa) and the EU (130.65 aa). The unknown categories (GU and EU) have the highest level of completion, that is the proportion of complete genes per GC. The KWP GCs contain the smallest percentage of complete genes. We evaluated the levels of darkness and disorder of the GCs using the information on the DPD (*Perdigão et al., 2017*) annotations (*Appendix 6—table 1*). The categories K, KWP and GU showed a degree of darkness inversely proportional to their functional characterization. Interestingly the KWP presented the highest level of disorder (*Supplementary file 2B*), while the proper characterization of these proteins is beyond the scope of this paper, our preliminary analyses suggest that KWP are enriched in intrinsically disordered proteins (*Habchi et al., 2014*, *Appendix 6—table 1*). These proteins, usually involved in signaling and regulatory functions, don't have a well-defined 3-D structure and they can adopt many different conformations.

**Appendix 6—table 1.** Number of MG +GTDB GCs annotated to the DPD per functional category.

| K | KWP | GU | EU |
|---|---|---|---|
| 374,555 | 8,874 | 22,135 | 0 |

We used the taxonomy of 214,392,608 genes to evaluate the taxonomic variation within a GC and generated consensus taxonomic annotations for 2,630,338 GCs. The GCs taxonomic variation is low at higher taxonomic levels and it steadily increases towards Genus and Species (*Supplementary file 2C*).

A general overview of the MG +GTDB main properties for the whole GCs dataset can be found in *Supplementary file 2B*.

## Appendix 7

### Gene cluster communities

Metagenomic and genomic gene cluster community inference detailed results.

We aggregated the gene clusters (GCs) into gene cluster communities (GCCs) based on their shared distant homologies, which couldn't be detected with the sequence similarity approach. The GCC inference, described in the Methods-Cluster communities inference section, was implemented and tuned on the known sequence space, which is constrained by the domain architectures (DAs). Then, we used the information retrieved for the known sequence space to aggregate the unknown GCs. Since the number of DAs in the known GCs may be inflated due to the fragmented nature of metagenomic genes, a key step for the inference process was the retrieval of a set of non-redundant DAs (Methods - **A set of non-redundant domain architectures** section).

We reduced the complete set of 29,341 Pfam DAs found in the metagenomic dataset, to 23,681 non-redundant DAs, and the 38,765 Pfam DAs found in the genomic dataset to 38,060 non-redundant DAs.

To find how the different clusters aggregate at the DA level, we then applied a combination of HMM-HMM searches and community identification using the Markov Cluster Algorithm (MCL) (**van Dongen and Abreu-Goodger, 2012**) (see Materials and methods - **Cluster communities inference**). MCL is very sensitive to the inflation value, which determines the granularity of the partitioning. The results of our iterative approach are summarized in the radar plots of *Appendix 7—figure 7–1*. We determined the best inflation value at 2.2 for the metagenomic dataset, value corresponding to the radar plot with the largest area (*Appendix 7—figure 1A*). This value is in agreement with the value empirically determined to be the optimal (**van Dongen and Abreu-Goodger, 2012**). The inference led to a set of 283,314 metagenomic GCCs out of ~2.9 M GCs, with a reduction rate of 90% (*Appendix 7—table 1A*).

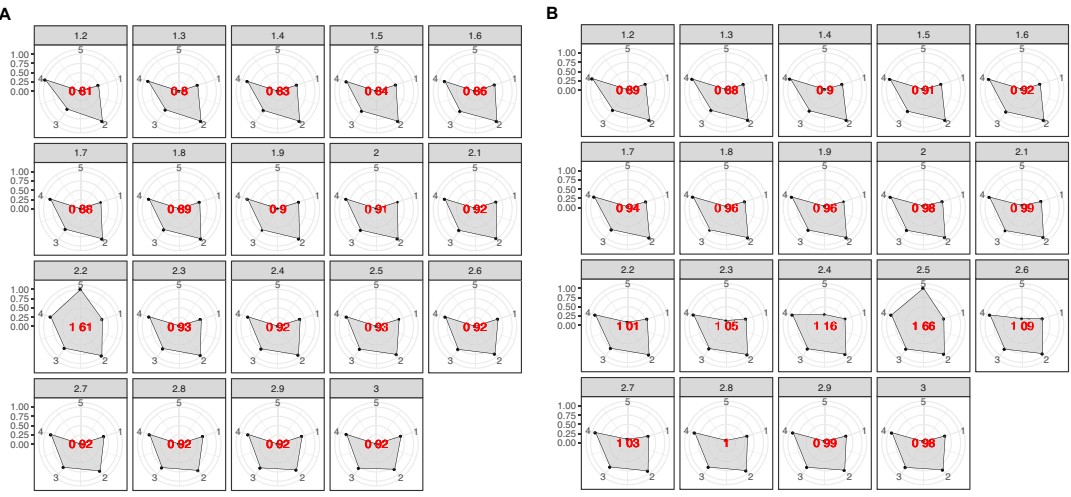

**Appendix 7—figure 1.** Radar plots used to determine the best MCL inflation value for the partitioning of the K into cluster components. The plots were built using a combination of five variables: 1 = proportion of clusters with one component and 2 = proportion of clusters with more than one member, 3 = clan entropy (proportion of clusters with entropy = 0), 4 = intra HHblits-Score/Aligned-columns (normalized by the maximum value), and 5 = number of clusters (related to the non-redundant set of DAs). (**A**) Metagenomic dataset. (**B**) Genomic dataset.

**Appendix 7—table 1.** Number of gene clusters, cluster communities, and reduction rate shown by functional category.

**(A) Metagenomic dataset (MG)**

|  | K | KWP | GU | EU | Total |
|---|---|---|---|---|---|
| Clusters | 1,050,166 | 632,453 | 1,121,809 | 135,829 | 2,940,257 |
| Communities | 24,181 | 64,938 | 146,100 | 48,095 | 283,314 |
| Reduction (%) | 97.7 | 89.73 | 86.98 | 64.59 | 90.36 |

**(B) Genomic dataset (GTDB)**

|  | K | KWP | GU | EU | Total |
|---|---|---|---|---|---|
| Clusters | 617,344 | 136,406 | 1,525,550 | 68,202 | 2,347,502 |
| Communities | 52,360 | 47,203 | 339,468 | 57,899 | 496,930 |
| Reduction (%) | 91.52 | 65.39 | 77.75 | 15.11 | 79.30 |

For the genomic dataset, we first identified the GCs with remote homologies to the metagenomic GCCs. To do this, we searched the genomic GC HMM profiles against the metagenomic ones, using HHblits (*Remmert et al., 2011*) (-n 2 -Z 10000000 -B 10000000 -e 1). We assigned the genomic GCs sharing a HHblits probability ≥50% and a bidirectional coverage >60% to the respective metagenomic GCCs. We processed the remaining genomic GCs through the GCC inference workflow. We determined the best inflation value at 2.5 (*Appendix 7—figure 1B*), which led to the inference of a total of 496,930 GCCs, with a reduction rate of 79% (*Appendix 7—table 1B*). The numbers of identified cluster GCCs for each category are shown in *Appendix 7—table 1*.

## Gene cluster community validation

The biological significance of the gene cluster communities (GCC) was tested by exploring their distribution within the phylogeny of proteorhodopsin and a set of ribosomal protein families.

## Methods

Analysis of the GCC distribution within the proteorhodopsin phylogeny.

We searched the proteorhodopsin (PR) HMM profiles from *Olson et al., 2018* against the K and KWP cluster consensus sequences, using the hmmsearch program of the HMMER software (version 3.1b2) (*Finn et al., 2011*). We filtered the results for alignment coverage >0.4 and e-value ≥1e-5. The filtered results were placed in the MicRhoDE PR tree (*Boeuf et al., 2015*) using pplacer (*Matsen et al., 2010*). Then we placed the query PR sequences into the MicRhode (*Boeuf et al., 2015*) PR tree. We de-duplicated the placed queries with CD-HIT (v4.6) (*Li and Godzik, 2006*) and we cleaned them from sequences with less than 100 amino acids using SEQKIT (v0.10.1) (*Shen et al., 2016*). Next, we calculated the best substitution model using the EPA-NG modeltest-ng (v0.3.5) (*Barbera et al., 2019*) and we optimized the MicRhoDE PR tree initial parameters and branch lengths using RAxML (v8.2.12) (*Stamatakis, 2014*). Afterward, we incrementally aligned the query PR sequences against the PR tree reference alignment using the PaPaRA (v2.5) software (*Berger and Stamatakis, 2012*). We divided the query alignment and the reference alignment using EPA-NG –split v0.3.5. We combined the PR tree with the related contextual data and the tree alignment, into a phylogenetic reference package using Taxtastic (v0.8.9), and we placed the PR query sequences in the tree using pplacer (v1.1.alpha19-0-g807f6f3) (*Matsen et al., 2010*) with the option -p (–keep-at-most) set to 20. We grafted the PR tree with the query sequences using Guppy, a tool part of pplacer. 3. As the last step, we assigned the PR Supercluster affiliation to the query sequence, transferring the annotation of its closest relative in the MicRhoDE tree (*Boeuf et al., 2015*) the R packages APE v5.3 and phanghorn v2.5.3 (*Schliep, 2011*).

Furthermore, we aligned the query sequences annotated as viral to the six viral PRs from *Needham et al., 2019*, using Parasail (*Daily, 2016*) (-a sg_stats_scan_sse2_128_16 t 8 c 1 x). We then built a sequence similarity network (SSN) using the sequence similarity values to weight the graph edges.

Analysis of standard and high-quality GCCs distribution within ribosomal protein families.

As an additional evaluation, the distributions of standard GCCs and HQ GCCs within ribosomal protein families were investigated and compared. The ribosomal proteins used for the analysis were

obtained combining the set of 16 ribosomal proteins from *Méheust et al., 2019* and those contained in the collection of bacterial single-copy genes of anvi'o (*Eren et al., 2021*), that can be downloaded from (https://github.com/merenlab/anvio/blob/master/anvio/data/hmm/Bacteria_71/genes.txt).

## Results

The results of both distribution analyses are shown in *Figure 2D and C*, respectively, and described in the main text.

We found 63 of the viral genes placed in the PR tree showing an average similarity of 50% with the viral PR of *Needham et al., 2019* (*Supplementary file 1H*). Additionally, we found two genes (from two TARA samples: TARA_093_SRF_0.22–3 and TARA_145_SRF_0.22–3) sharing a similarity of 100% with one of the Needham et al. PRs (ChoanoV2_VirRyml_1). These genes, however, were not placed in the PR tree.

## HMM-HMM homology network weighting metrics

Validation of the edge weight metrics used for the gene cluster homology network community inference.

## Methods

A critical step in the gene cluster community (GCC) inference relies on the determination of the edge weights for the GC HMM-HMM network. We tested two possible metrics to weight the GC homology network resulting from the all-vs-all HMM GC comparison with HHblits (*Remmert et al., 2011*): (1) the ratio between the HHblits score and the number of aligned columns (*HHblits-Score/Aligned-columns*), metric chosen in this paper; (2) the maximum(HHblits-probability x coverage), weight used in *Méheust et al., 2019*. In addition, we tested the two different metrics using the ribosomal protein families as reference. For this second test, we filtered the GCCs for those annotated to the 16 ribosomal proteins used in *Méheust et al., 2019*, and those contained in the collection of bacterial single-copy genes of Anvi'o (*Eren et al., 2021*), which can be downloaded from https://github.com/merenlab/anvio/blob/master/anvio/data/hmm/Bacteria_71/genes.txt. To then compare the two metrics, we used the functions of the R package *aricode* (https://github.com/jchiquet/aricode) (*Vinh et al., 2009*), which allow comparisons between clustering methods.

## Results

The results of the test of the different HHblits metrics used to weight the GC homology network are shown separately in *Appendix 7—figure 2* and the comparison in *Appendix 7—figure 3*. Both metrics present a very different behavior (*Appendix 7—figure 2*), the metric used in Méheust et al. is rescaling the *HHblits-probability* (*Appendix 7—figure 3*). While the *HHblits-probability* is useful for deciding if two HMMs are reliable homologs, it is not suitable for measuring similarities due to its dependence on the length of the alignment. On top of this, we can see how the *HHblits-Score/Aligned-columns* values present a similar and more homogenous distribution in all four categories, being more suitable for the MCL clustering.

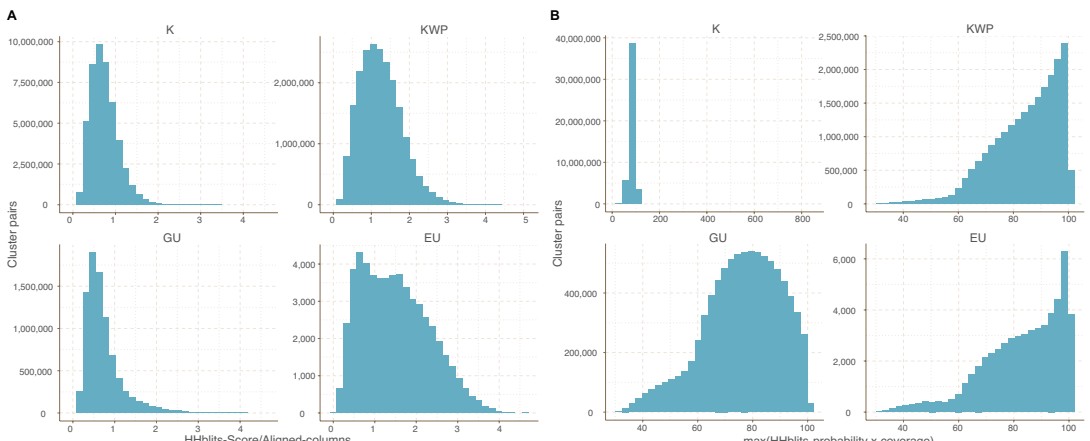

**Appendix 7—figure 2.** Cluster pairs distribution based on the metrics used to weight the gene cluster HMM-HMM homology network. (**A**) HHblits-Score/Aligned-columns (*Vanni et al., 2021*). (**B**) maximum(HHblits-probability x coverage) (Méheust et al.).

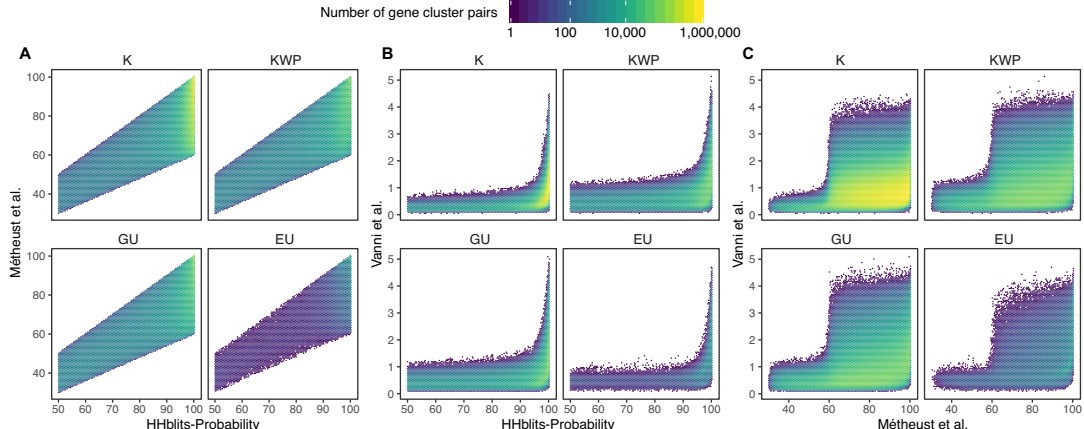

**Appendix 7—figure 3.** Determination of the edge-weight metrics for the GC HMM-HMM homology network. We tested the metrics used in Méheust et al. and this paper (Vanni et al.). The correlations between metrics are shown per functional category. The metric used by Méheust et al. corresponds to the maximum(HHblits-probability x coverage). The metric applied in this manuscript is *HHblits-Score/Aligned-columns*. (**A**) Comparison between the metric of Méheust et al. and the HHblits-Probability. (**B**) Comparison between the metric used in this manuscript and the HHblits-Probability. (**C**) Comparison between the metric used in this manuscript and the metric of Méheust et al.

Overall, our approach generated fewer GCCs, as can be observed in ***Appendix 7—figure 4***. Our clustering was found closer to the "*ground truth*" represented by the ribosomal protein families compared to the partitioning proposed by Méheust et al. The results from the comparison between the two clustering approaches and the ribosomal protein reference are reported in ***Appendix 7—table 2***.

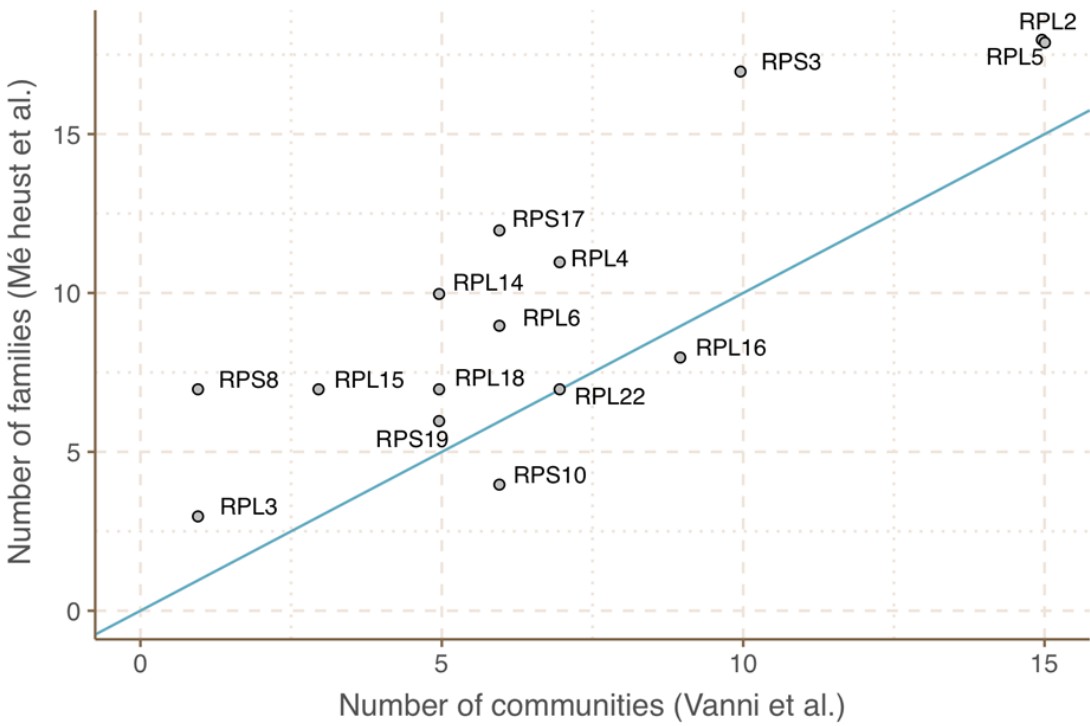

**Appendix 7—figure 4.** Agreement between the number of communities within ribosomal protein families between our approach and the one described in Méheust et al.

**Appendix 7—table 2.** Measures of similarity between the community inference approach proposed in this paper, the one used in Méheust et al and the "ground truth" represented by the ribosomal protein families.

|  | Vanni et al. vs meheust et al. | Vanni et al. vs ribosomal families | Meheust et al. vs ribosomal families |
|---|---|---|---|
| ARI | 0.915 | 0.944 | 0.906 |
| AMI | 0.928 | 0.916 | 0.878 |
| NVI | 0.101 | 0.0858 | 0.124 |
| NID | 0.0717 | 0.0841 | 0.122 |
| NMI | 0.928 | 0.916 | 0.878 |

Note: ARI = Adjusted Rand Index; AMI = Adjusted Mutual Information; NVI = Normalized Variation Information; NID = Normalized Information Distance; NMI = Normalized Mutual Information.

## Appendix 8

## Singletons effect on the sequence space diversity

Insights into the metagenomic and genomic singletons and their influence on the gene cluster rate of accumulation.

Singletons represent a significant fraction in both the metagenomic (60%) and genomic (55%) datasets. Although we discarded them from the primary analyses presented in this paper, we analyzed their composition in terms of functional categories. The analysis steps are described for the metagenomic singletons in Appendix 2, and, after the integration, we applied the same steps to the genomic singletons (*Appendix 8—table 1*). As shown in Appendix Note 1, the metagenomic singletons are highly represented by EU genes, while in the genomes we observed the majority of the singletons shared between GU and EU. In general, the singletons are characterized by a high percentage of genes of unknown function.

**Appendix 8—table 1.** Number of genomic singletons per functional category.

|       | K       | KWP     | GU        | EU        |
|-------|---------|---------|-----------|-----------|
| Genes | 473,460 | 896,127 | 2,528,370 | 1,660,481 |

We tested the singletons role in the rate of accumulation of GCs and GCCs as a function of the number of genomes and metagenomes, as shown in *Figure 3C and D* (to be compared with *Appendix 1—figure 5A and B*). For the metagenomic collector curves, we included only the singletons with a sample abundance of 8.36. This value corresponds to the mode sample abundance of the set of metagenomic singletons that became clusters with more than ten genes after the integration of the genomic data.

We observed that, excluding the 19,911,324 singletons from the metagenomic dataset, the accumulation curves of the GCs flatten and approach a plateau. The same effect is observed, excluding the set of 5,558,438 singletons from the genomic dataset (*Appendix 1—figure 5B*; *Appendix 8—table 2*).

**Appendix 8—table 2.** Minimum slope values for the collector curves.

**(A )Excluding singletons. In parenthesis, the number of genomes or metagenomes for the first occurrence of slope <1**

|         | Gene Clusters | | Gene cluster Communities | |
|---------|----------|--------|----------------|-----------------|
|         | metaG    | GTDB   | metaG          | GTDB            |
| Known   | 209.235  | 6.556  | 0.1344 (440)   | 0.07 (15,120)   |
| Unknown | 374.5147 | 5.851  | 0.1375 (600)   | 0.621 (27,690)  |

**(B) Including singletons (with a mode abundance in the samples of 8.36).**

|         | Gene Clusters | |
|---------|---------------|----------|
|         | metaG         | GTDB     |
| Known   | 1329.489      | 66.063   |
| Unknown | 4843.570      | 158.891  |

## Appendix 9

### Coverage of external databases

Analysis of the coverage, by our metagenomic dataset, of seven external microbial gene and gene cluster datasets.

#### Methods

We searched seven different state-of-the-art databases against our dataset of cluster HMM profiles. The different profile searches were all performed using the MMSeqs2 (version 8.fac81) *search* program (*Steinegger and Söding, 2017*), setting an e-value threshold of 1e-20 and a query coverage threshold of 60% (`-e 1e-20 --cov-mode 2 -c 0.6`). We kept the hits within 90% of the log10(best-e-value). Then we applied a majority vote function to retrieve the consensus functional category (K, KWP, GU or EU) for each search hit. In the end, the results were sorted by the lowest e-value and the largest query and target coverage to keep only the best hits.

We applied the described method to the following datasets: the Families of Unknown Functions (FUnkFams) (61,970 genes) (*Wyman et al., 2018*), the Pacific Ocean Virome (POV) (4,238,638 genes) (*Hurwitz and Sullivan, 2013*) and the Tara Ocean Virome (TOV) (6,642,187 genes) (*Brum et al., 2016*). The Genome Taxonomy Database (GTDB) (93,723,190 archaeal and bacterial genes) (*Parks et al., 2018*). The *MGnify* proteins from the EBI metagenomics database (release 2018_09) (*Mitchell et al., 2020*) (843,535,611 genes). The manually curated collection of 957 MAGs from TARA metagenomes (*Delmont et al., 2022*) (TARA MAGs) (2,288,202 genes), and the one made of 92 MAGs, from the fecal microbiota transplantation study (FMT MAGs) of *Lee et al., 2017* (188,983 genes). And also, the collection of unannotated genes with mutant phenotypes identified in *Price et al., 2018* (37,684 mutant genes).

### Results

We found our metagenomic GCs in all the main biomes defined by EBI metagenomics (*Appendix 1—figure 6*), with an overall coverage of 74% of the MGnify peptides (*Appendix 9—figure 1*). Our GCs also covered 62% of the FUnkFam genes of *Wyman et al., 2018*; 70% of the GTDB genes; and 85% of the genes tested for mutant phenotypes in *Price et al., 2018*. We also covered 50% of the Pacific Ocean Virome proteins, and 77% of the TARA Ocean Virome proteins, for overall coverage of 70% of the selected viral proteins. The majority of genes from both the FMT MAGs of *Lee et al., 2017* and the TARA MAGs of *Delmont et al., 2022*, were found homologous to genes in our dataset (91% and 77% respectively). With the only exception of the FUnkFams, and the mutant genes, for which we did not find any homology to EU GCs, the other datasets reported homologies to clusters from all four functional categories. Moreover, we found that 20% of the Wyman et al FUnkFams and 44% of the unknowns included in the RB-TnSeq experiments by *Price et al., 2018* belong to the known sequence space (*Appendix 9—table 1*).

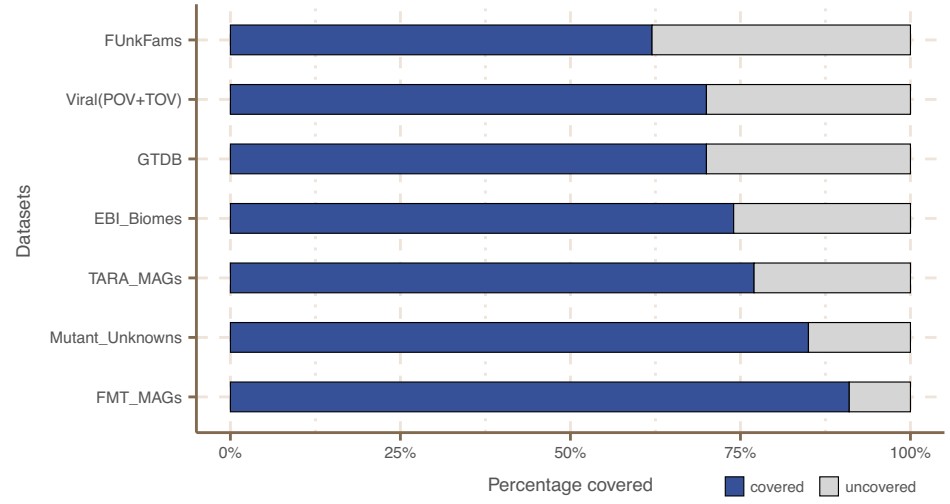

**Appendix 9—figure 1.** Coverage of external datasets. The bar plot is showing the proportion of covered genes in each of the seven datasets that were screened against the metagenomic set of clusters' HMM profiles.

**Appendix 9—table 1.** Re-classification of the unknowns identified in Wyman et al and Price et al.

| Study | Original unknown set | Covered fraction | Found as known | Found as unknown |
|---|---|---|---|---|
| Wyman et al. | 61,970 | 38,174 | 12,366 | 25,808 |
| Price et al. | 49,736 | 33,016 | 21,967 | 11,049 |

## Appendix 10

### EU gene cluster in metagenome-assembled genomes

Metagenome-assembled genomes (MAGs) as a resource to contextualize the environmental unknown gene clusters and cluster communities.

Overall, the MG + GTDB integrated cluster dataset contains 204,031 EU gene clusters (GCs) (grouped in 103,195 cluster communities (GCCs)). The EUs are divided into 127,032 metagenomic, 70,470 genomic, and 9,024, both metagenomic and genomic GCs. The last two subsets contain 52,231 (26%) EU found in GTDB metagenome-assembled genomes (MAGs). To test whether we could also place the subset of metagenomic EU in the context of MAGs, we searched the GCs of this set against the manually curated TARA Ocean MAG collection from *Delmont et al., 2022*.

In addition, we deepened the investigation of the metagenomic EU subset, focusing on the GCCs found broadly distributed in metagenomes according to the results of Levin's niche breadth analysis (*Figure 4*). The details of the metagenomic EU analysis are described below.

### Methods

We searched the metagenomic EU GCs HMM profiles, obtained from the cluster MSA using the *hhmake* program of the HH-SUITE (*Steinegger et al., 2019a*), against the set of 957 high-quality MAGs binned from the TARA Ocean prokaryotic dataset (*Delmont et al., 2022*). We performed the sequence-profile search using the MMSeqs2 *search* program (*Steinegger and Söding, 2017*), using `-e 1e-20 --cov-mode 2 -c 0.6`. We filtered the results to keep the hits within 90% of the log10(best-e-value). We applied a majority vote function to retrieve the consensus category for each hit. Then, we sorted the results by the smallest e-value and the largest query and target coverage to keep only the best hits. We then filtered the search results focusing on the broadly distributed EU GCs and GCCs. We retrieved MAG contigs containing the EU GCs and GCCs from the Anvi'o MAG profiles using the program *anvi-export-gene-calls* from Anvi'o v4 (Eren et al., 2015). We functionally annotated the contigs searching their genes against the Pfam database (v. 31.0) (*Finn et al., 2016*), using the *hmmsearch* program from the *HMMER* package (version: 3.1b2) (*Finn et al., 2011*), and complementing the search using Prokka (Seemann, 2014) in metagenomic mode. We then selected the contig with the lowest percentage of hypothetical proteins, and we extracted a region of 1 kb surrounding the genes mapping to the EU GCCs.

### Results

We found a total of 5,420 EU clusters homologous to 7,661 genes in the 691 TARA MAGs. These EU clusters belong to 4,365 GCCs. We kept only the 71 EU GCCs that showed a broad distribution in TARA samples. These GCCs contained 3,119 clusters and were found in 83 different TARA MAGs. Next, we examined the genomic neighborhood of the broad distributed EU on the MAG contigs. Investigating the genomic neighborhood can lead to the inference of a possible function of the EU. We selected the MAG most enriched with broadly distributed EU, which resulted in being the Atlantic North-West MAG "TARA_ANW_MAG_00076" (*Appendix 10—figure 1A*). This MAG contains 23 EU (0.3%) of its genes. It belongs to the bacterial order of *Flavobacteriales*. Of its 1,283 contigs, 317 include at least one EU. We functionally annotated these contigs with Prokka (and Pfam). Then, we sorted the contigs based on the proportion of genes annotated to hypothetical or characterized proteins, as shown in *Appendix 10—figure 1B*. The presence of genes of known function around the EU contributes to prove that these unknown genes are part of a real contig, and possibly an operon. Therefore, we selected for exploration, the contigs with the highest proportion of characterized genes, "TARA_ANW_MAG_00076_000000000672", with 7 characterized genes out of a total of 13 annotated genes. The contig with the second least number of hypothetical proteins was "TARA_ANW_MAG_00076_000000001247", which contained nine characterized genes out of 20. The contig "TARA_ANW_MAG_00076_000000000672" is shown in *Appendix 10—figure 1C* and highlighted in red are the two predicted genes with significant homology to the EU GCs, members of the broadly distributed EU GCCs eu_com_769 and eu_com_5081. Within their genomic neighborhood, we observe genes relating to nucleotide metabolism, DNA repair and phosphate regulation/sensing, including dUTPase, phoH and protein RecA. Gene placement in prokaryotic genomes is not random. Genes are grouped to increase transcriptional efficiency to respond to stimuli in the environment. Therefore, we can hypothesize that these EU have functions related to their neighboring genes.

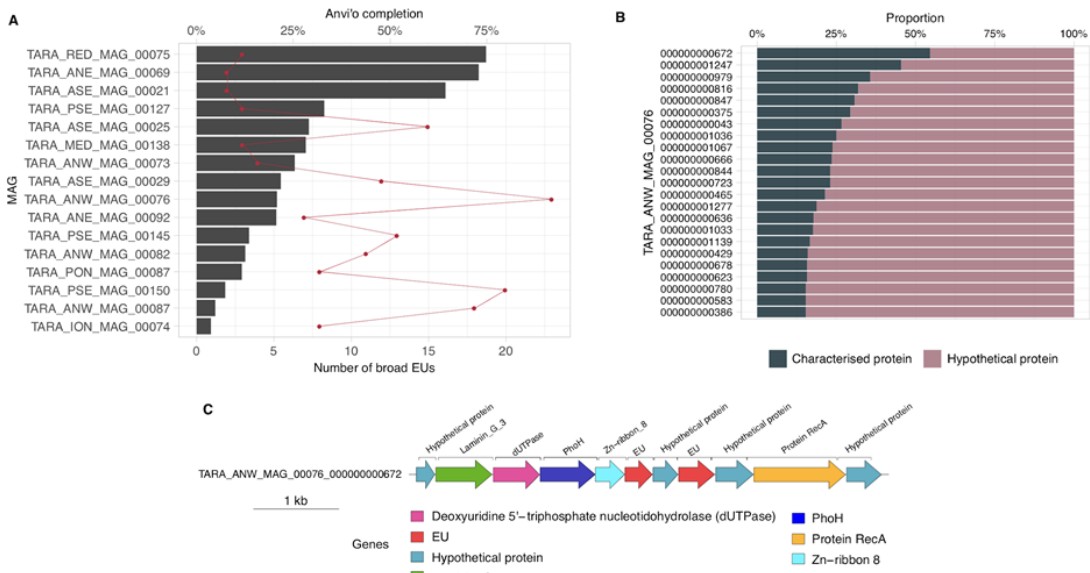

**Appendix 10—figure 1.** Broadly distributed EU mapping on TARA MAGs results. (**A**) . Histogram of TARA MAG percent completeness (checkM). The red line represents the number of EU found in the MAGs. (**B**) Contigs from TARA MAGs TARA_ANW_MAG_00076 in descending order of highest proportion of non-hypothetical gene content. (**C**) EU communities in the context of a MAG contig. Contig genomic neighborhood around two potential EU communities.

## Appendix 11

### Archaea gene cluster phylogenomic analysis

Gene clusters phylogenetic analysis - results for the archaeal genomes.

In the main text are shown the results for the gene clusters (GCs) phylogenetic analyses (clusters phylogenetic conservation and specificity) for the GTDB bacterial genomes. The same methods/analyses were applied for the archaeal genomes, and the results are presented here.

Out of the 230,340 GCs found in GTDB archaeal genomes, we identified 48,518 lineage-specific GCs (precision and sensitivity both ≥95%, *Mendler et al., 2019*). As seen for the Bacteria in *Figure 5A*, the number of known and unknown archaea lineage-specific GCs increases with the Relative Evolutionary Distance (*Parks et al., 2018*), with the differences between the known and the unknown fraction starting to be evident at the Family level (*Appendix 11—figure 1A*). The number of unknown lineage-specific GCs for Family, Genus and Species are 2,937, 12,966 and 21,002 respectively (*Supplementary file 1I*). A total of 34,893 GCs were phylogenetically conserved (*P* < 0.05), where 19,693 were known GCs and 15,200 were unknown GCs. Overall, the unknown GCs are more phylogenetically conserved than the known GCs (*Appendix 11—figure 1B*, *P* < 0.0001). However, considering only the lineage-specific clusters, we observe the opposite, the unknown GCs result in less phylogenetically conserved (*Appendix 11—figure 1B*). The GTDB archaeal genomes were also screened for prophages. In total, we identified 2,082 lineage-specific GCs in prophage genomic regions, and 86% of them resulted in clusters of unknown function (*Appendix 11—figure 1C*). To identify archaeal phyla enriched in unknown GCs, we partitioned the phyla based on the ratio of known to unknown GCs and vice versa (*Appendix 11—figure 1D*). We observed the same pattern found for bacterial phyla in *Figure 5D*, where the archaeal phyla with a larger number of MAGs are enriched in GCs of unknown function (*Appendix 11—figure 1D*).

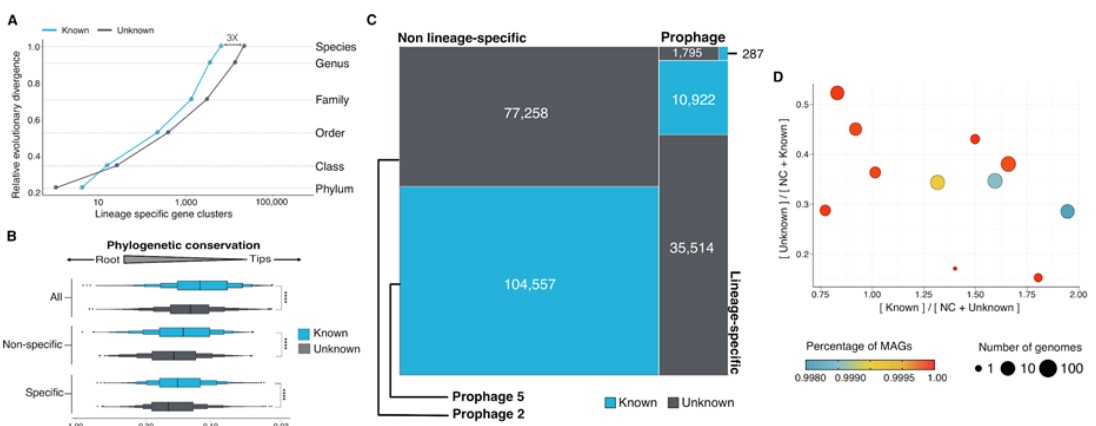

**Appendix 11—figure 1.** Phylogenomic exploration of the unknown sequence space in Archaea. (**A**) Distribution of the lineage-specific gene clusters by taxonomic level. Lineage-specific unknown gene clusters are more abundant at the lower taxonomic levels (genus, species). (**B**) Phylogenetic conservation of the known and unknown sequence space in 1,569 archaeal genomes from GTDB. We calculated the mean trait depth (add symbol $_D$) with the consenTRAIT algorithm and the lineage specificity using the F1-score approach from *Mendler et al., 2019*. We observe differences in the conservation between the known and the unknown sequence space for lineage- and non-lineage-specific gene clusters (paired Wilcoxon rank-sum test; all *P*-values < 0.0001). (**C**) The majority of the lineage-specific clusters are part of the unknown sequence space, being a small proportion found in prophages present in the GTDB genomes. (**D**) Known and unknown sequence space of the 1,569 GTDB archaeal genomes grouped by archaeal phyla. Phyla are partitioned based on the ratio of known to unknown gene clusters and vice versa from the set of genomes. Phyla enriched in Metagenomic assembled genomes (MAGs) have a higher proportion in gene clusters of unknown function.

## Appendix 12

### *Cand*. Patescibacteria lineage-specific gene clusters analysis

The investigation of the lineage-specific clusters was deepened, focusing on those specific to the Cand. Patescibacteria phylum (former Candidate Phyla Radiation-CPR) and analyzing their cluster distribution in both the Human and marine (TARA and Malaspina) metagenomes.

We found two GU clusters phylum-specific, and a total of 54,343 clusters of unknown function, lineage-specific within the *Cand*. Patescibacteria phylum (*Appendix 12—table 1*). The majority of this phylum members are particularly poorly understood microorganisms, mostly due to undersampling and the incompleteness of the available genomes. Therefore, we decided to investigate the distribution in the human and marine (TARA and Malaspina) metagenomes of all the clusters lineage-specific inside the *Cand*. Patescibacteria phylum (*Appendix 12—figure 1A*).

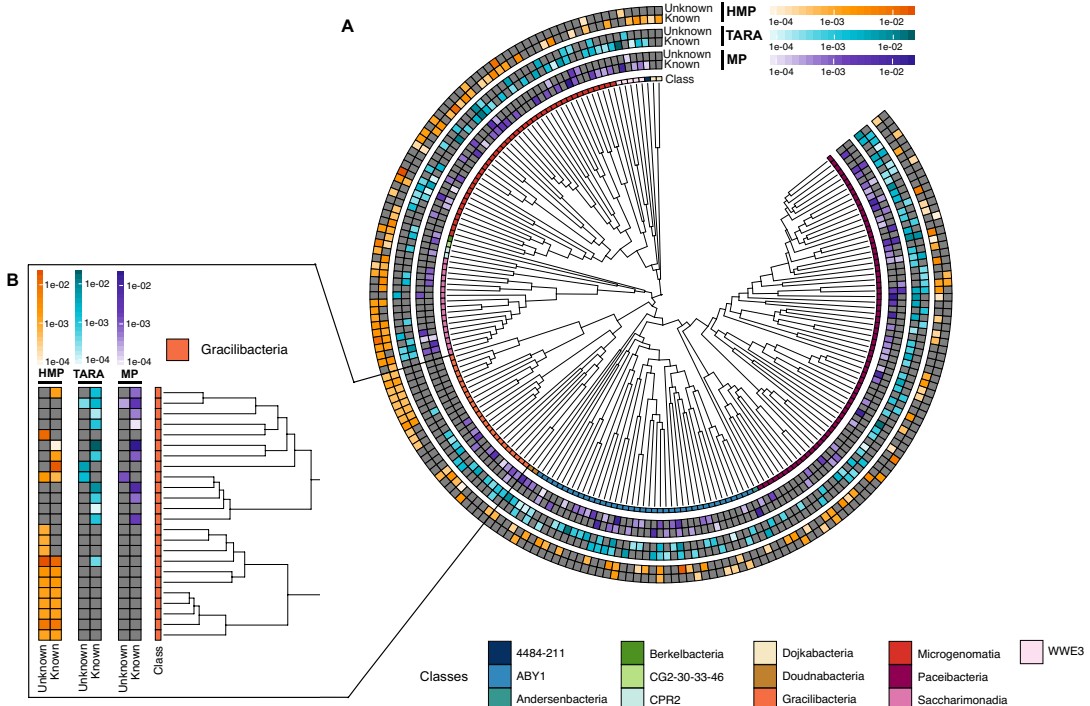

**Appendix 12—figure 1.** *Cand* Patescibacteria metagenomic lineage-specific clusters. (**A**) Phylogenetic tree of *Cand*. Patescibacteria genera, colored by classes. The heatmaps around the tree show the proportion of lineage-specific gene clusters of knowns and unknowns in the metagenomes from TARA, Malaspina and the HMP. (**B**) Metagenomic lineage-specific clusters in the class of *Gracilibacteria*.

**Appendix 12—table 1.** Number of lineage-specific clusters within the *Cand*. Patescibacteria phylum, at different taxonomic levels, subdivided by cluster categories.

| Taxonomic level | K | KWP | GU | EU |
|---|---|---|---|---|
| Phylum | 1 | 0 | 2 | 0 |
| Class | 11 | 0 | 6 | 0 |
| Order | 41 | 1 | 104 | 0 |
| Family | 452 | 9 | 1,443 | 13 |
| Genus | 625 | 98 | 6,649 | 338 |
| Species | 4,116 | 818 | 42,710 | 3,078 |

We chose to have a closer look at the class of *Gracilibacteria*, which shows to be present in both human and marine environments. The first genome for this class was retrieved in a hydrothermal

vent environment in the deep sea (*Rinke et al., 2013*). The same organisms were then also identified in an oil-degrading community (*Rinke et al., 2013*; *Sieber et al., 2019*) and as a part of the oral microbiome (*Espinoza et al., 2018*). As shown in *Appendix 12—figure 1B*, we found both known and unknown clusters lineage-specific to this class, distributed in human and marine metagenomes. Among these clusters, we observed cases of environment specificity. For instance, three clusters of unknowns were found exclusive to HMP samples. These clusters could be proposed as novel targets for human-health study since *Gracilibacteria* was found enriched in healthy individuals (*Espinoza et al., 2018*). We also observed lineage-specific clusters of known and unknown functions specific to the marine environment.

