## [Editor Report]

In this paper, the authors develop a sensitive and specific computational workflow for comprehensively summarizing known and unknown gene content across large collections of genomes and metagenomes. In addition to clustering and categorizing genes on a large scale, the authors show how to use their approach to both explore lineage-specific genes and generate hypotheses for the function of unknown genes.

---

## [Decision Letter]

**Decision letter after peer review:**

Thank you for submitting your article "Unifying the known and unknown microbial coding sequence space" for consideration by *eLife*. Your article has been reviewed by 2 peer reviewers, including C Titus Brown as the Reviewing Editor and Reviewer #1, and the evaluation has been overseen by Gisela Storz as the Senior Editor. The following individual involved in review of your submission has agreed to reveal their identity: Byron Smith (Reviewer #2).

Essential revisions:

1) We request that the authors restructure the results significantly. In particular, the results need more fine-grained paragraphs with more clarity about the results presented in each paragraph. Please see Reviewer 1 for many suggestions here!

2) Discuss the rationale for prioritizing metagenome-derived sequence clustering over clustering the reference-derived sequences. Similarly, please describe the reason for post-hoc annotation of clusters with UniProt and MGnify, rather than incorporation up front.

3) Please provide a rationale for choosing only two "sources" of metagenomes.

4) Please discuss the parameter selections, and/or cross-compare parameters with other kinds of analysis systems.

5) Add the Schloss and Handelsman reference as appropriate.

6) Please describe the broken-stick model in a bit more detail, focusing on how thresholds were chosen.

7) Consider further expanding the contribution of AGNOSTOS in relation to existing gene family frameworks (COG/EggNOG, etc.) that already capture the high-specificity side of the spectrum, vs the high-sensitivity/distant-homology issues tackled here.

8) Version the code and put it in Zenodo or some other archive.

*Reviewer #1 (Recommendations for the authors):*

I do think this paper is ground-breaking, but it is not clear to me that readers will be able to navigate clearly through the paper to understand that; although highly motivated readers will probably push through, my guess is that the impact will be greatly improved by substantial reorganization of the Results section.

Starting at the top,

I think the Introduction is quite good and have no big comments!

Results – need restructuring.

In general terms, I think the Results need more paragraphs, with better structure. Moreover, starting from a perspective of optimism and interest, I really struggled to move between the abundance of detailed results and some kind of understanding of their slightly broader meaning within the Results sections.

Conceptual framework:

Perhaps this first half of this could be better split between the bottom of the introduction and the beginning of the discussion? Many of the concepts introduced here only become clear after you've read later Results, which is not (in my quite strong view) not how Results should work.

There is a frequent and slightly frustrating tendency of the authors in this section to use …florid language in preparing us for their work; to my mind, I would prefer to be less prepared for how awesome the work is going to be, and experience it in the results and discussion and then be reminded of it in the Conclusions. Specifically, comments like these could be reconsidered as part of the Results and perhaps moved later:

– "subtle change of paradigm".

– "conceptual and technical foundations".

– "pushing search space beyond twilight zone of seq similarity".

Partitioning section:

I really appreciate the (excellent and expert) triage method.

This section was confusing – it seems like one long run-on paragraph?

One specific question – "72% of genes from GTDB already found, 22% created new GCs, 6% singletons." Question: does this tell us anything? If so, what?

(I have many similar questions; this is a common frustration of mine with the details in the Results. I can guess and think and intuit, but it requires an awful lot of effort to read each sentence! Which is not good.)

Beyond the twilight zone section:

I think this section should start with something like, "We next grouped GCs into gene cluster communities." As it is, the process and motivation is not clearly laid out, and I had to guess. (It looks great once you understand it!)

The results are clear, however!

Line 191, "One Known GCC." Is this a category, "Known"? I'm assuming so, because it's in italics? and capitalized? And while it definitely seems good that one GCC contains almost all of the PR, it would be good to state that clearly.

Paragraph breaks would be good here, too.

A smaller but highly diverse … section:

Lots of numbers. Little in the way of structure to help walk me through them.

Line 232, for example. Is this good? Bad? Interesting? Unexpected? (I can guess, but I want to know what you think it could mean, in fairly cut and dry terms.)

Ecological distribution section -

"Compared to what is reported by traditional genomic and metagenomic analysis approaches" – citation? numbers? comparison?

Paragraphs would be good here.

Unknown coding sequence space -

Meandering structure. I'm not sure what I'm supposed to understand from the specific sample analysis. Phages are responsible for all weirdness?

Fascinating observations in lines 304-307; good to leave as result.

Section – unknown coding space is lineage specific:

I have hard time grasping line 319. Perhaps flip sentence? "Fors GCs that are lineage specific and phylogenetically conserved, they are less conserved if they are unknown…" ok I still can't understand it. clarify?

Not really sure of implication of line 323.

The last paragraph here shows how powerful this is as an organizing principle.

Section – structured coding sequence space augments the interp of experimental data.

Really nice example of how to use this to dig into potential gene specific stuff. GETTING HERE is one (major) point of this paper! Nicely done.

Discussion –

line 414 is nice and clear.

line 422 – implemented?

The language is great, maybe using paragraphs or section headers would be good.

Separate into headers, give main point of each header.

Have "looking forward" in discussion.

Suggest organizing results to tie more clearly into points made in discussion?

Methods – please version the code and put it in Zenodo or some other archive, thanks!

*Reviewer #2 (Recommendations for the authors):*

– The choice to prioritize clustering of metagenomic sequences above references is surprising. If there is a computational reason not to cluster both together, that could be justified in the text.

– The impact of this work would be increased by extending the reference sequence corpus to which it is applied. Rather than post-hoc annotation of clusters with UniProt and MAGnify, these reference sets could be included in the initial clustering, "unifying" clustering and homology detection in these databases and simplifying the overall pipeline.

– Likewise, the availability of many additional metagenomes from a huge diversity of environments presents an opportunity to greatly extend this work, making the resulting database of broad interest across numerous fields. While the authors may have considered this and decided that it was out of scope, the possibility and its limitations should be explicitly discussed in the manuscript.

– Expanded comparisons of AGNOSTOS to traditional analyses and across parameters, or at least discussing parameter selections, would help readers to understand the decisions that were made.

– The manuscript as it is currently written ignores previous applications of de novo sequence clustering in the analysis of metagenomes (e.g. Schloss and Handelsman, 2008, "A statistical toolbox for metagenomics: assessing functional diversity in microbial communities"). Citing these and putting the current work in the context of other approaches to protein families (COGs, KOs, etc.) would be valuable for readers.

– I did not find the four-class (K/KWP/GU/EU) conceptual framework to be particularly helpful. I think it conflates two orthogonal concepts: (1) the existence of homologues in reference genomes, and (2) homology to sequences with characterized function. As such, it complicated explanations of downstream analyses that focused on just one of those two axes.

– I found many of the explanations in the associated blog post to be easier to follow than those in the introduction. Presenting some of the major concepts earlier and in a more explicit way would be helpful. For instance:

– Replacing "CDS" with "protein" throughout (and clarifying early that these are predicted from DNA sequences using prodigal) and replace "CDS-space" with "proteins" or "protein sequences".

– Explicitly state that "gene clusters" refer to homologous groups. There is a risk that readers will instead think of e.g. "biosynthetic gene clusters" or other groupings of genes based on proximity in a linear sequence.

Clarify ambiguous explanations with simpler phrasing. E.g.:

– "This inability to handle shades of the unknown is an immense impediment to realizing the potential for discovery of microbiology and molecular biology at large […]" (lines 69-71)

– "[…] adds context to vast amounts of unknown biology, providing an invaluable resource to understand the unknown functional fraction better and boost the current methods for its experimental characterization." (lines 87-89)

– Flesh out the distinction between classes more clearly in the introduction, for instance replace lines 101-103 with: 'Only a fraction of sequences predicted to code for proteins possess homology to domains of known function described by Pfam, here we refer to this class as the "Known" (K) fraction. A portion of sequences without Pfam domains of known function can nonetheless be annotated based on homology to characterized proteins, and are classified as "Known without Pfam" (KWP). Finally, sequences not annotated by either of these criteria may be either found in the sequenced genome of an (isolated) bacterium: "Genomic Unknown" (GU), or may have only been observed in environmental samples: "Environmental Unknown" (EU). These four reflect a hierarchy of increasingly "unknown" classifications.'

– Several results are reported as comparisons, but where the "other" is not explicit. E.g. "is phylogenetically more conserved" (line 92). Phylogenetically more conserved than what?

– Similarly: "is smaller than expected" (line 91) and "creating more GCs than expected" (line 482-483). Expected by whom?

– Several references to cluster thresholding using the "broken-stick model" need better explanations. I was not familiar with this approach, and would benefit from a clear description of how thresholds were chosen using this method.

[Editors' note: further revisions were suggested prior to acceptance, as described below.]

Thank you for resubmitting your work entitled "Unifying the known and unknown microbial coding sequence space" for further consideration by *eLife*. Your revised article has been evaluated by Gisela Storz (Senior Editor) and a Reviewing Editor.

The manuscript has been improved but there are some remaining issues that need to be addressed, as outlined below:

Essential revisions:

(1) The abstract should be revised somewhat. The sentence starting with the "We quantify the extent…" could be fleshed out with a bit more detail and numbers – perhaps some highlights of the quantification? The next sentence on Patescibacteria stands out because this result seems to be only cursorily discussed in the paper – at the very least the same number (283k) should be mentioned in the Results. Please adjust.

(2) The phrase "lineage-specific at the Species level" is used a few times and is unclear to me – e.g. Line 100: is the intent to say that the unknown fraction of genes is largely restricted to multiple members of individual species? Also, around line 493. Please clarify.

3) Some of the subjective language commented on by both reviewers in initial reviews remains and is not adequately supported by the results. For instance, line 98-101 "By contextualizing the different categories with information from several sources (Figure 1C), we provide an invaluable resource […]". While this may prove true with time, I don't believe that the authors have justified this degree of elevation. Phrasing such as, "We hope that this will prove an invaluable resource […]" would be more appropriate.

(4) Please consider how to improve the description of the pipeline. Some questions we had:

(A) Is this a data product, an analysis tool (slash pipeline), a conceptual framework, or a biological result? In their response to reviews it seems that it is primarily an analysis pipeline with a small number of biological results used to demonstrate its utility; This is not immediately apparent to readers, nor do the authors spend much text explaining to a reader why they should apply it to their own data.

(B) Which of the steps described in the results is built into the AGNOSTOS pipeline and which is applied post-hoc to the results of that pipeline? Clean delineation between the featured tool and additional analyses would improve reader understanding.

(C) Are the hyperparameters arrived at by the authors intended to be used on future users' data as well? If not, are the methods used by the authors for hyper-parameter selection built into the software, or would users need to re-build those themselves?

(D) In Figure 3, panels B and C, which of the K, KWP, GU, or EU categories are included under each of the "Known" and "Unknown" labels? Are both K and KWP combined under known and both GU and EU under unknown? This is one example of where the four-category conceptual framework feels like it might be better described as two axes.

We also suggest the authors consider addressing the following two questions in their revisions:

1. While I now understand the justification for picking the MCL hyperparameter for super-clusters based on the Known fraction alone, it would nonetheless be valuable to use this higher-order clustering to further the search for distant homology. Given that the authors prioritize sensitivity over specificity everywhere else in the paper, this feels like a missed opportunity.

2. In their response to reviewers, the authors express confidently that, "The results of clustering both data sets together or updating the existing gene clusters will be almost identical, but by doing it in two steps, one can track the dynamics of the singletons, the stability of the gene clusters and many other interesting processes that can provide a better understanding of the data." If this is true, I'd be very interested to see it demonstrated. The ability to "stream" in additional genes to the analysis and get the same result has major (positive) implication for the computational scalability of this analysis. Given the size of both current and future metagenomic datasets, this would indeed be a ‘very’ valuable feature of the pipeline.

---

## [Author Response]

Essential revisions:Reviewer #1 (Recommendations for the authors):I do think this paper is ground-breaking, but it is not clear to me that readers will be able to navigate clearly through the paper to understand that; although highly motivated readers will probably push through, my guess is that the impact will be greatly improved by substantial reorganization of the Results section.

Thank you very much for the suggestions. We have modified the results and Discussion sections and included the suggestions from Reviewer 1 to make it clearer.

Starting at the top,I think the Introduction is quite good and have no big comments!Results – need restructuring.

Thank you very much for the suggestions. We have restructured the results following your suggestions described in detail in the following comments.

In general terms, I think the Results need more paragraphs, with better structure. Moreover, starting from a perspective of optimism and interest, I really struggled to move between the abundance of detailed results and some kind of understanding of their slightly broader meaning within the Results sections.Conceptual framework:Perhaps this first half of this could be better split between the bottom of the introduction and the beginning of the discussion? Many of the concepts introduced here only become clear after you've read later Results, which is not (in my quite strong view) not how Results should work ;.

We have split the first part of this section between the introduction and the discussion. We hope now the different concepts are more easily understandable and the reader will be able to follow the story better from the beginning. Thanks for the suggestion.

There is a frequent and slightly frustrating tendency of the authors in this section to use …florid language in preparing us for their work; to my mind, I would prefer to be less prepared for how awesome the work is going to be, and experience it in the results anddiscussion and then be reminded of it in the Conclusions. Specifically, comments like these could be reconsidered as part of the Results and perhaps moved later:– "subtle change of paradigm".– "conceptual and technical foundations".– "pushing search space beyond twilight zone of seq similarity".

We modified a bit the language and moved these parts (almost completely) to the discussion.

Partitioning section:I really appreciate the (excellent and expert) triage method.

Thanks!

This section was confusing – it seems like one long run-on paragraph?

We have split this section into multiple paragraphs, hopefully now it is more clear and reads

One specific question – "72% of genes from GTDB already found, 22% created new GCs, 6% singletons." Question: does this tell us anything? If so, what?

We rewrote this section to make these results clearer. Briefly, here we describe how most of the genes in GTDB were already present in the metagenomic dataset, and the majority of them were part of the known sequence space. A small fraction of the genes from GTDB only created singleton gene clusters.

(I have many similar questions; this is a common frustration of mine with the details in the Results. I can guess and think and intuit, but it requires an awful lot of effort to read each sentence! Which is not good.)

We hope that with the new text this is will not be the case and everything is more concise.

Beyond the twilight zone section:I think this section should start with something like, "We next grouped GCs into gene cluster communities." As it is, the process and motivation is not clearly laid out, and I had to guess.(It looks great once you understand it!)

We modified the start of the section to make the motivation clear from the beginning as suggested. Thank you for the recommendation.

The results are clear, however!Line 191, "One Known GCC." Is this a category, "Known"? I'm assuming so, because it's in italics? and capitalized? And while it definitely seems good that one GCC contains almost all of the PR, it would be good to state that clearly.

We rewrote it and stated the results clearly.

Paragraph breaks would be good here, too.A smaller but highly diverse … section:Lots of numbers. Little in the way of structure to help walk me through them.

We rewrote this section to make these results clearer. We hope now it is much easier to follow.

Line 232, for example. Is this good? Bad? Interesting? Unexpected? (I can guess, but I want to know what you think it could mean, in fairly cut and dry terms.)

We expanded the explanation of the amino acid level analysis and highlighted its relevance on the potential number of DUFs that remain undiscovered, when we compare the known and unknown sequence space.

Ecological distribution section -"Compared to what is reported by traditional genomic and metagenomic analysis approaches" – citation? numbers? comparison?

We clarified the sentence and added relevant references.

Paragraphs would be good here.

We separated the text into paragraphs, and make the text easier to follow.

Unknown coding sequence space -Meandering structure. I'm not sure what I'm supposed to understand from the specific sample analysis. Phages are responsible for all weirdness?

We reorganized and improved the text to show the potential of the ratio between the unknown and known GCs to identify samples that might contain more sequence novelty. As expected, in our set of samples, the ones with a higher load of viral sequences show more novelty.

Fascinating observations in lines 304-307; good to leave as result.

Thanks, there is still a whole world of unknowns to explore.

Section – unknown coding space is lineage specific:I have hard time grasping line 319. Perhaps flip sentence? "Fors GCs that are lineage specific and phylogenetically conserved, they are less conserved if they are unknown…" ok I still can't understand it. clarify?

We clarified the different concepts and streamlined the description of the results. We hope that know reads more easily.

Not really sure of implication of line 323.

We added an explanation of the potential cofounding effect of prophages when analyzing the unknowns in genomic sequences. We performed the prediction of the prophage regions on all GTDB genomes to quantify the number of genomic unknowns with a potential foreign origin.

The last paragraph here shows how powerful this is as an organizing principle.Section – structured coding sequence space augments the interp of experimental data.Really nice example of how to use this to dig into potential gene specific stuff. GETTING HERE is one (major) point of this paper! Nicely done.

Thanks!

Discussion –line 414 is nice and clear.line 422 – implemented?The language is great, maybe using paragraphs or section headers would be good.Separate into headers, give main point of each header.Have "looking forward" in discussion.Suggest organizing results to tie more clearly into points made in discussion?

We have split the discussion into paragraphs and rewrote it to make it easier to follow. We also added some parts of the results into the discussion as suggested above.

Methods – please version the code and put it in Zenodo or some other archive, thanks!

Done, thanks for pointing this out.

Reviewer #2 (Recommendations for the authors):– The choice to prioritize clustering of metagenomic sequences above references is surprising. If there is a computational reason not to cluster both together, that could be justified in the text.

We explained better in the text the rationale for the different decisions we took. Briefly, by using metagenomic data instead of references as initial data, we can show the robustness of AGNOSTOS to deal with noisy and incomplete data. Most of the studies that will use our methods will use data derived from metagenomes (contigs or MAGs), and it is crucial to show that our validation and refinement steps perform as expected. Later, we added the GTDB sequences to show the capabilities of AGNOSTOS to enrich already processed data. The results of clustering both data sets together or updating the existing gene clusters will be almost identical, but by doing it in two steps, one can track the dynamics of the singletons, the stability of the gene clusters and many other interesting processes that can provide a better understanding of the data. Also one can “paint” the gene clusters by integrating other sources of data, like enzymatic sequences like we did in Dittmar et al., 2021 where we integrated CAZymes, KEGG and other data sources in the seed database used in this manuscript.

– The impact of this work would be increased by extending the reference sequence corpus to which it is applied. Rather than post-hoc annotation of clusters with UniProt and MAGnify, these reference sets could be included in the initial clustering, "unifying" clustering and homology detection in these databases and simplifying the overall pipeline.

The main objective of the manuscript is to propose a new way of analyzing (meta)genomic data, and not providing a data product in a form of a database, even though we distribute our seeds. Each user can create its own database using sequence data specific to answer their research questions.

We use UniRef as part of the classification step in AGNOSTOS as UniRef provides a controlled vocabulary for the unknown. This classification step also includes a two-step search like the one used by the 2blca that we use to refine the annotations terms and assess the uncertainty of the annotations of the unknown terms. Unfortunately, this step cannot be integrated in the clustering itself.

Regarding MGnify, we performed this search to demonstrate that although we used only two sources of metagenomic data, the known space is already very well covered.

– Likewise, the availability of many additional metagenomes from a huge diversity of environments presents an opportunity to greatly extend this work, making the resulting database of broad interest across numerous fields. While the authors may have considered this and decided that it was out of scope, the possibility and its limitations should be explicitly discussed in the manuscript.

As a proof of concept of AGNOSTOS, we used two of the most well-known metagenomic datasets, HMP and TARA Oceans. While we agree with the reviewer that adding more metagenomes would provide an incredible resource to investigate the known and unknown, it falls out of the scope of this manuscript, were we wanted to demonstrate the flexibility and possibilities of AGNOSTOS.

– Expanded comparisons of AGNOSTOS to traditional analyses and across parameters, or at least discussing parameter selections, would help readers to understand the decisions that were made.

We clarified in the text the rationale behind some of the decisions and parameter selection regarding the clustering, validation, refinement and classification.

– The manuscript as it is currently written ignores previous applications of de novo sequence clustering in the analysis of metagenomes (e.g. Schloss and Handelsman, 2008, "A statistical toolbox for metagenomics: assessing functional diversity in microbial communities"). Citing these and putting the current work in the context of other approaches to protein families (COGs, KOs, etc.) would be valuable for readers.

We added the reference, thank you very much. We clarified in the text that our approach is complementary to the existing orthologous-like databases like COG or KO, and that one can use these databases to enrich the gene clusters in a similar fashion as we did in Delmont et al., 2020.

– I did not find the four-class (K/KWP/GU/EU) conceptual framework to be particularly helpful. I think it conflates two orthogonal concepts: (1) the existence of homologues in reference genomes, and (2) homology to sequences with characterized function. As such, it complicated explanations of downstream analyses that focused on just one of those two axes.

We reformulated the description of the four basic categories used for the partitioning to make it clearer. Briefly, the purpose of partitioning the knowns between K and KWP is (1) to have a category to account for those proteins that have a conserved domain, but not included yet in PFAM, and (2) and most importantly, to accommodate the intrinsically disordered proteins and small proteins. These proteins lack any structure, but their function is known. Regarding the unknowns, EUs are very useful to infer the novelty of a metagenome or to identify novel regions of a single genome based on non-uniform distribution of EUs.

– I found many of the explanations in the associated blog post to be easier to follow than those in the introduction. Presenting some of the major concepts earlier and in a more explicit way would be helpful. For instance:– Replacing "CDS" with "protein" throughout (and clarifying early that these are predicted from DNA sequences using prodigal) and replace "CDS-space" with "proteins" or "protein sequences".

We changed CDS-space for sequence space in Figure 1, and used *genes* instead of CDS-space on the main text.

– Explicitly state that "gene clusters" refer to homologous groups. There is a risk that readers will instead think of e.g. "biosynthetic gene clusters" or other groupings of genes based on proximity in a linear sequence.

We added a clarification to avoid the confusion.

Clarify ambiguous explanations with simpler phrasing. E.g.:– "This inability to handle shades of the unknown is an immense impediment to realizing the potential for discovery of microbiology and molecular biology at large […]" (lines 69-71)– "[…] adds context to vast amounts of unknown biology, providing an invaluable resource to understand the unknown functional fraction better and boost the current methods for its experimental characterization." (lines 87-89)– Flesh out the distinction between classes more clearly in the introduction, for instance replace lines 101-103 with: 'Only a fraction of sequences predicted to code for proteins possess homology to domains of known function described by Pfam, here we refer to this class as the "Known" (K) fraction. A portion of sequences without Pfam domains of known function can nonetheless be annotated based on homology to characterized proteins, and are classified as "Known without Pfam" (KWP). Finally, sequences not annotated by either of these criteria may be either found in the sequenced genome of an (isolated) bacterium: "Genomic Unknown" (GU), or may have only been observed in environmental samples: "Environmental Unknown" (EU). These four reflect a hierarchy of increasingly "unknown" classifications.'

As part of the re-structuring of the results we added the reviewer’s suggestions and made the text easier to follow.

– Several results are reported as comparisons, but where the "other" is not explicit. E.g. "is phylogenetically more conserved" (line 92). Phylogenetically more conserved than what?

We modified the text to make the comparisons more explicit.

– Similarly: "is smaller than expected" (line 91) and "creating more GCs than expected" (line 482-483). Expected by whom?

Same as previous point.

– Several references to cluster thresholding using the "broken-stick model" need better explanations. I was not familiar with this approach, and would benefit from a clear description of how thresholds were chosen using this method.

We added in the methods a more in-depth explanation of the broken-stick model to make it clearer for the reader.

[Editors' note: further revisions were suggested prior to acceptance, as described below.]

The manuscript has been improved but there are some remaining issues that need to be addressed, as outlined below:Essential revisions:(1) The abstract should be revised somewhat. The sentence starting with the "We quantify the extent…" could be fleshed out with a bit more detail and numbers – perhaps somehighlights of the quantification? The next sentence on Patescibacteria stands out because this result seems to be only cursorily discussed in the paper – at the very least the same number (283k) should be mentioned in the Results. Please adjust.

We rewrote the abstract and introduced some of the main results. We also modified in the results and discussion the *Cand*. Patescibacteria results.

(2) The phrase "lineage-specific at the Species level" is used a few times and is unclear to me – e.g. Line 100: is the intent to say that the unknown fraction of genes is largely restricted to multiple members of individual species? Also, around line 493. Please clarify.

We have clarified the term by first introducing it as taxonomically restricted and added a new reference (Johnson, 2018).

(3) Some of the subjective language commented on by both reviewers in initial reviews remains and is not adequately supported by the results. For instance, line 98-101 "By contextualizing the different categories with information from several sources (Figure 1C), we provide an invaluable resource […]". While this may prove true with time, I don't believe that the authors have justified this degree of elevation. Phrasing such as, "We hope that this will prove an invaluable resource […]" would be more appropriate.

We toned it down and added references where AGNOSTOS has been already applied to discover a new lineage of giant viruses (Gaïa et al., 2021), helped to understand the functional repertoire convergence of distantly related eukaryotic plankton lineages (Delmont et al., 2021) and shed light on the ecological role of the genes of unknown function in soils (Holland-Moritz et al., 2021).

After reviewing the previous round of revisions, we amended the issues raised by the reviewers that might still remain unresolved.

(4) Please consider how to improve the description of the pipeline. Some questions we had:(A) Is this a data product, an analysis tool (slash pipeline), a conceptual framework, or a biological result? In their response to reviews it seems that it is primarily an analysis pipeline with a small number of biological results used to demonstrate its utility; This is not immediately apparent to readers, nor do the authors spend much text explaining to a reader why they should apply it to their own data.

We updated the text to provide a better explanation of the different parts of the manuscript. Briefly, we introduce a new conceptual framework for analyzing (meta)genomic data, how to translate it into a computational workflow, and examples of how it can be used to answer biological questions.

B) Which of the steps described in the results is built into the AGNOSTOS pipeline and which is applied post-hoc to the results of that pipeline? Clean delineation between the featured tool and additional analyses would improve reader understanding.

The text has been updated to clarify where we describe AGNOSTOS and its results; and the part where we use those results to provide ecological and evolutionary insights into the unknown fraction. We combined the sections "A computational workflow to unify the known and the unknown coding sequence space" and "Partitioning and contextualizing the coding sequence space of genomes and metagenomes" in the section "AGNOSTOS, a computational workflow to unify the known and the unknown sequence space". Hence, the main workflow description and results related to the data processing by AGNOSTOS are together. We renamed the section that describes the inference of GCCs as "Beyond the twilight zone with AGNOSTOS, communities of gene clusters" as it needs a more detailed explanation.

Furthermore, we renamed "A smaller but highly diverse unknown coding sequence space" to "AGNOSTOS uncovers a smaller yet highly diverse unknown sequence space" and added a short introduction to clarify that this section and the following section are analyses one can do downstream with the results generated by AGNOSTOS.

(C) Are the hyperparameters arrived at by the authors intended to be used on future users' data as well? If not, are the methods used by the authors for hyper-parameter selection built into the software, or would users need to re-build those themselves?

We have updated the text emphasize that all parameters are automatically estimated by AGNOSTOS.

(D) In Figure 3, panels B and C, which of the K, KWP, GU, or EU categories are included under each of the "Known" and "Unknown" labels? Are both K and KWP combined under known and both GU and EU under unknown? This is one example of where the four-category conceptual framework feels like it might be better described as two axes.

We clarified in the text that we combined the categories for the sake of simplicity for this analysis. Indeed, here the two axes are better descriptors. We aim to provide a flexible framework that can use two or four axes depending on the question we are trying to answer.

We also suggest the authors consider addressing the following two questions in their revisions:1. While I now understand the justification for picking the MCL hyperparameter for super-clusters based on the Known fraction alone, it would nonetheless be valuable to use this higher-order clustering to further the search for distant homology. Given that the authors prioritize sensitivity over specificity everywhere else in the paper, this feels like a missed opportunity.

While we resolved to not perform additional analyses, we appreciate the reviewer’s point and updated the text so that future research can consider including this additional revenue our study makes possible.

We would also like to emphasize that even though our methods are very sensitive, they also maintain a high specificity, as shown by the different results described in the main text and in Appendices 1 and 2, as well as the Supplementary File 2C.

2. In their response to reviewers, the authors express confidently that, "The results of clustering both data sets together or updating the existing gene clusters will be almost identical, but by doing it in two steps, one can track the dynamics of the singletons, the stability of the gene clusters and many other interesting processes that can provide a better understanding of the data." If this is true, I'd be very interested to see it demonstrated. The ability to "stream" in additional genes to the analysis and get the same result has major (positive) implication for the computational scalability of this analysis. Given the size of both current and future metagenomic datasets, this would indeed be a ‘very’ valuable feature of the pipeline.

We have expanded Appendix 5 to demonstrate how the incremental clustering computed by the MMseqs2 *clusterupdate* module produces almost identical results to clustering in one step. Using the dataset in the manuscript, we randomly selected 20M genes and separated them into those that came from metagenomes (17,049,704 genes) and those that came from GTDB (2,950,296). We performed a clustering with both sets of genes combined (one-step) and then used the integrative approach used in the manuscript (two-step). The two-step clustering produced 0.57% more clusters than the one-step clustering, and the number of singletons decreased a 0.25%. As a result, we have evaluated the agreement between the two clustering strategies using Adjusted Mutual Information, which takes into account the fact that the reference clustering (one-step) is unbalanced and contains many small clusters (Romano et al., 2015), obtaining an agreement of 0.96 (where 1 is full agreement). Observed differences are primarily due to the fact that some of the new sequences were not included in an existing cluster and therefore formed a new one. These gene clusters will be combined when we infer the gene cluster communities. Additionally, we added a reference to the AGNOSTOS-DB preprint, showing the gene clusters' stability after multiple integration steps.

References

Delmont TO, Gaia M, Hinsinger DD, Fremont P, Vanni C, Guerra AF, Murat Eren A, Kourlaiev A, d’Agata L, Clayssen Q, Villar E, Labadie K, Cruaud C, Poulain J, Da Silva C, Wessner M, Noel B, Aury J-M, Coordinators TO, de Vargas C, Bowler C, Karsenti E, Pelletier E, Wincker P, Jaillon O. 2021. Functional repertoire convergence of distantly related eukaryotic plankton lineages revealed by genome-resolved metagenomics. *bioRxiv*. doi:10.1101/2020.10.15.341214

Gaïa M, Meng L, Pelletier E, Forterre P, Vanni C, Fernendez-Guerra A, Jaillon O, Wincker P, Ogata H, Delmont TO. 2021. Discovery of a class of giant virus relatives displaying unusual functional traits and prevalent within plankton: the Mirusviricetes. *bioRxiv*. doi:10.1101/2021.12.27.474232

Holland-Moritz H, Vanni C, Fernandez-Guerra A, Bissett A, Fierer N. 2021. An ecological perspective on microbial genes of unknown function in soil. *bioRxiv*. doi:10.1101/2021.12.02.470747

Johnson BR. 2018. Taxonomically Restricted Genes Are Fundamental to Biology and Evolution. *Front Genet* 9:407.

Romano S, Vinh NX, Bailey J, Verspoor K. 2015. Adjusting for Chance Clustering Comparison Measures. *arXiv [statML]*.